**Gas transfer velocities of CO$_2$ in subtropical monsoonal climate streams and**
**small rivers**
**Siyue Li[a]\*, Rong Mao[a], Yongmei Ma[a], Vedula V. S. S. Sarma[b]**
a. Research Center for Eco-hydrology, Chongqing Institute of Green and Intelligent
Technology, Chinese Academy of Sciences, Chongqing 400714, China
b. CSIR-National Institute of Oceanography, Regional Centre, Visakhapatnam, India
**Correspondence**
**Siyue Li**
*Chongqing Institute of Green and Intelligent Technology (CIGIT),*
*Chinese Academy of Sciences (CAS).*
*266, Fangzheng Avenue, Shuitu High-tech Park, Beibei, Chongqing 400714, China.*
*Tel: +86 23 65935058; Fax: +86 23 65935000*
*Email: syli2006@163.com*
**Abstract**
$CO_2$ outgassing from rivers is a critical component for evaluating riverine carbon
cycle, but it is poorly quantified largely due to limited measurements and modeling of
gas transfer velocity in subtropical streams and rivers. We measured $CO_2$ flux rates,
and calculated k and partial pressure ($pCO_2$) in 60 river networks of the Three Gorges
Reservoir (TGR) region, a typical area in the upper Yangtze River with monsoonal
climate and mountainous terrain. The determined $k_{600}$ (gas transfer velocity
normalized to a Schmidt number of 600 ($k_{600}$) at a temperature of 20 ℃) value
(48.4±53.2 cm/h) showed large variability due to spatial variations in physical
processes on surface water turbulence. Our flux-derived k values using chambers
were comparable with model derived from flow velocities based on a subset of data.
Unlike in open waters, e.g. lakes, $k_{600}$ is more pertinent to flow velocity and water
depth in the studied river systems. Our results show that TGR river networks emitted
approx. 0.69 to 0.71 Tg $CO_2$ (1 Tg=$10^{12}$ g) during monsoon period using varying
approaches such as chambers, derived $k_{600}$ values and model. This study suggests that
incorporating scale-appropriate k measurements into extensive $pCO_2$ investigations
are required to refine basin-wide carbon budgets in the subtropical streams and small
rivers. We concluded that simple parameterization of $k_{600}$ as a function of
morphological characteristics is site specific for regions / watersheds and hence
highly variable in rivers of the upper Yangtze. $K_{600}$ models should be developed for
stream studies to evaluate the contribution of these regions to the atmospheric $CO_2$.
**Key words**: $CO_2$ outgassing, riverine C flux, flow velocity, physical controls, Three
Gorge Reservoir, Yangtze River

## 1. Introduction

Rivers serve as a significant contributor of $CO_2$ to the atmosphere (Raymond et al., 2013;Cole et al., 2007;Li et al., 2012;Tranvik et al., 2009). As a consequence, accurate quantification of riverine $CO_2$ emissions is a key component to estimate net continental carbon (C) flux (Raymond et al., 2013). More detailed observational data and accurate measurement techniques are critical to refine the riverine C budgets (Li and Bush, 2015;Raymond and Cole, 2001). Generally two methods are used to estimate $CO_2$ areal fluxes from the river system, such as direct measurements using floating chambers (FCs), and indirect calculation of thin boundary layer (TBL) model, which is depended on gas concentration gradient at air-water interface and gas transfer velocity, k (Guerin et al., 2007;Xiao et al., 2014). Direct measurements are normally laborious, while the latter method is ease and simple and thus is preferred (Butman and Raymond, 2011;Lauerwald et al., 2015;Li et al., 2013;Li et al., 2012;Ran et al., 2015).

The areal flux of $CO_2$ (F, mmol/m$^2$/d) *via* the water–air interface by TBL is described as follows:

$$F = k \times K_h \times \triangle pCO_2 \qquad (1)$$

$$K_h = 10^{-(1.11 + 0.016 * T - 0.00007 * T^2)} \qquad (2)$$

where $k$ (m/d) is the gas transfer velocity of $CO_2$ (also referred to as piston velocity) at the *in situ* temperature (Li et al., 2016). $\triangle pCO_2$ (µatm) is the $pCO_2$ gradient at air-water interface (Borges et al., 2004). $K_h$ (mmol/m$^3$/µatm) is the aqueous-phase solubility coefficient of $CO_2$ corrected using *in situ* temperature ($T$ in ℃) (Li et al.,

2016).

$\triangle p\text{CO}_2$ can be measured well in various aquatic systems, however, the accuracy

of the estimation of flux is depended on the k value. Broad ranges of k for $CO_2$
(Raymond and Cole, 2001;Raymond et al., 2012;Borges et al., 2004) were reported
due to variations in techniques, tracers used and governing processes. k is controlled
by turbulence at the surface aqueous boundary layer, hence, $k_{600}$ (the standardized gas
transfer velocity at a temperature of 20 $^{0}$C is valid for freshwater) is parameterized as
a function of wind speed in open water systems of reservoirs, lakes, and oceans
(Borges et al., 2004;Guerin et al., 2007;Wanninkhof et al., 2009). While in streams
and small rivers, turbulence at the water-air interface is generated by shear stresses at
streambed, thus k is modeled using channel slope, water depth, and water velocity in
particular (Raymond et al., 2012;Alin et al., 2011). Variable formulations of k have
been established by numerous theoretical, laboratory and field studies, nonetheless,
better constraint on k levels is still required as its levels are very significant and
specific due to large heterogeneity in hydrodynamics and physical characteristics of
river networks. This highlights the importance of k measurements in a wide range of
environments for accurate upscaling of $CO_2$ evasion, and for parameterizing the
physical controls on $k_{600}$. However, only few studies provide information of k for
riverine $CO_2$ flux in Asia (Alin et al., 2011;Ran et al., 2015), and those studies do not
address the variability of k in China's small rivers and streams.

Limited studies demonstrated that higher levels of k in the Chinese large rivers

(Liu et al., 2017;Ran et al., 2017;Ran et al., 2015;Alin et al., 2011), which contributed
to much higher $CO_2$ areal flux particularly in China's monsoonal rivers that are
impacted by hydrological seasonality. The monsoonal flow pattern and thus flow
velocity is expected to be different than other rivers in the world, as a consequence, k
levels should be different than others, and potentially is higher in subtropical
monsoonal rivers.
Considerable efforts, such as purposeful (Crusius and Wanninkhof,
2003;Jean-Baptiste and Poisson, 2000) and natural tracers (Wanninkhof, 1992) and
FCs (Alin et al., 2011;Borges et al., 2004;Prytherch et al., 2017;Guerin et al., 2007),
have been carried out to estimate accurate k values. The direct determination of k by
FCs is more popular due to simplicity of the technique for short-term $CO_2$ flux
measurements (Prytherch et al., 2017;Raymond and Cole, 2001;Xiao et al., 2014).
Prior reports, however, have demonstrated that k values and the parameterization of k
as a function of wind and/or flow velocity (probably water depth) vary widely across
rivers and streams (Raymond and Cole, 2001;Raymond et al., 2012). To contribute to
this debate, extensive investigation was firstly accomplished for determination of k in
rivers and streams of the upper Yangtze using FC method. Models of k were further
developed using hydraulic properties (i.e., flow velocity, water depth) by flux
measurements with chambers and TBL model. Our recent study preliminarily
investigated $pCO_2$ and air – water $CO_2$ areal flux as well as their controls from fluvial
networks in the Three Gorges Reservoir (TGR) area (Li et al., 2018). The past study
was based on two field works, and the diffusive models from other rivers / regions
were used. Here, we derive k levels and develop the gas transfer model in this area
(mountainous streams and small rivers) for more accurate quantification of $CO_2$ areal
flux, and also to serve for the fluvial networks in the Yangtze River or others with
similar hydrology and geomorphology. Moreover, we did detail field campaigns in
the two contrasting rivers: Daning and Qijiang for models (Fig. 1), the rest were TGR
streams and small rivers (abbreviation in TGR rivers). The study thus clearly stated
distinct differences than the previous study (Li et al., 2018) by the new contributions
of specific objectives and data supplements, as well as wider significance. Our new
contributions to the literature thus include (1) determination and controls of k levels
for small rivers and streams in subtropical areas of China, and (2) new models
developed in the subtropical mountainous river networks. The outcome of this study
is expected to help in accurate estimation of $CO_2$ evasion from subtropical rivers and
streams, and thus refine riverine C budget over a regional/basin scale.

**2. Materials and methods**
**2.1. Study areas**
All field measurements were carried out in the rivers and streams of the Three
Gorges Reservoir (TGR) region (28°44′–31°40′N, 106°10′–111°10′E) that is locating
in the upper Yangtze River, China (Fig. 1). This region is subject to humid subtropical
monsoon climate with an average annual temperature ranging between15 and 19 ℃.
Average annual precipitation is approx. 1250 mm with large intra- and inter-annual
variability. About 75% of the annual total rainfall is concentrated between April and
September (Li et al., 2018).
The river sub-catchments include large scale river networks covering the
majority of the tributaries of the Yangtze in the TGR region, i.e., data of 48 tributaries
were collected. These tributaries have drainage areas that vary widely from 100 to
4400 km$^2$ with width ranging from 1 m to less than 100 m. The annual discharges
from these tributaries have a broad spectrum of 1.8 – 112 m$^3$/s. Detailed samplings
were conducted in the two largest rivers of Daning (35 sampling sites) and Qijiang
(32 sites) in the TGR region. These two river basins drain catchment areas of 4200
and 4400 km$^2$.The studied river systems had width < 100 m, we thus defined them as
small rivers and streams. The Daning and Qijiang river systems are underlain by
widely carbonate rock, and locating in a typical karst area. The location of sampling
sites is deciphered in Fig. 1. The detailed information on sampling sites and primary
data are presented in the Supplement Materials (Appendix Table A1). The sampling
sites are outside the Reservoirs and are not affected by dam operation.

**2.2. Water sampling and analyses**
Three fieldwork campaigns from the main river networks in the TGR region
were undertaken during May through August in 2016 (i.e., 18-22 May for Daning, 21
June-2 July for the entire tributaries of TGR, and 15-18 August for Qijiang). A total
of 115 discrete grab samples were collected (each sample consisted of three
replicates). Running waters were taken using pre acid-washed 5-L high density
polyethylene (HDPE) plastic containers from depths of 10 cm below surface. The
samples were filtered through pre-baked Whatman GF/F (0.7-μm pore size) filters on

the sampling day and immediately stored in acid-washed HDPE bottles. The bottles

were transported in ice box to the laboratory and stored at 4 ℃ for analysis.

Concentrations of dissolved organic carbon (DOC) were determined within 7 days of

water collection (Mao et al., 2017).

Water temperature (T), $p$H, DO saturation (DO%) and electrical conductivity

(EC) were measured *in situ* by the calibrated multi-parameter sondes (HQ40d HACH,

USA, and YSI 6600, YSI incorporated, USA). pH, the key parameter for $p$CO$_2$

calculation, was measured to a precision of $\pm 0.01$, and pH sonde was calibrated by

the certified reference materials (CRMs) before measurements with an accuracy of

better than $\pm 0.2\%$. Atmospheric CO$_2$ concentrations were determined *in situ* using

EGM-4 (Environmental Gas Monitor; PP SYSTEMS Corporation, USA). Total

alkalinity was measured using a fixed endpoint titration method with 0.0200 mol/L

hydrochloric acid (HCl) on the sampling day. DOC concentration was measured using

a total organic carbon analyzer (TOC-5000, Shimadzu, Japan) with a precision better

than 3% (Mao et al., 2017). All the used solvents and reagents in experiments were of

analytical-reagent grade.

Concomitant stream width, depth and flow velocity were determined along the

cross section, and flow velocity was determined using a portable flow meter LS300-A

(China), the meter shows an error of <1.5%. Wind speed at 1 m over the water surface

($U_1$) and air temperature (Ta) were measured with a Testo 410-1 handheld

anemometer (Germany). Wind speed at 10 m height ($U_{10}$, unit in m/s) was calculated

using the following formula (Crusius and Wanninkhof, 2003):

$$U_{10} = U_Z \left[ 1 + \frac{(C_{d10})^{1/2}}{K} \times \ln\left(\frac{10}{z}\right) \right] \quad (3)$$
where $C_{d10}$ is the drag coefficient at 10 m height (0.0013 m/s), and K is the von
Karman constant (0.41), and z is the height (m) of wind speed measurement.
$U_{10}=1.208 \times U_1$ as we measured the wind speed at a height of 1m ($U_1$).
Aqueous $pCO_2$ was computed from the measurements of pH, total alkalinity, and
water temperature using $CO_2$ System ($k_1$ and $k_2$ are from Millero, 1979) (Lewis et al.,
1998). This program can yield high quality data (Li et al., 2013;Li et al., 2012;Borges
et al., 2004).

**2.3. Water-to-air $CO_2$ fluxes using FC method**
FCs (30 cm in diameter, 30 cm in height) were deployed to measure air-water
$CO_2$ fluxes and transfer velocities. They were made of cylindrical polyvinyl chloride
(PVC) pipe with a volume of 21.20 L and a surface area of 0.071 m$^2$. These
non-transparent, thermally insulated vertical tubes, covered by aluminum foil, were
connected *via* $CO_2$ impermeable rubber-polymer tubing (with outer and inner
diameters of 0.5 cm and 0.35 cm, respectively) to a portable non-dispersive infrared
$CO_2$ analyzer EGM-4 (PPSystems). Air was circulated through the EGM-4 instrument
*via* an air filter using an integral pump at a flow rate of 350 ml/min. The chamber
method was widely used and more details of advantages and limits on chambers were
reviewed elsewhere (Alin et al., 2011;Borges et al., 2004;Xiao et al., 2014).
Chamber measurements were conducted by deploying two replicate chambers or
one chamber for two times at each site. In sampling sites with low and favorable flow
conditions (Fig. S1), freely drifting chambers (DCs) were executed, while sampling
sites in rivers and streams with higher flow velocity were conducted with anchored
chambers (ACs) (Ran et al., 2017). DCs were used in sampling sites with current
velocity of < 0.1 m/s, this resulted in limited sites (a total of 6 sites) using DCs. ACs
would create overestimation of $CO_2$ emissions by a factor of several folds (i.e., > 2) in
our study region (Lorke et al., 2015). Data were logged automatically and
continuously at 1-min interval over a given span of time (normally 5-10 minutes) after
enclosure. The $CO_2$ area flux (mg/m$^2$/h) was calculated using the following formula.
$$F = 60 \times \frac{dpco2 \times M \times P \times T_0}{dt \times V_0 \times P_0 \times T} H \qquad (4)$$

Where d$p$co$_2$/dt is the rate of concentration change in FCs (μl/l/min); M is the

molar mass of $CO_2$ (g/mol); P is the atmosphere pressure of the sampling site (Pa); T
is the chamber absolute temperature of the sampling time (K); $V_0$ is the molar volume
(22.4 l/mol), $P_0$ is atmosphere pressure (101325 Pa), and $T_0$ is absolute temperature
(273.15 K) under the standard condition; H is the chamber height above the water
surface (m) (Alin et al., 2011). We accepted the flux data that had a good linear
regression of flux against time ($R^2 \geq 0.95$, p<0.01) following manufacturer'
specification. In our sampling points, all measured fluxes were retained since the
floating chambers yielded linearly increasing $CO_2$ against time.

Water samples from a total of 115 sites were collected. Floating chambers with

replicates were deployed in 101 sites (32 sampling sites in Daning, 37 sites in TGR
river networks and 32 sites in Qijiang). The sampling period covered spring and
summer season, our sampling points are reasonable considering a water area of 433
km$^2$. For example, 16 sites were collected for Yangtze system to examine
hydrological and geomorphological controls on $p\mathrm{CO_2}$ (Liu et al., 2017), and 17 sites
for dynamic biogeochemical controls on riverine $p\mathrm{CO_2}$ in the Yangtze basin (Liu et al.,
2016). Similar to other studies, sampling and flux measurements in the day would
tend to underestimate $\mathrm{CO_2}$ evasion rate (Bodmer et al., 2016).

## 2.4. Calculations of the gas transfer velocity

The k was calculated by reorganizing Eq (1). To make comparisons, k is
normalized to a Schmidt (Sc) number of 600 ($k_{600}$) at a temperature of 20 ℃.
$$k_{600} = k_T(\frac{600}{S_{CT}})^{-0.5} \qquad (5)$$
$$S_{CT} = 1911.1 - 118.11T + 3.4527T^2 - 0.04132T^3 \qquad (5)$$
Where $k_T$ is the measured values at the *in situ* temperature (T, unit in ℃), $S_{CT}$ is the
Schmidt number of temperature T. Dependency of -0.5 was employed here as
measurement were made in turbulent rivers and streams in this study (Alin et al.,
2011;Borges et al., 2004;Wanninkhof, 1992).

## 2.5. Estimation of river water area

Water surface is an important parameter for $\mathrm{CO_2}$ efflux estimation, while it
depends on its climate, channel geometry and topography. River water area therefore
largely fluctuates with much higher areal extent of water surface particularly in
monsoonal season. However, most studies do not consider this change, and a fraction
of the drainage area is used in river water area calculation (Zhang et al., 2017). In our
study, a 90 m resolution SRTM DEM (Shuttle Radar Topography Mission digital
elevation model) data and Landsat images in dry season were used to delineate river
network, and thus water area (Zhang et al., 2018), whilst, stream orders were not
extracted. Water area of river systems is generally much higher in monsoonal season
in comparison to dry season, for instance, Yellow River showed 1.4-fold higher water
area in the wet season than in the dry season (Ran et al., 2015). Available dry-season
image was likely to underestimate $CO_2$ estimation.

**2.6. Data processing**
Prior to statistical analysis, we excluded $k_{600}$ data for samples with the air-water
$pCO_2$ gradient <110 µatm, since the error in the $k_{600}$ calculations drastically enhances
when $\triangle pCO_2$ approaches zero (Borges et al., 2004;Alin et al., 2011), and datasets with
$\triangle pCO_2$ >110 µatm provide an error of <10% on $k_{600}$ computation. Thus, we discarded
the samples (36.7% of sampling points with flux measurements) with $\triangle pCO_2$ <110
µatm for $k_{600}$ model development, while all samples were included for the flux
estimations from diffusive TBL model and floating chambers.
Spatial differences (Daning, Qijiang and entire tributaries of TGR region) were
tested using the nonparametric Mann Whitney U-test. Multivariate statistics, such as
correlation and stepwise multiple linear regression, were performed for the models of
$k_{600}$ using potential physical parameters of wind speed, water depth, and current
velocity as the independent variables (Alin et al., 2011). Data analyses were
conducted from both separated data and combined data of river systems. k models
were obtained by water depth using data from the TGR rivers, while by flow velocity
in the Qijiang, whilst, models were not developed for Daning and combined data. All
statistical relationships were significant at p < 0.05. The statistical processes were
conducted using SigmaPlot 11.0 and SPSS 16.0 for Windows (Li et al., 2009;Li et al.,

2016).


**3. Results**
**3.1. CO$_2$ partial pressure and key water quality variables**

Significant spatial variations in water temperature, pH, $p$CO$_2$ and DOC were

observed among Daning, TGR and Qijiang rivers whereas alkalinity did not display
such variability (Fig. S2). $p$H varied from 7.47 to 8.76 with exceptions of two quite
high values of 9.38 and 8.87 (total mean: 8.39 $\pm$0.29). Significantly lower $p$H was
observed in TGR rivers (8.21 $\pm$0.33) (Table 1; p<0.001; Fig. S2). $p$CO$_2$ varied
between 50 and 4830 $\mu$atm with mean of 846 $\pm$819 $\mu$atm (Table 1). There were 28.7%
of samples that had $p$CO$_2$ levels lower than 410 $\mu$atm, while the studied rivers were
overall supersaturated with reference to the atmospheric CO$_2$ and act as a source for
the atmospheric CO$_2$. The $p$CO$_2$ levels were 2.1 to 2.6-fold higher in TGR rivers than
Daning (483 $\pm$294 $\mu$atm) and Qijiang Rivers (614 $\pm$316 $\mu$atm) (Fig. S2).

There was significantly higher concentration of DOC in the TGR rivers (12.83 $\pm$

7.16 mg/l) than Daning and Qijiang Rivers (3.76 $\pm$5.79 vs 1.07 $\pm$0.33 mg/l in Qijiang
and Daning) (p<0.001; Fig. S3). Moreover, Qijiang showed significantly higher
concentration of DOC than Daning (3.76 $\pm$ 5.79 vs 1.07 $\pm$ 0.33 mg/l in Qijiang and
Daning) (p<0.001 by Mann-Whitney Rank Sum Test; Fig. S3).
**3.2. $CO_2$ flux using floating chambers**
The calculated $CO_2$ areal fluxes were higher in TGR rivers (217.7 $\pm$ 334.7
mmol/m$^2$/d, n = 35), followed by Daning (122.0 $\pm$ 239.4 mmol/m$^2$/d, n = 28) and
Qijiang rivers (50.3 $\pm$ 177.2 mmol/m$^2$/d, n = 32) (Fig. 2). The higher $CO_2$ evasion
from the TGR rivers is consistent with high riverine $pCO_2$ levels. The mean $CO_2$
emission rate was 133.1 $\pm$ 269.1 mmol/m$^2$/d (n = 95) in all three rivers sampled. The
mean $CO_2$ flux differed significantly between TGR rivers and Qijiang (Fig. 2).

**3.3. k levels**
A total of 64 data were used (10 for Daning River, 33 for TGR rivers and 21 for
Qijiang River) to develop k model after removal of samples with $\triangle pCO_2$ less than 110
µatm (Table 2). No significant variability in $k_{600}$ values were observed among the
three rivers sampled (Fig. 3). The mean $k_{600}$ was relatively higher in Qijiang (60.2 $\pm$
78.9 cm/h), followed by Daning (50.2 $\pm$ 20.1 cm/h) and TGR rivers (40.4 $\pm$ 37.6
cm/h), while the median $k_{600}$ was higher in Daning (50.5cm/h), followed by TGR
rivers (30.0 cm/h) and Qijiang (25.8 cm/h) (Fig. 3; Table S1). Combined $k_{600}$ data
were averaged to 48.4 $\pm$ 53.2 cm/h (95% CI: 35.1-61.7), and it is 1.5-fold higher than
the median value (32.2 cm/h) (Fig. 3).
Contrary to our expectations, no significant relationship was observed between
$k_{600}$ and water depth, and current velocity using the entire data in the three river
systems (TGR streams and small rivers, Danning and Qjiang) (Fig. S4). There were
not statistically significant relationships between $k_{600}$ and wind speed using separated
data or combined data. Flow velocity showed slightly linear relation with $k_{600}$, and the
extremely high value of $k_{600}$ was observed during the periods of higher flow velocity
(Fig. S4a) using combined data. Similar trend was also observed between water depth
and $k_{600}$ values (Fig. S4b). $k_{600}$ as a function of water depth was obtained in the TGR
rivers, but it explained only 30% of the variance in $k_{600}$. However, model using data
from Qijiang could explain 68% of the variance in $k_{600}$ (Fig. 4b), and it was in line
with general theory.

**4. Discussion**
**4.1. Uncertainty assessment of $p$CO$_2$ and flux-derived $k_{600}$ values**
The uncertainty of flux-derived k values mainly stem from $\Delta p$CO$_2$ and flux
measurements (Bodmer et al., 2016;Golub et al., 2017;Lorke et al., 2015). Thus we
provided uncertainty assessments caused by dominant sources of uncertainty from
measurements of aquatic $p$CO$_2$ and CO$_2$ areal flux since uncertainty of atmospheric
CO$_2$ measurement could be neglected.
In our study, aquatic $p$CO$_2$ was computed based on pH, alkalinity and water
temperature rather than directly measured. Recent studies highlighted $p$CO$_2$
uncertainty caused by systematic errors over empiric random errors (Golub et al.,
2017). Systematic errors are mainly attributed to instrument limitations, i.e., sondes of
pH and water temperature. The relative accuracy of temperature meters was $\pm 0.1\ ^{0}$C
according to manufacturers' specifications, thus the uncertainty of water T propagated
on uncertainty in $pCO_2$ was minor (Golub et al., 2017). Systematic errors therefore
stem from pH, which has been proved to be a key parameter for biased $pCO_2$
estimation calculated from aquatic carbon system (Li et al., 2013;Abril et al., 2015).
We used a high accuracy of pH electrode and the pH meters were carefully calibrated
using CRMs, and *in situ* measurements showed an uncertainty of ±0.01. We then run
an uncertainty of ±0.01 pH to quantify the $pCO_2$ uncertainty, and an uncertainty of ±3%
was observed. Systematic errors thus seemed to show little effects on $pCO_2$ errors in
our study.

Random errors are from repeatability of carbonate measurements. Two replicates

for each sample showed the uncertainty of within ±5%, indicating that uncertainty in
$pCO_2$ calculation from alkalinity measurements could be minor.

The measured pH ranges also exhibited great effects on $pCO_2$ uncertainty (Hunt

et al., 2011;Abril et al., 2015). At low pH, $pCO_2$ can be overestimated when
calculated from pH and alkalinity (Abril et al., 2015). Samples for $CO_2$ fluxes
estimated from pH and alkalinity showed pH average of 8.39±0.29 (median 8.46 with
quartiles of 8.24-8.56) (n=115). Thus, overestimation of calculated $CO_2$ areal flux
from pH and alkalinity is likely to be minor. Further, contribution of organic matter to
non-carbonate alkalinity is likely to be neglected because of low DOC (mean 6.67
mg/L; median 2.51 mg/L) (Hunt et al., 2011;Li et al., 2013).

Efforts have been devoted to measurement techniques (comparison of FC, eddy

covariance-EC and boundary layer model-BLM) for improving $CO_2$ quantification
from rivers because of a notable contribution of inland waters to the global C budget,
which could have a large effect on the magnitude of the terrestrial C sink. Whilst,
prior studies reported inconsistent trends of $CO_2$ area flux by these methods. For
instance, $CO_2$ areal flux from FC was much lower than EC (Podgrajsek et al., 2014),
while areal flux from FC was higher than both EC and BLM elsewhere (Erkkila et al.,
2018), however, Schilder et al (Schilder et al., 2013) demonstrated that areal flux from
BLM was 33-320% of *in-situ* FC measurements. Albeit unsatisfied errors of varied
techniques and additional perturbations from FC occurs, however, FC method is
currently a simple and preferred technique for $CO_2$ flux because that choosing a right
k value remains a major challenge and others require high workloads (Martinsen et al.,

2018).

Recent study further reported fundamental differences in $CO_2$ emission rates

between ACs and freely DFs (Lorke et al., 2015), i.e., ACs biased the gas areal flux
higher by a factor of 2.0-5.5. However, some studies observed that ACs showed
reasonable agreement with other flux measurement techniques (Galfalk et al., 2013),
and this method is straightforward, inexpensive and relatively simple hence it is
widely used (Ran et al., 2017). Water-air interface $CO_2$ flux measurements were
primarily made using ACs in our studied streams and small rivers because of
relatively high current velocity; otherwise, floating chambers will travel far during the
measurement period. In addition, inflatable rings were used for sealing the chamber
headspace and submergence of ACs was minimal, therefore, our measurements were
potentially overestimated, but reasonable. We could not test the overestimation of ACs
in this study, the modified FCs, i.e., DCs and integration of ACs and DCs, and
multi-method comparison study including FCs, ECs and BLM should be conducted
for a reliable chamber method.

Our model was from a subset of the data (i.e., Qijiang), while $CO_2$ flux from our

model was in good agreement with the fluxes from FC, determined k and other
models when the developed model was applied for the whole dataset (please refer to
Tables 2 and 3). The comparison of the fluxes from variable methods suggested that
the model can be used for riverine $CO_2$ flux at catchment scale or regional scale
though it cannot be used at individual site. Recent studies, however, did not test the
applicability of models when $k_{600}$ models from other regions were employed. Our $k_{600}$
values were close to the average of Ran et al. (2015) (measured with drifting
chambers) and Liu et al. (2017) (measured with static chambers in canoe shape), this
indicated that our potential overestimation was limited. However, since we had very
limited drifting chamber measurements because of high current velocity, the
relationships with chamber derived $k_{600}$ values and flow velocity/depth only with the
drifting chamber data could not be tested. Whereas, we acknowledged that $k_{600}$ could
be over-estimated using AFs.

The extremely high values (two values of 260 and 274 cm/h) are outside of the

global ranges and also considerably higher than $k_{600}$ values in Asian rivers.
Furthermore, the revised model was comparable to the published models (Fig. 4), i.e.,
models of Ran et al. (2015) (measured with drifting chambers) and Liu et al. (2017)
(measured with static chambers in canoe shape), which suggested that exclusion of
the two extremely values were reasonable, and this was further supported by the $CO_2$
flux using different approaches (Tables 2 and 3).

Sampling seasonality considerably regulated riverine $pCO_2$ and gas transfer

velocity and thus water-air interface $CO_2$ evasion rate (Ran et al., 2015;Li et al., 2012).
We sampled waters in wet season (monsoonal period) due to that it showed wider
range of flow velocity and thus it covered the $k_{600}$ levels in the whole hydrological
season. Wet season generally had higher current velocity and thus higher gas transfer
velocity (Ran et al., 2015), while aquatic $pCO_2$ was variable with seasonality. We
recently reported that riverine $pCO_2$ in the wet season was 81% the level in the dry
season (Li et al., 2018), and prior study on the Yellow River reported that k level in
the wet season was 1.8-fold higher than in the dry season (Ran et al., 2015), while
another study on the Wuding River demonstrated that k level in the wet season was
83%-130% of that in the dry season (Ran et al., 2017). Thus, we acknowledged a
certain amount of errors on the annual flux estimation from sampling campaigns
during the wet season in the TGR area, while this uncertainty could not be significant
because that the diluted $pCO_2$ could alleviate the overestimated emission by increased
k level in the wet season (stronger discussion please refer to SOM).

**4.2. Determined k values relative to world rivers**

We derived first-time the k values in the subtropical streams and small rivers.

Our determined $k_{600}$ levels with a 95% CI of 35.1 to 61.7 (mean: 48.4) cm/h is
compared well with a compilation of data for streams and small rivers (e.g., 3-70
cm/h) (Raymond et al., 2012). Our determined $k_{600}$ values are greater than the global
rivers' average (8 - 33 cm/h) (Raymond et al., 2013;Butman and Raymond, 2011), and
much higher than mean for tropical and temperate large rivers (5-31 cm/h) (Alin et al.,
2011). These studies evidences that $k_{600}$ values are highly variable in streams and
small rivers (Alin et al., 2011;Ran et al., 2015). Though the mean $k_{600}$ in the TGR,
Daning and Qijiang is higher than global mean, however, it is consistent with $k_{600}$
values in the main stream and river networks of the turbulent Yellow River (42 $\pm$17
cm/h) (Ran et al., 2015), and Yangtze (38 $\pm$40 cm/h) (Liu et al., 2017) (Table S2).

The calculated $p$CO$_2$ levels were within the published range, but towards the

lower-end of published concentrations compiled elsewhere (Cole and Caraco, 2001;Li
et al., 2013). The total mean $p$CO$_2$ (846 $\pm$819 μatm) in the TGR, Daning and Qijiang
rivers was one third lower than global river's average (3220 μatm) (Cole and Caraco,
2001). The lower $p$CO$_2$ than most of the world's river systems, particularly the
under-saturated values, demonstrated that heterotrophic respiration of terrestrially
derived DOC was not significant. Compared with high alkalinity, the limited delivery
of DOC particularly in the Daning and Qijiang river systems (Figs. S2 and S3) also
indicated that in-stream respiration was limited. These two river systems are
characterized by karst terrain and underlain by carbonate rock, where photosynthetic
uptake of dissolved $CO_2$ and carbonate minerals dissolution considerably regulated
aquatic $p$CO$_2$ (Zhang et al., 2017).

Higher pH levels were observed in Daning and Qijiang river systems (p<0.05 by

Mann-Whitney Rank Sum Test), where more carbonate rock exists that are
characterized by karst terrain. Our pH range was comparable to the recent study on
the karst river in China (Zhang et al., 2017). Quite high values (8.39 $\pm$0.29, ranging
between 7.47 and 9.38; 95% confidence interval: 8.33-8.44) could increase the
importance of the chemical enhancement, nonetheless, few studies did take chemical
enhancement into account (Wanninkhof and Knox, 1996;Alshboul and Lorke, 2015).
The chemical enhancement can increase the $CO_2$ areal flux by a factor of several folds
in lentic systems with low gas transfer velocity, whist enhancement factor decreased
quickly as $k_{600}$ increased (Alshboul and Lorke, 2015). Our studied rivers are located
in mountainous area with high $k_{600}$, which could cause minor chemical enhancement
factor. This chemical enhancement of $CO_2$ flux was also reported to be limited in
high-pH and also turbulent rivers (Zhang et al., 2017).

**4.3. Hydraulic controls of $k_{600}$**

It has been well established that $k_{600}$ is governed by a multitude of physical

factors particularly current velocity, wind speed, stream slope and water depth, of
which, wind speed is the dominant factor of k in open waters such as large rivers and
estuaries (Alin et al., 2011;Borges et al., 2004;Crusius and Wanninkhof,
2003;Raymond and Cole, 2001). In contrast $k_{600}$ in small rivers and streams is closely
linked to flow velocity, water depth and channel slope (Alin et al., 2011;Raymond et
al., 2012). Several studies reported that the combined contribution of flow velocity
and wind speed to k is significant in the large rivers (Beaulieu et al., 2012;Ran et al.,
2015). Thus, $k_{600}$ values are higher in the Yellow River (ca. 0-120 cm/h) as compared
to the low-gradient River Mekong (0-60 cm/h) (Alin et al., 2011;Ran et al., 2015), due
to higher flow velocity in the Yellow River (1.8 m/s) than Mekong river (0.9 ±0.4 m/s),
resulting in greater surface turbulence and higher $k_{600}$ level in the Yellow (42 ± 17
cm/h) than Mekong river (15 ± 9 cm/h). This could substantiate the higher $k_{600}$ levels
and spatial changes in $k_{600}$ values of our three river systems. For instance, similar to
other turbulent rivers in China (Ran et al., 2017;Ran et al., 2015), high $k_{600}$ values in
the TGR, Daning and Qjiang rivers were due to mountainous terrain catchment, high
current velocity (10 – 150 cm/s) (Fig. 4b), bottom roughness, and shallow water depth
(10 - 150 cm) (Fig. 4a). It has been suggested that shallow water enhances bottom
shear, and the resultant turbulence increases k values (Alin et al., 2011;Raymond et al.,
2012). These physical controls are highly variable across environmental types (Figs.
4a and 4b), hence, k values are expected to vary widely (Fig. 3). The $k_{600}$ values in the
TGR rivers showed wider range (1-177 cm/h; Fig. 3; Table S1), spanning more than 2
orders of magnitude across the region, and it is consistent with the considerable
variability in the physical processes on water turbulence across environmental settings.
Similar broad range of $k_{600}$ levels was also observed in the China's Yellow basin (ca.
0-123 cm/h) (Ran et al., 2015;Ran et al., 2017).

Insignificant relationships between riverine $k_{600}$ and wind speed were consistent

with earlier studies (Alin et al., 2011;Raymond et al., 2012). The lack of strong
correlation between $k_{600}$ and physical factors using the combined data were probably
due to combined effect of both flow velocity and water depth, as well as large
diversity of channel morphology, both across and within river networks in the entire
catchment (60, 000 km$^2$). This is further collaborated by weak correlations between
$k_{600}$ and flow velocity in the TGR rivers (Fig. 4), where one or two samples were
taken for a large scale examination. We provided new insights into $k_{600}$ parameterized
using current velocity. Nonetheless, $k_{600}$ from our flow velocity based model (Fig. 4b)
was potentially largely overestimated with consideration of other measurements (Alin
et al., 2015;Ran et al., 2015;Ran et al., 2017). When several extreme values were
removed, $k_{600}$ (cm/h) was parameterized as follows ($k_{600} = 62.879FV + 6.8357$, R $^2$=
0.52, p=0.019, FV-flow velocity with a unit of m/s), and this revised model was in
good agreement with the model in the river networks of the Yellow River (Ran et al.,
2017), but much lower than the model developed in the Yangtze system (Liu et al.,
2017) (Fig. 4c). This was reasonable because of $k_{600}$ values in the Yangtze system
were from large rivers with higher turbulence than Yellow and our studied rivers.
Furthermore, the determined $k_{600}$ using FCs was, on average, consistent with the
revised model (Table 2). These differences in relationship between spatial changes in
$k_{600}$ values and physical characteristics further corroborated heterogeneity of channel
geomorphology and hydraulic conditions across the investigated rivers.

The subtropical streams and small rivers are biologically more active and are

recognized to exert higher $CO_2$ areal flux to the atmosphere, however, their
contribution to riverine carbon cycling is still poorly quantified because of data
paucity and the absence of k in particular. Larger uncertainty of riverine $CO_2$ emission
in China was anticipated by use of $k_{600}$ from other continents or climate zones. For
instance, $k_{600}$ for $CO_2$ emission from tributaries in the Yellow River and karst rivers
was originated from the model in the Mekong (Zhang et al., 2017), and Pearl (Yao et
al., 2007), Longchuan (Li et al., 2012), and Metropolitan rivers (Wang et al., 2017),
which are mostly from temperate regions. Our $k_{600}$ values will therefore largely
improve the estimation of $CO_2$ evasion from subtropical streams and small rivers, and
improve to refine riverine carbon budget. More studies, however, are clearly needed
to build the model, based on flow velocity and slope/water depth given the difficulty
in k quantification on a large scale.

**4.4. Implications for large scale estimation**

We compared $CO_2$ areal flux by FCs and models developed here (Fig. 4) and

other studies (Alin et al., 2011) (Tables 2 and 3). $CO_2$ evasion was estimated for rivers
in China with k values ranged between 8 and 15 cm/h (Li et al., 2012;Yao et al.,
2007;Wang et al., 2011) (Table S2). These estimates of $CO_2$ evasion rate were
considerably lower than using present $k_{600}$ values (48.4 $\pm$53.2 cm/h). For instance,
$CO_2$ emission rates in the Longchuan River (e.g., k = 8 cm/h) and Pearl River
tributaries (e.g., k = 8-15 cm/h) were 3 to 6 times higher using present k values
compared to earlier estimates. We found that the determined $k_{600}$ average was
marginally beyond the levels from water depth based model and the model developed
by Alin et al (Alin et al., 2011), while equivalent to the flow velocity based revised
model, resulting in similar patterns of $CO_2$ emission rates (Table 2). Hence selection
of k values would significantly hamper the accuracy of the flux estimation. Therefore
k must be estimated along with $p$CO$_2$ measurements to accurate flux estimations.
We used our measured $CO_2$ emission rates by FCs for upscaling flux estimates
during monsoonal period given the sampling in this period and it was found to be 0.70
Tg $CO_2$ (1 Tg=$10^{12}$ g) for all rivers sampled in our study (Table 3a). The estimated
emission in the monsoonal period was close to that of the revised model (0.71 $\pm$0.66
(95% confidence interval: 0.46 - 0.94) Tg $CO_2$), and using the determined k average,
i.e., 0.69 $\pm$0.65 (95% confidence interval: 0.45-0.93) Tg $CO_2$, but slightly higher than
the estimation using water-depth based model (0.54 $\pm$0.51 Tg $CO_2$) and Alin's model
(0.53 $\pm$0.50 Tg $CO_2$) (Table 3b). This comparable $CO_2$ flux further substantiated the
exclusion of extremely $k_{600}$ values for developing model (Fig. 4). The $CO_2$ evasion
comparison by variable approaches also implied that the original flow velocity based
model (two extremely $k_{600}$ values were included; Fig. 4b) largely over-estimated the
$CO_2$ fluxes, i.e., 1.66 $\pm$1.55 (1.08-2.23) Tg $CO_2$, was 2.3-3 fold higher than other
estimations (Table 3b), and our earlier evasion using TBL on the TGR river networks
(Li et al., 2018). Moreover, our estimated $CO_2$ emission during monsoonal period also
suggests that $CO_2$ annual emissions from rivers and streams in this area were
previously underestimated, i.e., 0.03 Tg $CO_2$/y (Li et al., 2017) and 0.37-0.44 Tg
$CO_2$/y (Yang et al., 2013) as the former used TBL model with a lower k level, and the
latter employed floating chambers, but they both sampled very limited tributaries (i.e.,
2-3 rivers). Therefore, measurements of k must be made mandatory along with $p$CO$_2$
measurement in the river and stream studies.

**5. Conclusion**

We provided first determination of gas transfer velocity (k) in the subtropical streams and small rivers in the upper Yangtze. High variability in k values (mean 48.4 ± 53.2 cm/h) was observed, reflecting the variability of morphological characteristics on water turbulence both within and across river networks. We highlighted that k estimate from empirical model should be pursued with caution and the significance of incorporating k measurements along with extensive $pCO_2$ investigation is highly essential for upscaling to watershed/regional scale carbon (C) budget.

Riverine $pCO_2$ and $CO_2$ areal flux showed pronounced spatial variability with much higher levels in the TGR rivers. The $CO_2$ areal flux was averaged at 133.1 ± 269.1 mmol/m$^2$/d using FCs, the resulting emission during monsoonal period was around 0.7 Tg $CO_2$, similar to the scaling up emission with the determined k, and the revised flow velocity based model, while marginally above the water depth based model. More work is clearly needed to refine the k modeling in the river systems of the upper Yangtze River for evaluating regional C budgets.

**Acknowledgements**

This study was funded by "the Hundred-Talent Program" of the Chinese Academy of Sciences (R53A362Z10; granted to Dr. Li), and the National Natural Science Foundation of China (Grant No. 31670473). We are grateful to Mrs. Maofei Ni and Tianyang Li, and Miss Jing Zhang for their assistance in the field works. Users can access the original data from an Appendix. Special thanks are given to editor David Butman and anonymous reviewers for improving the manuscript.

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

**Table 1**. Statistics of all the data from three river systems (separated statistics please
refer to Figs. S2 and S3 in the Supplementary material).

|  |  | Water T ($^0$C) | pH | Alkalinity ($\mu$eq/l) | $p$CO$_2$ ($\mu$atm) | DO% | DOC (mg/L) |
|---|---|---|---|---|---|---|---|
| Number |  | 115 | 115 | 115 | 115 | 56 | 114 |
| Mean |  | 22.5 | 8.39 | 2589.1 | 846.4 | 91.5 | 6.67 |
| Median |  | 22.8 | 8.46 | 2560 | 588.4 | 88.8 | 2.51 |
| Std. Deviation |  | 6.3 | 0.29 | 640.7 | 818.5 | 8.7 | 7.62 |
| Minimum |  | 11.7 | 7.47 | 600 | 50.1 | 79.9 | 0.33 |
| Maximum |  | 34 | 9.38 | 4488 | 4830.4 | 115.9 | 37.48 |
| Percentiles | 25 | 16.3 | 8.24 | 2240 | 389.8 | 84.0 | 1.33 |
|  | 75 | 29 | 8.56 | 2920 | 920.4 | 99.1 | 9.96 |
| 95% CI for Mean | Lower Bound | 21.4 | 8.33 | 2470.8 | 695.2 | 89.1 | 5.26 |
|  | Upper Bound | 23.7 | 8.44 | 2707.5 | 997.6 | 93.8 | 8.09 |


CI-Confidence Interval.

**Table 2.** Comparison of different model for $CO_2$ areal flux estimation using combined data (unit is mmol/m$^2$/d for $CO_2$ areal flux and cm/h for $k_{600}$).

| | | From FC | Flow velocity-based model (Fig. 4b)[a] | Water depth-based model (Fig.3a) | Alin's model |
|---|---|---|---|---|---|
| $k_{600}$ | | 48.4[b] | 116.5[c] | 38.3 | 37.6 |
| $CO_2$ areal flux | | | | | |
| Mean | | 198.1 | 476.7 | 156.6 | 154.0 |
| S.D. | | 185.5 | 446.2 | 146.6 | 144.2 |
| 95% CI for Mean[a] | Lower Bound | 129.5 | 311.5 | 102.3 | 100.6 |
| | Upper Bound | 266.8 | 641.8 | 210.8 | 207.4 |

CI-Confidence Interval

[a]Flow velocity –based model is from a subset of the data (please refer to Fig. 4)

[b]Mean value determined using floating chambers (FC).

c-This figure is revised to be 49.6 cm/h if the model ($k_{600} = 62.879FV + 6.8357$, R$^2$= 0.52, p=0.019) is used (the model is obtained by taking out two extremely values; please refer to Fig. 4c), and the corresponding $CO_2$ areal flux is 203 ±190 mmol/m$^2$/d.

**Table 3.** $CO_2$ emission during monsoonal period (May through Oct.) from total rivers
sampled in the study.
(a) Upscaling using $CO_2$ areal flux (mean $\pm$S.D.) by FC during monsoonal period.

| | Catchment Area km$^2$ | Water surface km$^2$ | $CO_2$ areal flux mmol/m$^2$/d | $CO_2$ emission Tg $CO_2$ |
|---|---|---|---|---|
| Daning | 4200 | 21.42 | 122.0 $\pm$239.4 | 0.021 |
| Qijiang | 4400 | 30.8 | 50.3 $\pm$177.2 | 0.0125 |
| TGR river | 50000 | 377.78 | 217.7 $\pm$334.7 | 0.666 |
| Total | | | | 0.70 |


(b) Upscaling using determined $k_{600}$ average and models (whole dataset are used
here).

| | | From determined $k_{600}$ mean | Flow velocity-based model (Fig. 4b) (numbers in bracket is from the revised model; Fig. 4c) | Water depth-based model (Fig. 4a) | Alin's model |
|---|---|---|---|---|---|
| Mean | | 0.69 | 1.66 (0.71) | 0.54 | 0.53 |
| S.D. | | 0.65 | 1.55 (0.66) | 0.51 | 0.50 |
| 95% CI for Mean | Lower Bound | 0.45 | 1.08 (0.46) | 0.36 | 0.35 |
| | Upper Bound | 0.93 | 2.23 (0.94) | 0.74 | 0.72 |

A total water area of approx. 430 km$^2$ for all tributaries (water area is from Landsat
ETM+ in 2015); $CO_2$ emission upscaling (Tg $CO_2$ during May through October) was
conducted during the monsoonal period because of the sampling in this period.

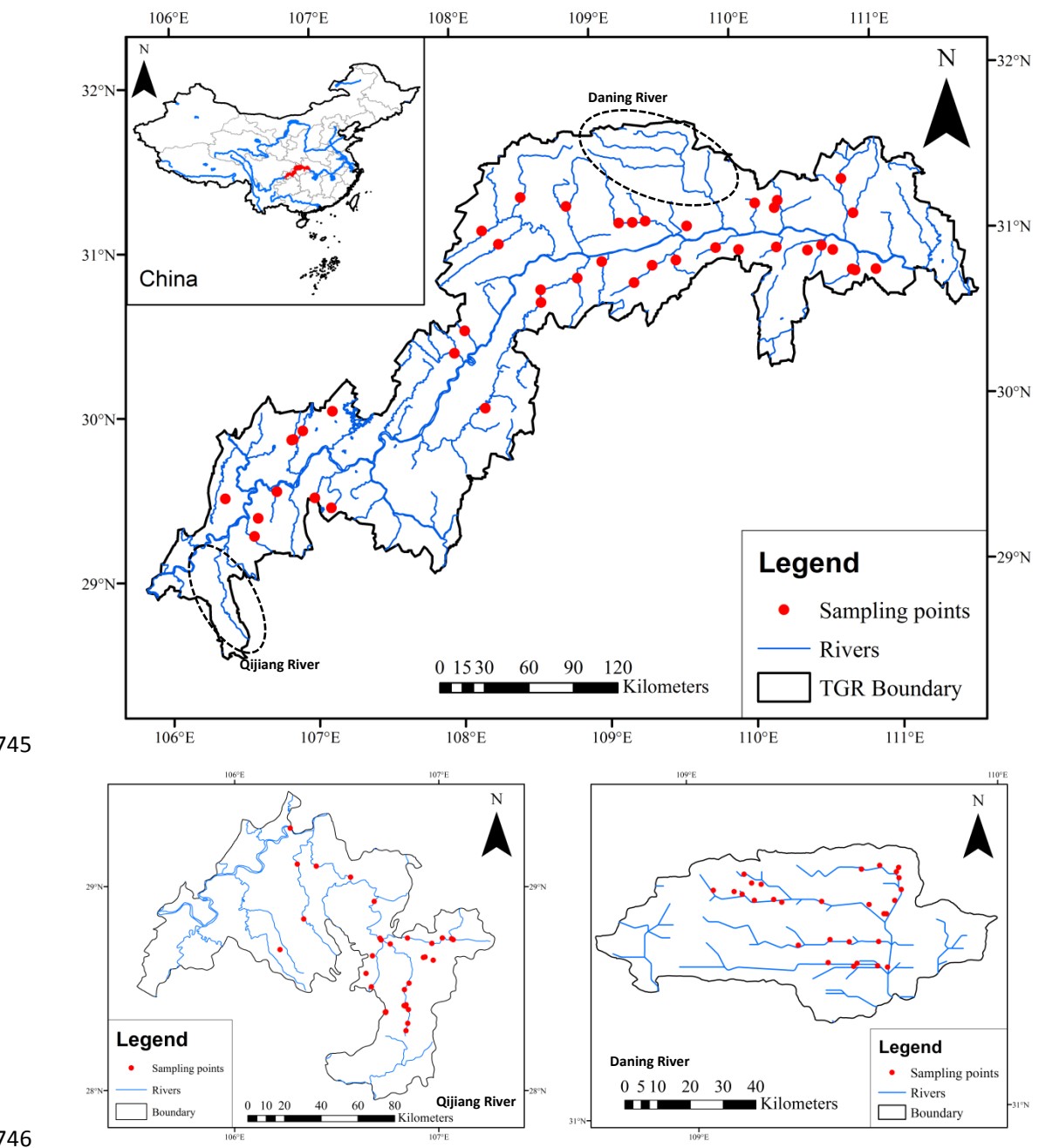



**Fig. 1.** Map of sampling locations of major rivers and streams in the Three Gorges
Reservoir region, China.

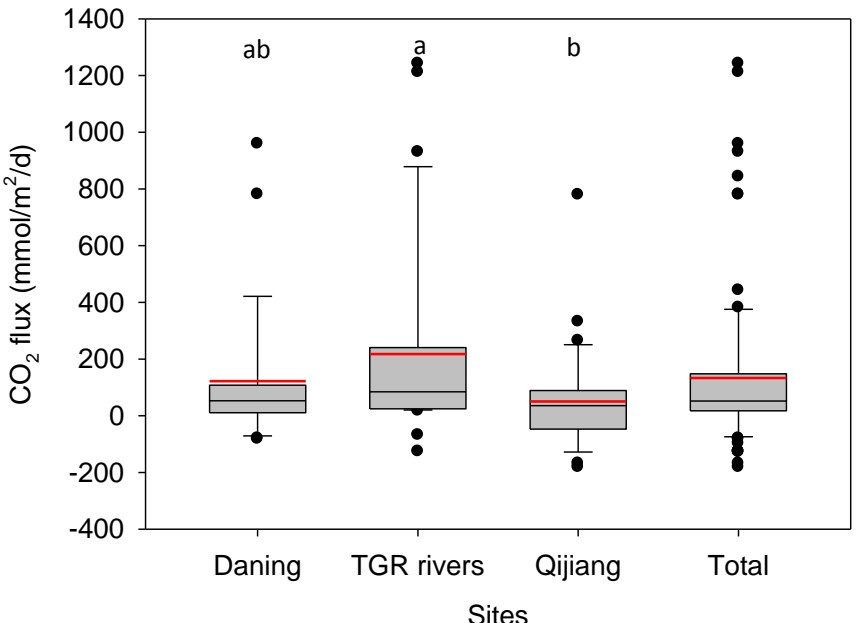


**Fig. 2.** Boxplots of $CO_2$ emission rates by floating chambers in the investigated three
river systems (different letters represent statistical differences at $p<0.05$ by
Mann-Whitney Rank Sum Test). (the black and red lines, lower and upper edges, bars
and dots in or outside the boxes demonstrate median and mean values, 25th and
75th, 5th and 95th, and <5th and >95th percentiles of all data, respectively). (For
interpretation of the references to color in this figure legend, the reader is referred
to the web version of this article) (Total means combined data from three river
systems).

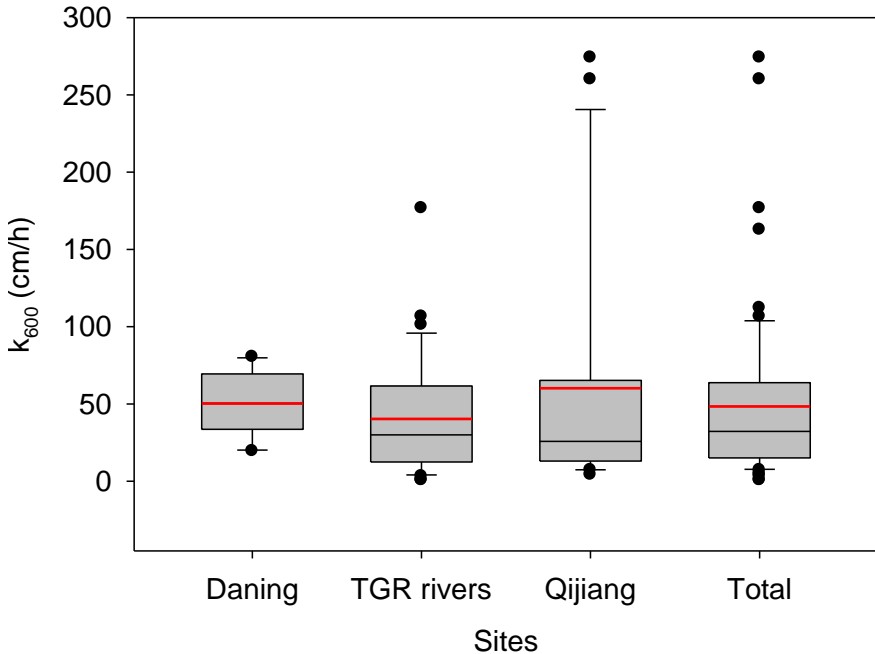


**Fig. 3.** Boxplots of $k_{600}$ levels in the investigated three river systems (there is not a
statistically significant difference in k among sites by Mann-Whitney Rank Sum Test).
(the black and red lines, lower and upper edges, bars and dots in or outside the
boxes demonstrate median and mean values, 25th and 75th, 5th and 95th, and <5th
and >95th percentiles of all data, respectively). (For interpretation of the references
to color in this figure legend, the reader is referred to the web version of this article)
(Total means combined data from three river systems).

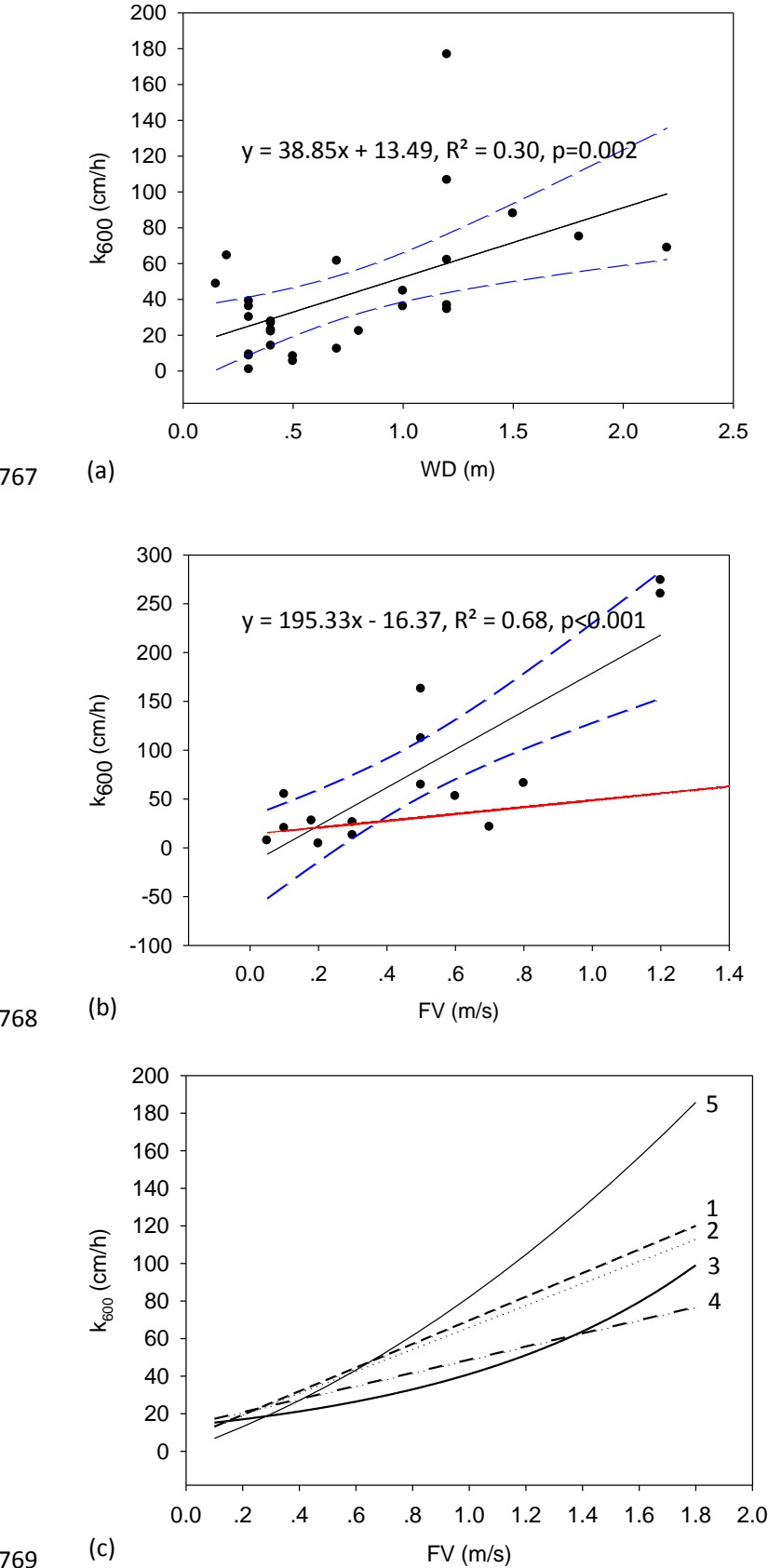

(a)
(b)
(c)
**Fig. 4.** The $k_{600}$ as a function of water depth (WD) using data from TGR rivers (a), flow
velocity (FV) using data from Qijiang (b), and comparison of the developed model

with other models (c) (others without significant relationships between k and physical factors are not shown). The solid lines show regression, the dashed lines represent 95% confidence band, and the red dash-dotted line represents the model developed by Alin et al (2011) (Extremely values of 260 and 274 cm/h are removed in panel b, the revised model would be $k_{600}$ = 62.879FV + 6.8357, $R^2$ = 0.52, p=0.019) (in panel c, 1-the revised model, 2-model from Ran et al., 2017, 3-model from Ran et al., 2015, 4-model from Alin et al,, 2011, 5-model from Liu et al., 2017) (1- $k_{600}$ = 62.879FV + 6.8357; 2- $k_{600}$ = 58.47FV+7.99; 3- $k_{600}$ = 13.677exp (1.1FV); 4- $k_{600}$ = 35 FV + 13.82; 5- $k_{600}$ = 6.5FV$^2$ + 12.9FV+0.3) (unit of k in models 1-4 is cm/h, and unit of m/d for model 5 is transferred to cm/h).