# Peer review of "1. Introduction"

_Biogeosciences, 2018_

## Referee Comment (RC1) · Anonymous Referee #1 · 29 Jun 2018

General comments: The paper "Gas transfer velocities of CO2 in subtropical monsoonal climate streams and small rivers" appears to be something of a companion piece to "Riverine CO2 supersaturation and outgassing in a subtropical monsoonal mountainous area (Three Gorges Reservoir Region) of China" https://doi.org/10.1016/j.jhydrol.2018.01.057 published in the Journal of Hydrology. In this current submission, the authors present k calculated from floating chamber flux measurements and using models, and discuss the implications of the differing approaches to k for making regional scale flux estimates. Using chambers to determine CO2 fluxes, the authors then use pCO2 to derive the gas transfer velocity. These flux-derived k values are compared to modeled k values. It is good to see the spatial aspects of the gas transfer velocity addressed. However, I do not feel that there is an

adequate consideration of the uncertainty in the estimates/calculations provided.

For the flux-derived k values, there is little provided in terms of uncertainty assessments. pCO2 was not measured, but rather was computed based on pH, alkalinity and temperature. This would have large uncertainties that then propagate into k estimates. Golub et al. (2017, doi: 10.1002/2017JG003794) note that "freshwater researchers must make significant efforts to standardize and reduce errors in pCO2 predictions". I encourage the authors to undertake a more systematic uncertainty analysis for their pCO2 values and propagate this error into uncertainty estimates for k.

Further, the authors here excluded deriving k values for samples that did not have a very large gradient in CO2 across the air-water interface. The authors chose 110 uatm as the threshold for excluding data, but this was presented without any indication of choice of threshold, making it appear rather arbitrary. Given the pH of the rivers sampled and the pCO2 that was at times undersaturated, this appears rather problematic in that it introduces bias that carries through to the regional estimates provided.

The authors in this paper refer to their k values as "observed", but these are in fact derived, and so need to have uncertainty better characterized. Upscaling from X floating chamber measurements to a river network draining 58000 km2. How many flux measurements were made with floating chambers is not clearly stated, but it appears to be about 100 all made during summer 2016. Going from summer measurements for ~100 points to annual estimates for 58000 km2 also requires some consideration of error propagation and bias. Fluxes were only retained when the floating chambers yielded linearly increasing CO2 against time, which again biases against low flux locations. Of the attempted flux measurements, what fraction was discarded?

Finally, a minor point is that the authors state several times that theirs is the first determination of k for subtropical streams and small rivers. I would point the authors towards global syntheses on CO2 evasion as well as individual studies that include k estimates.

Minor comments The figure S1 does not show the sample locations within the Daning

or Qiijiang basins. These may be the same locations as Figure 1 in Li et al. (2018) Journal of Hydrology doi: 10.1016/j.jhydrol.2018.01.057?

There are a number of grammatical issues throughout the paper that the authors should address.

---

## Referee Comment (RC2) · Anonymous Referee #2 · 13 Jul 2018

The manuscript reports on transfer velocities of CO2 (K) in streams and small rivers for assessing the gas fluxes. CO2 released from lakes and rivers has been recently recognized as an important component in the global carbon cycle. The accurate estimation of CO2 flux is still challenging primarily due to the difficulty in obtaining an appropriate K value. The topic would be of great interest for the community of scientists working on carbon cycles and can be considered for publication. However, the current version need to revised (see below).

General comments

As emphasized by the authors, the study focuses on the subtropical monsoonal streams and small rivers which are characterized by large seasonal variations in climate and discharge. Hence, the K value in these rivers should also have obvious

seasonality. Unfortunately, the samples presented in this study were collected in the rainy season. The K value were calculated based on the one-time sampling campaign, which might result in a certain amount of errors on the annual flux estimation. Regarding this, the uncertainty of the sampling data and the calculations, as well as the reliability of the argument should be sufficiently discussed.

In my point of view, the variation in K value of the rivers studied are obvious and need to be discussed. In addition, the spatial difference of K values is only sorted out for the three river systems (Daning, Qijiang and TGR). I would suggest the authors examine the variations of K following the physical characteristics of rivers (such as the current velocity, slope and the water depth) or/and the river orders.

The pCO2 calculated in this paper is between 50-4830ppm, which indicates that the river pCO2 value is sometimes much lower than that of the atmosphere, that is, the studied rivers can sometimes absorb CO2 from the atmosphere. It seems that the annual CO2 flux for the whole basin was calculated in this paper based on the averaged K value from the observed results using floating chamber method. The question is that is it reliable to estimate both directions of the CO2 flux at the air-water interface (including river CO2 outgassed to the atmosphere and the atmospheric CO2 input to rivers) by using the same K value? Or what uncertainty will it cause?

This study measured DOC, DTN, and DTP, but the authors did not mention these measurements in the discussion section. What is the relationship between these variables and the K values?

Specific comments

L 94-97: I would suggest rephrase these sentences, since they cannot convey clearly the real contribution or scientific merit of this study.

L 111-112, 117: The classification method of the river order used here should be clarified. The number of a river order defined by different classification system may

represent different size or hierarchy of a river.

L 214-216: This statement is problematic. Clearly, the studied rivers are not always supersaturated reference to atmospheric CO2 as the pCO2 in rivers is between 50-4830 uatm.

L 274-285 These arguments need more solid evidence to support. As mentioned in the general comments, I would suggest that the authors focus on discussions on relationship between spatial change in K values and physical characteristics of rivers or/ and the river orders.

L 497-498 The water area is a very critical parameter for the calculation of CO2 flux in a basin, so the acquisition of water area is essential and should be described more in detail. For example, what is the resolution of the satellite image? In addition, the variation of surface area of water between wet and dry seasons should be considered.

Finally, I would suggest the authors polish the English grammar and writing, as well as the figs presenting.

―――――――――――――――――――――――

---

## Referee Comment (RC3) · Anonymous Referee #3 · 17 Jul 2018

General comments

The manuscript of Li et al. presents measured CO2 fluxes, transport coefficients based on CO2, and calculated pCO2 data of running waters in a subtropical monsoonal climate zone. These data are complemented by among others water chemistry parameters such as DOC, DTN, DTP, as well as hydrogeomorphology data (e.g. water depth, flow velocity). They provide data and insights about transport coefficients for a so far understudied region and highlight the spatial variability and subsequent uncertainty for regional upscale estimates.

By investigating the key parameter for CO2 flux estimates - the transport coefficient - in an understudied region, Li et al. address a very relevant topic. Narrowing down the uncertainties of regional upscaling estimates of riverine CO2 fluxes is of wide interest,

hence this study would make a good contribution to the literature and the subject matter is thus of interest to Biogeosciences readers.

However, in my opinion, the manuscript has some problems:

(1) The terminology used in this manuscript is quite confusing to me. It seems to me that "streams", "rivers", "river networks" are used interchangeably (without definition and consistency), which makes it hard to follow the red line of the story. The terminology needs to be clarified and unified.

(2) The sampling design is not very clear to me. All investigated running waters seem to be in the Three Gorges Reservoir (TGR) region, but in addition two larger streams (Daning and Qijiang) were sampled. In the results and discussion, these investigated running waters are combined, sometimes split, which makes it hard to follow (in the main text and tables). In my opinion, these three "regions" need to be presented in a unified way (always separated or combined, possibly both in each table and figure), and presented more clearly in the text.

(3) One of the main messages is the presentation of transport coefficients in a subtropical monsoonal climate zone, which is interesting, but I can imagine that there is a large difference in the wet and dry season. However, all the measurements were done in the wet season. I think this issue should be clearly acknowledged and discussed.

(4) There are two technical issues: (i) The measurements with the floating chambers are poorly described. The only information Li et al. provide is that the floating chambers were "deployed". If the flux measurements are done in an anchored or free floating manner is critical (see e.g. Lorke, A., Bodmer, P., Noss, C., Alshboul, Z., Koschorreck, M., Somlai-Haase, C., Bastviken, D., Flury, S., McGinnis, D. F., Maeck, A., Müller, D., and Premke, K.: Technical note: drifting versus anchored flux chambers for measuring greenhouse gas emissions from running waters, Biogeosciences, 12, 7013-7024, https://doi.org/10.5194/bg-12-7013-2015, 2015.). Hence, this issue needs to be addressed clearly. (ii) It seems that all the flux and pCO2 measurements were done distributed during the day. The fact that there is a diurnal cycle of CO2 was not considered (see e.g. Pascal Bodmer, Marlen Heinz, Martin Pusch, Gabriel Singer, Katrin Premke, Carbon dynamics and their link to dissolved organic matter quality across contrasting stream ecosystems, Science of The Total Environment, Volume 553, 2016, Pages 574-586, ISSN 0048-9697, https://doi.org/10.1016/j.scitotenv.2016.02.095.), and values directly compared. This issue should at least be discussed.

(5) Developing models to estimate transport coefficients is meaningful, but the process of the model development is poorly described. Additionally, which data were used for the models, and which not is confusing to me (goes along with my comment (2) above).

(6) From what I see in these data, there are several running waters undersaturated with respect to CO2 (Fig. S2 and Fig. 1), and hence a sink of CO2. This aspect is totally neglected and the investigated running waters are generalized as CO2 sources to the atmosphere. In my opinion, this aspect of influxes of CO2 is very valuable and should be properly discussed.

(7) As far as I see, there is some arbitrariness regarding data handling/processing. The cut-off at 110 $\mu$atm (line 198) for the air-water CO2 gradient for k600 calculations, as well as "When several extremely values are removed..." (line 303), needs to be described/demonstrated/justified much more clear.

(8) CO2 fluxes were measured, while pCO2 and transport coefficients were calculated. This should be clearly stated throughout the manuscript to be transparent.

(9) I am not a native English speaker, but I think that the manuscript should be revised for the English language. (see exemplarily in the specific comments and technical corrections below)

I think this is a valuable study, but the combination of the points mentioned above make the manuscript hard to follow and the conclusions and main messages drawn in the current state of the manuscript too general. In a revised version, the study would get

more shaped, more detailed and informative, and the conclusions and main messages can be specified and more related to the investigated region.

Specific comments

Abstract:

Line 20: Indicate how many river networks (see general comment (2))

Line 24: As far as I understood not when all data were included. Please be more specific here.

Lines 30 – 33: This is not really new. Maybe you can specify this statement for the investigated region?

Introduction:

Line 41: Bastviken et al., 2011 totally focuses on CH4. I suggest replacing this reference with a more suitable one.

Line 42: But you did not present "new accurate measurement techniques" in your study, what are your reasons to mention this in the introduction?

Lines 50 – 58: This equation is pretty standard knowledge and can just be described in words here. The equation can be moved to the methods.

Line 63: The standardized transport coefficient (k600) should be explained here.

Line 80: You set the scene of seasonal precipitation, but in the study, you only measure in the wet season. This is contradictory. This issue should be discussed.

Lines 84-89: Kind of repetition and partially contradictory to the text in lines 43-49. These two paragraphs should be unified and clarified.

Lines 92-99: The relevance of the study (first time in this region, etc.), the objectives and how these objectives were approached should be written more clearly. At this point, the input parameters for the model development is totally unclear.

[Figure]

Materials and methods:

Line 105: In my opinion, Figure S1 should go to the main text. There are no sampling points for Daning and Qijiang, which is confusing to me.

Lines 105-109: Please add a reference for this statement.

Lines 110-118: Please see my general comment (2): Please restructure this, make clear how many running waters were sampled where, the size of the sampled running waters (Strahler stream order is fine), and why in these three regions. Otherwise, it is hard to follow your storyline.

Line 141-142: What is "PP"?

Line 148: I don't really understand what you mean by this sentence, please revise.

Lines 155-156: This sentence is confusing to me, please revise.

Line 158: Do you mean CO2SYS? If yes, please add the corresponding reference.

Line 167: What was the brand of the tubing?

Line 170: What is DC?

Line 173: Please see my general comment (4) (i)

Line 177: The units are confusing to me. Why is there two times pressure and temperature? Please double check if the units match up in the end, to me they do not.

Line 187: Please be more specific: k was calculated by reorganizing Eq (1)

Line 192: Sc to the power of 0.5? This seems weird. What do you mean here?

Line 198: Please justify the cut-off at 110 $\mu$atm. Maybe add a figure to the supplementary material.

Line 203: I read about water depth and current velocity the first time here. These measurements need to be described before.

Line 213: The pH is quite high. This in combination with influxes of CO2 requires at least a short discussion about chemical enhancement.

Line 214: Please see my general comment (6)

Lines 218-222: This paragraph belongs to the discussion section.

Lines 223-227: This paragraph should be revised because it is not very clearly written. Please add the p-values to the text in case of significances.

Lines 235-237: What is the meaning of this ratio? Please add a few words what the reader can get from this information.

Line 242: These models and how you developed them should be explained better (in the method section).

Lines 246-248: I do not understand this sentence. What do you mean by "binned"?

Discussion:

Lines 270-274: How does this paragraph support the discussion of your study?

Lines 286-288: No significances... But still, you developed the models considering all data? This is not at all clear to me. Did you split/separate the data set for the models? This is not clear in Table 2. Please see my general comment (2). I think these data/models are valuable, but at the moment they seem arbitrary and should be better explained. This would help to give them more weight.

Lines 286-309: I see a lot of results here, which are presented in the discussion for the first time. The part presenting pure results should be moved to the results section.

Line 303: Please justify the removal of "extremely values". Maybe add a figure to the supplementary material. If there is no objective criteria and justification, I do not see why data should be removed.

Lines 327-333: Why discussing k values and not k600 values? I think this needs to be

unified/consistent throughout the manuscript.

Conclusion:

Lines 358-360: Very general, but actually, the regions had to be separated/ split, no?

Lines 368-369: I think you should focus the conclusion on the investigated region.

Tables:

Table 3: b) Why not presenting k600 values here which can be directly compared with other studies?

Figures:

Fig. S1: Was always everything sampled at each point? "Samples" should be replaced by "sampling point"

Fig. S4: There is no reference to this figure in the main text.

Technical corrections

Line 22: Delete "were"

Line 44: add "by" or "via" before "floating chambers"

Line 59: Replace "precisely" with "well"

Line 127: "consisted" instead of "consists"

Line 139: "pH sonde" "was" instead of "is"

Line 225: "Daning" instead of "Danning"

---

## Author Comment (AC1) · 7 Sep 2018

General comments: The paper "Gas transfer velocities of CO2 in subtropical monsoonal climate streams and small rivers" appears to be something of a companion piece to "Riverine CO2 supersaturation and outgassing in a subtropical monsoonal mountainous area (Three Gorges Reservoir Region) of China" https://doi.org/10.1016/j.jhydrol.2018.01.057 published in the Journal of Hydrology. In this current submission, the authors present k calculated from floating chamber flux measurements and using models, and discuss the implications of the differing approaches to k for making regional scale flux estimates. Using chambers to determine CO2 fluxes, the authors then use pCO2 to derive the gas transfer velocity. These flux-derived k values are compared to modeled k values. It is good to see the spatial

aspects of the gas transfer velocity addressed. However, I do not feel that there is an adequate consideration of the uncertainty in the estimates/calculations provided. For the flux-derived k values, there is little provided in terms of uncertainty assessments.

Response: We thanked the referee for the comment. In our previous article, we studied the pCO2 and emission rate as well as their controls from fluvial networks in the TGR area, which is based on two field works in the TGR region, and the diffusive models from other studies were used. In this study, we attempted to derive k levels and develop the gas transfer model in this area (mountainous streams and small rivers) for more accurate quantification of CO2 areal flux, and also to serve for the fluvial networks in the Yangtze River or others with similar hydrology and geomorphology. In addition, we did more detailed field study in the two contrasting rivers Daning and Qijiang for developing models (see the sampling locations map). This study clearly showed original contribution to the current literature and this study is different than the article published in the Journal of Hydrology. We clearly state the new contributions and significances in the last paragraph of the "Introduction" as follows.

"Our recent study preliminarily investigated pCO2 and air – water CO2 areal flux as well as their controls from fluvial networks in the Three Gorges Reservoir (TGR) area (Li et al., 2018). The past study was based on two field works, and the diffusive models from other continents were used. In this study, we attempted to derive k levels and develop the gas transfer model in this area (mountainous streams and small rivers) for more accurate quantification of CO2 areal flux, and also to serve for the fluvial networks in the Yangtze River or others with similar hydrology and geomorphology. Moreover, we did detailed field campaigns in the two contrasting rivers Daning and Qijiang for models (Fig. 1). The study thus clearly stated distinct differences than the previous study (Li et al., 2018) by the new contributions of specific objectives and data supplements, as well as wider significance."

We added a section (4.4.) for "Uncertainty assessment of pCO2 and flux-derived k values" in the part of "Discussion".

The uncertainty of flux-derived k values mainly stem from air–water gradient of CO2 ($\Delta$pCO2 in ppm) and flux measurements (Golub et al., 2017; Lorke et al., 2015; Bodmer et al., 2016). Thus we provided uncertainty assessments caused by dominant sources of uncertainty from measurements of aquatic pCO2 and CO2 areal flux since uncertainty of atmospheric CO2 measurement could be neglected.

In our study, aquatic pCO2 was computed based on pH, alkalinity and water temperature rather than directly measured. Recent studies highlighted pCO2 uncertainty caused by systematic errors over empiric random errors (Golub et al., 2017). Systematic errors are mainly attributed to instrument limitations, i.e., sondes of pH and water temperature. The relative accuracy of temperature meters was $\pm$0.1 0C according to manufacturers' specifications, thus the uncertainty of water T propagated on uncertainty in pCO2 was minor (Golub et al., 2017). Systematic errors therefore stem from pH, which has been proved to be a key parameter for biased pCO2 estimation calculated from aquatic C system (Li et al., 2013; Abril et al., 2015). We used a high accuracy of pH electrode and the pH meters were carefully calibrated using CRMs, and in situ measurements showed an uncertainty of $\pm$0.01. We then run an uncertainty of $\pm$0.01 pH to quantify the pCO2 uncertainty, and an uncertainty of $\pm$3% was observed. Systematic errors thus seemed to show little effects on pCO2 errors in our study.

Random errors are from repeatability of carbonate measurements. Two replicates for each sample showed the uncertainty of within $\pm$5%, indicating that uncertainty in pCO2 calculation from alkalinity measurements could be minor.

The measured pH ranges also exhibited great effects on pCO2 uncertainty (Hunt et al., 2011; Abril et al., 2014). At low pH, pCO2 can be overestimated when calculated from pH and alkalinity (Abril et al., 2014). Samples for CO2 fluxes estimated from pH and alkalinity showed pH average of 8.39$\pm$0.29 (median 8.46 with quartiles of 8.24-8.56) (n=115). Thus, overestimation of calculated CO2 areal flux from pH and alkalinity is likely to be minor. Further, contribution of organic matter to non-carbonate alkalinity is likely to be neglected because of low DOC (mean 6.67 mg/L; median 2.51 mg/L) (Hunt

et al., 2011; Li et al., 2013).

Recent study reported fundamental differences in CO2 emission rates between anchored chambers (ACs) and freely drifting chambers (DFs) (Lorke et al., 2015), i.e., ACs biased the gas areal flux higher. However, some studies observed that ACs showed reasonable agreement with other flux measurement techniques (Crawford et al., 2013; Galfalk et al., 2013), and this straightforward, inexpensive and relatively simple method AC was widely used (Ran et al., 2017). Water-air interface CO2 flux measurements were made using ACs in our studied streams and small rivers because of relatively high current velocity; otherwise, floating chambers will travel far during the measurement period. In addition, inflatable rings were used for sealing the chamber headspace and submergence of ACs was minimal, therefore, our measurements were potentially overestimated but reasonable.

pCO2 was not measured, but rather was computed based on pH, alkalinity and temperature. This would have large uncertainties that then propagate into k estimates. Golub et al. (2017, doi: 10.1002/2017JG003794) note that "freshwater researchers must make significant efforts to standardize and reduce errors in pCO2 predictions". I encourage the authors to undertake a more systematic uncertainty analysis for their pCO2 values and propagate this error into uncertainty estimates for k.

Response: Thanks for the comment. We reported the quality control such as systematic errors and random errors of pH and alkalinity, water temperature, as well as non-carbonate alkalinity effects. Please refer to the response above.

Further, the authors here excluded deriving k values for samples that did not have a very large gradient in CO2 across the air-water interface. The authors chose 110 uatm as the threshold for excluding data, but this was presented without any indication of choice of threshold, making it appear rather arbitrary. Given the pH of the rivers sampled and the pCO2 that was at times undersaturated, this appears rather problematic in that it introduces bias that carries through to the regional estimates provided.

Response: We addressed this issue as follows. "Prior to statistical analysis, we excluded k600 data for samples with the air-water pCO2 gradient <110 $\mu$atm, since the error in the k600 calculations drastically enhances when âŬşpCO2 approaches zero (Borges et al., 2004), and datasets with âŬşpCO2 >110 $\mu$atm provide an error of <10% on k600 computation (see Fig. 1 as follows)"

The additional section 4.4 Uncertainty assessment of pCO2 and flux-derived k values included uncertainty of pH and scaling-up estimation, for example, effects of chemical enhancement for quite high pH values.

The authors in this paper refer to their k values as "observed", but these are in fact derived, and so need to have uncertainty better characterized. Upscaling from X floating chamber measurements to a river network draining 58000 km2. How many flux measurements were made with floating chambers is not clearly stated, but it appears to be about 100 all made during summer 2016. Going from summer measurements for 100 points to annual estimates for 58000 km2 also requires some consideration of error propagation and bias. Fluxes were only retained when the floating chambers yielded linearly increasing CO2 against time, which again biases against low flux locations.

Response: We have changed "observed, or measured" to "flux-derived or derived", and discussed the uncertainty of k values as mentioned above. We agreed that more samples could improve the CO2 estimates, while our sampling locations were much more or at least comparable to the other publications.

A total of 115 discrete grab samples were collected (each sample consisted of three replicates). Floating chambers with replicates were deployed in 101 sites (32 sampling sites in Daning, 37 sites in TGR river networks and 32 sites in Qijiang). The sampling period covers spring and summer season, our sampling points are reasonable considering a water area of 433 km2. For example, 16 sites were collected for Yangtze system to examine hydrological and geomorphological controls on pCO2 (i.e., Liu et al., 2017), and 17 sites for dynamic biogeochemical controls on riverine pCO2 in the

Yangtze basin (Liu et al., 2016, Global Biogeochemical Cycles).

In our sampling points, all measured fluxes were retained since the floating chambers yielded linearly increasing $CO_2$ against time following manufacturer' specification.

Liu, S., Lu, X.X., Xia, X., Yang, X., Ran, L., 2017. Hydrological and geomorphological control on $CO_2$ outgassing from low-gradient large rivers: An example of the Yangtze River system. Journal of Hydrology 550, 26-41.

Of the attempted flux measurements, what fraction was discarded?

Response: We revised as follows. "Prior to statistical analysis, we excluded k600 data for samples with the air-water $pCO_2$ gradient <110 $\mu$atm, since the error in the k600 calculations drastically enhances when âŰşpCO2 approaches zero (Borges et al., 2004; Alin et al., 2011), and datasets with âŰşpCO2 >110 $\mu$atm provide an error of <10% on k600 computation. Thus, we discarded the samples (36.7% of sampling points with flux measurements) with âŰşpCO2 <110 $\mu$atm for k600 model development, while for the flux estimations from diffusive model and floating chambers, all samples were included."

Finally, a minor point is that the authors state several times that theirs is the first determination of k for subtropical streams and small rivers. I would point the authors towards global syntheses on $CO_2$ evasion as well as individual studies that include k estimates.

Response: We agree the comment, and addressed this issue. Several sentences for implication of k determination and comparison with other k studies were added in the part of Discussion. We also re-organised and added sub-headings of "Discussion".

"4.1. Determined k values relative to world rivers; 4.2. Hydraulic controls of k600; 4.3. Implications for large scale estimation; 4.4. Uncertainty assessment of $pCO_2$ and flux-derived k values"

Minor comments The figure S1 does not show the sample locations within the Daning or Qiijiang basins. These may be the same locations as Figure 1 in Li et al. (2018)

Journal of Hydrology doi: 10.1016/j.jhydrol.2018.01.057?

Response: The sampling sites and study aims are different than previous study (Please refer to the section of 2.1 and the last paragraph in the "Introduction"). In the revised Ms, we supplied the map of sampling locations in the main text as Fig. 1.

We added several sentences in the section of "INTRODUCTION" to highlight the differences between our study and previous study, as well as what is advanced by this study (please refer to the first Comment).

There are a number of grammatical issues throughout the paper that the authors should address.

Response: We carefully edited English, and also get helps from a native English scientist.

The additional Table was added to the Table S2 in SOM.

We also provided tracked PDF as supplement for your review.

Please also note the supplement to this comment:
https://www.biogeosciences-discuss.net/bg-2018-227/bg-2018-227-AC1-supplement.pdf

————————————————————

Fig. 1. Theoretical error ($\pm$%) on the computation of the gas transfer velocity of $CO_2$ ($k600$) as a function of the air–water gradient of $CO_2$ ($\Delta pCO_2$ in ppm), assuming a constant uncertainty on $\Delta pCO_2$ of $\pm 3$% (Borges et al., 2004).

**Fig. 1.**

[Figure]

**Fig. 1.** Map of sampling locations of major rivers and streams in the Three Gorges Reservoir region, China (Please see in the main text).

**Fig. 2.**

The additional Table was added to the Table S2 in SOM.

|  | Current velocity m/s | Water depth m/s | Wind speed m/s | K600 cm/h | Reference |
|---|---|---|---|---|---|
| Mekong tributary | 0.39±0.28 | 0.9±0.6 | 0.7±0.6 | 23.3±17.3 | Alin et al., 2011 |
| Yellow | 1.8 |  | 1.8 (1.2-2.3) | 42±17 | Ran et al., 2015 |
| Yangtze | 1.2±1.5 |  | 1.2±1.1 | 38±40 | Liu et al., 2017 |
| Mekong stem | 0.92±0.42 |  | 1.8±1.2 | 15±9 | Alin et al., 2011 |

**Fig. 3.**

---

## Author Comment (AC2) · 7 Sep 2018

The manuscript reports on transfer velocities of CO2 (K) in streams and small rivers for assessing the gas fluxes. CO2 released from lakes and rivers has been recently recognized as an important component in the global carbon cycle. The accurate estimation of CO2 flux is still challenging primarily due to the difficulty in obtaining an appropriate K value. The topic would be of great interest for the community of scientists working on carbon cycles and can be considered for publication. However, the current version need to revised (see below).

Response: We thanked the referee for the positive comment. We revised the Ms based on the comments and suggestions as follows.

[Figure]

General comments As emphasized by the authors, the study focuses on the subtropical monsoonal streams and small rivers which are characterized by large seasonal variations in climate and discharge. Hence, the K value in these rivers should also have obvious seasonality. Unfortunately, the samples presented in this study were collected in the rainy season. The K value were calculated based on the one-time sampling campaign, which might result in a certain amount of errors on the annual flux estimation. Regarding this, the uncertainty of the sampling data and the calculations, as well as the reliability of the argument should be sufficiently discussed.

Response: We agreed the comment. Sampling seasonality largely impacted riverine $pCO_2$ and gas transfer velocity and thus water-air interface $CO_2$ evasion rate. In our Ms, we sampled waters in the rainy season due to that it showed wider range of flow velocity and thus rainy season covered the k levels in the whole hydrological season. Rainy season generally had higher current velocity and thus higher gas transfer velocity, while aquatic $pCO_2$ was variable with seasonality.

We added a section of "4.4. Uncertainty assessment of $pCO_2$ and flux-derived k values" in the part of "Discussion". In this section, effects and uncertainty of sampling seasonality on errors of annual $CO_2$ flux estimation were included.

"Sampling seasonality considerably regulated riverine $pCO_2$ and gas transfer velocity and thus water-air interface $CO_2$ evasion rate (Li et al., 2012; Ran et al., 2015). We sampled waters in the rainy season due to that it showed wider range of flow velocity and thus rainy season covered the k levels in the whole hydrological season. Rainy season generally had higher current velocity and thus higher gas transfer velocity (Ran et al., 2015), while aquatic $pCO_2$ was variable with seasonality. We recently reported that riverine $pCO_2$ in the wet season was 81% the level in the dry season (Li et al., 2018), and prior study on the Yellow River reported that k level in the wet season was 1.8-fold higher than in the dry season (Ran et al., 2015), while another study on the Wuding River demonstrated that k level in the wet season was 83%-130% of that in the dry season (Ran et al., 2017). Thus, we acknowledged a certain amount of errors on

the annual flux estimation from one-time sampling campaign during the wet season in the TGR area, while this uncertainty could not be significant because that the diluted pCO2 could alleviate the potentially increased k level in the wet season."

In my point of view, the variation in K value of the rivers studied are obvious and need to be discussed. In addition, the spatial difference of K values is only sorted out for the three river systems (Daning, Qijiang and TGR). I would suggest the authors examine the variations of K following the physical characteristics of rivers (such as the current velocity, slope and the water depth) or/and the river orders.

Response: Spatial differences of k values were discussed for the three rivers systems (please refer to section 4.2). We discussed the controls of physical characteristics of current velocity, slope and the water depth, while river orders were not extracted.

"This could substantiate the higher k600 levels and spatial changes in k600 values of our three river systems. For instance, similar to other turbulent rivers in China (Ran et al., 2015; Ran et al., 2017), high k600 values in the TGR, Daning and Qjiang rivers were due to mountainous terrain catchment, high current velocity (10 – 150 cm/s) (Fig. 4b), bottom roughness, and shallow water depth (10 - 150 cm) (Fig. 4a). It has been suggested that shallow water enhances bottom shear, and the resultant turbulence increases k values (Alin et al., 2011; Raymond et al., 2012). These physical controls are highly variable across environmental types (Figs. 4a and 4b), hence, k values are expected to vary widely (Fig. 3). The k600 values in the TGR rivers showed wider range (1-177 cm/h; Fig. 3; Table S1), spanning more than 2 orders of magnitude across the region, and it is consistent with the considerable variability in the physical processes on water turbulence across environmental settings. Similar broad range of k600 levels was also observed in the China's Yellow basin (ca. 0-123 cm/h) (Ran et al., 2015; Ran et al., 2017).

Contrary to our expectations, no significant relationship was observed between k600 and water depth, and current velocity using the entire data in the three (TGR, Danning

and Qjiang) river systems (Fig. S4). There were not statistically significant relationships between k600 and wind speed using separated data or combined data, and it is consistent with earlier studies (Alin et al., 2011; Raymond et al., 2012). Flow velocity showed linear relation with k600, and the extremely high value of k600 was observed during the periods of higher flow velocity (Fig. S4a) using combined data. Similar trend was also observed between water depth and k600 values (Fig. S4b). The lack of strong correlation between k600 and physical factors are probably due to combined effect of both flow velocity and water depth, as well as large diversity of channel morphology, both across and within river networks in the entire catchment (60, 000 km2). This is further collaborated by weak correlations between k600 and flow velocity in the TGR rivers (Fig. 4), where one or two samples were taken for a large scale examination. k600 as a function of water depth was obtained in the TGR rivers, but it explained only 30% of the variance in k600. However, model using data from Qijiang could explain 68% of the variance in k600 (Fig. 4b), and it was in line with general theory. Nonetheless, k600 from our flow velocity based model (Fig. 4b) was potentially largely overestimated with consideration of other measurements (Alin et al., 2015; Ran et al., 2015; Ran et al., 2017). When several extremely values were removed, k600 (cm/h) was parameterized as follows (k600 = 62.879FV + 6.8357, $R^2$ = 0.52, p=0.019, FV-flow velocity with a unit of m/s), and this revised model was in good agreement with the model in the river networks of the Yellow River (Ran et al., 2017), but much lower than the model developed in the Yangtze system (Liu et al., 2017) (Fig. 4c). This was reasonable because of k600 values in the Yangtze system were from large rivers with higher turbulence than Yellow and our studied rivers. Furthermore, the determined k600 using FCs was, on average, consistent with the revised model (Table 2). These differences in relationship between spatial changes in k600 values and physical characteristics further corroborated heterogeneity of channel geomorphology and hydraulic conditions across the investigated rivers."

The pCO2 calculated in this paper is between 50-4830ppm, which indicates that the river pCO2 value is sometimes much lower than that of the atmosphere, that is, the

studied rivers can sometimes absorb CO2 from the atmosphere. It seems that the annual CO2 flux for the whole basin was calculated in this paper based on the averaged K value from the observed results using floating chamber method. The question is that is it reliable to estimate both directions of the CO2 flux at the air-water interface (including river CO2 outgassed to the atmosphere and the atmospheric CO2 input to rivers) by using the same K value? Or what uncertainty will it cause?

Response: We thanked for your critical comment. Worldwide studies reported the dependence of k on flow dynamics; while k average from statistical analysis with âŰşpCO2 >110 $\mu$atm was normally used in riverine CO2 flux estimation in rivers where a broad of range of pCO2 occurred (Alin et al., 2011). Considering that k was largely dependent on hydraulic characteristics, this uncertainty was not discussed. The method employed here for scaling up CO2 estimation was widely used (e.g., Alin et al., 2011). We added one section 4.4 ( see Discussion part) on systematic errors and empiric random errors of pCO2, CO2 areal flux and k from pCO2 determinations, sampling seasonality, flux measurements etc.

This study measured DOC, DTN, and DTP, but the authors did not mention these measurements in the discussion section. What is the relationship between these variables and the K values?

Response: k values are reported to be dominated by physical characteristics of river systems (Borges et al., 2004; Alin et al., 2011; Raymond et al., 2012). The basic water quality parameters were not further discussed. We re-wrote the pattern of nutrients in the "Result" section.

Specific comments L 94-97: I would suggest rephrase these sentences, since they cannot convey clearly the real contribution or scientific merit of this study.

Response: We revised as follows. "Our new contributions to the literature include (1) determination and controls of k levels for small rivers and streams in subtropical areas of China, and (2) new k models developed in the subtropical mountainous river

networks"

L 111-112, 117: The classification method of the river order used here should be clarified. The number of a river order defined by different classification system may represent different size or hierarchy of a river.

Response: We addressed this issue, and provided the map of sampling location (Fig. 1). We also supplied methodology of water areal extraction in section 2.5.

L 214-216: This statement is problematic. Clearly, the studied rivers are not always supersaturated reference to atmospheric $CO_2$ as the $pCO_2$ in rivers is between 50-4830 uatm.

Response: We changed to "$pCO_2$ varied between 50 and 4830 $\mu$atm with mean of 846 $\pm$ 819 $\mu$atm (Table 1). There were 28.7% of samples that had $pCO_2$ levels lower than 410 $\mu$atm, while the studied rivers were overall supersaturated with reference to atmospheric $CO_2$ and act as a source for the atmospheric $CO_2$."

L 274-285 These arguments need more solid evidence to support. As mentioned in the general comments, I would suggest that the authors focus on discussions on relationship between spatial change in K values and physical characteristics of rivers or/ and the river orders.

Response: Spatial differences of k values were discussed for the three rivers systems (please refer to section 4.2). We discussed the controls of physical characteristics of current velocity, slope and the water depth.

L 497-498 The water area is a very critical parameter for the calculation of $CO_2$ flux in a basin, so the acquisition of water area is essential and should be described more in detail. For example, what is the resolution of the satellite image? In addition, the variation of surface area of water between wet and dry seasons should be considered.

Response: We provided the information for the acquisition of water area and citation was included (please refer to Methodology section).

**2.5. Estimation of river water area**

Water surface is an important parameter for $CO_2$ efflux estimation, while it depends on its climate, channel geometry and topography. River water area therefore largely fluctuates with much higher areal extent of water surface particularly in monsoonal season. However, most studies do not consider this change, and a fraction of the drainage area is used in river water area calculation (Zhang et al., 2017). In our study, a 90 m resolution SRTM DEM (Shuttle Radar Topography Mission digital elevation model) data and Landsat images in dry season were used to delineate river network, and thus water area (Zhang et al., 2018), whilst, stream orders were not extracted. Water area of river systems is generally much higher in monsoonal season in comparison to dry season, for instance, Yellow River showed 1.4-fold higher water area in the wet season than in the dry season (Ran et al., 2015). Available dry-season image was likely to underestimate $CO_2$ estimation.

Ran, L.S., Lu, X.X., Yang, H., Li, L.Y., Yu, R.H., Sun, H.G., Han, J.T., 2015. CO2 outgassing from the Yellow River network and its implications for riverine carbon cycle. Journal of Geophysical Research-Biogeosciences 120, 1334-1347. Zhang, T., Li, J., Pu, J., Martin, J.B., Khadka, M.B., Wu, F., Li, L., Jiang, F., Huang, S., Yuan, D., 2017. River sequesters atmospheric carbon and limits the CO2 degassing in karst area, southwest China. Science of The Total Environment 609, 92-101. Zhang, J., Li, SY., Dong, RZ., Jiang, CS., 2018. Physical evolution of the Three Gorges Reservoir in Holocene using advanced SVM on Landsat images and SRTM DEM data. Environ Sci Pollut Res 25, 14911-14918.

Finally, I would suggest the authors polish the English grammar and writing, as well as the figs presenting.

Response: We carefully edited the Ms and re-organized the Tables and Figures presentation.

---

## Author Comment (AC3) · 7 Sep 2018

General comments The manuscript of Li et al. presents measured CO2 fluxes, transport coefficients based on CO2, and calculated pCO2 data of running waters in a subtropical monsoonal climate zone. These data are complemented by among others water chemistry parameters such as DOC, DTN, DTP, as well as hydrogeomorphology data (e.g. water depth, flow velocity). They provide data and insights about transport coefficients for a so far understudied region and highlight the spatial variability and subsequent uncertainty for regional upscale estimates. By investigating the key parameter for CO2 flux estimates - the transport coefficient - in an understudied region, Li et al. address a very relevant topic. Narrowing down the uncertainties of regional upscaling estimates of riverine CO2 fluxes is of wide interest, hence this study would make a good contribution to the literature and the subject matter is thus of interest to Biogeosciences readers.

Response: We thank you for your overall positive comments, and accordingly revised the Ms.

However, in my opinion, the manuscript has some problems: (1) The terminology used in this manuscript is quite confusing to me. It seems to me that "streams", "rivers", "river networks" are used interchangeably (without definition and consistency), which makes it hard to follow the red line of the story. The terminology needs to be clarified and unified.

Response: Based on the delineation of river systems by Alin et al., 2011 (JGR), small rivers and streams encompass rivers with channels < 100 m. We have clarified the term in the method.

(2) The sampling design is not very clear to me. All investigated running waters seem to be in the Three Gorges Reservoir (TGR) region, but in addition two larger streams (Daning and Qijiang) were sampled. In the results and discussion, these investigated running waters are combined, sometimes split, which makes it hard to follow (in the main text and tables). In my opinion, these three "regions" need to be presented in a unified way (always separated or combined, possibly both in each table and figure), and presented more clearly in the text.

Response: We provided both separated and combined data in Tables and Figures, please refer to Figs 2 and 3, as well as Figs. S2 and S3. In addition, we also clearly stated in the "Method" part.

"Spatial differences (Daning, Qijiang and entire tributaries of TGR region) were tested using the nonparametric Mann Whitney U-test. Multivariate statistics, such as correlation and stepwise multiple linear regression, were performed for the models of k600 using potential physical parameters of wind speed, water depth, and current velocity from separated data and combined data (Alin et al., 2011)."

(3) One of the main messages is the presentation of transport coefficients in a sub-tropical monsoonal climate zone, which is interesting, but I can imagine that there is a large difference in the wet and dry season. However, all the measurements were done in the wet season. I think this issue should be clearly acknowledged and discussed.

Response: We agreed your opinion and addressed this issue in an additional section 4.4. In this section, effects and uncertainty of sampling seasonality on errors of annual CO2 flux estimation were included.

"Sampling seasonality considerably regulated riverine pCO2 and gas transfer velocity and thus water-air interface CO2 evasion rate (Li et al., 2012; Ran et al., 2015). We sampled waters in the rainy season due to that it showed wider range of flow velocity and thus rainy season covered the k levels in the whole hydrological season. Rainy season generally had higher current velocity and thus higher gas transfer velocity (Ran et al., 2015), while aquatic pCO2 was variable with seasonality. We recently reported that riverine pCO2 in the wet season was 81% the level in the dry season (Li et al., 2018), and prior study on the Yellow River reported that k level in the wet season was 1.8-fold higher than in the dry season (Ran et al., 2015), while another study on the Wuding River demonstrated that k level in the wet season was 83%-130% of that in the dry season (Ran et al., 2017). Thus, we acknowledged a certain amount of errors on the annual flux estimation from one-time sampling campaign during the wet season in the TGR area, while this uncertainty could not be important because that the diluted pCO2 could alleviate the potentially increased k level in the wet season."

(4) There are two technical issues: (i) The measurements with the floating chambers are poorly described. The only information Li et al. provide is that the floating chambers were "deployed". If the flux measurements are done in an anchored or free floating manner is critical (see e.g. Lorke, A., Bodmer, P., Noss, C., Alshboul, Z., Koschorreck, M., Somlai-Haase, C., Bastviken, D., Flury, S., McGinnis, D. F., Maeck, A., Müller, D., and Premke, K.: Technical note: drifting versus anchored flux chambers for measuring greenhouse gas emissions from running waters, Biogeosciences, 12, 7013-7024, https://doi.org/10.5194/bg-12-7013-2015, 2015.). Hence, this issue needs to be addressed clearly. (ii) It seems that all the flux and pCO2 measurements were done distributed during the day. The fact that there is a diurnal cycle of CO2 was not considered (see e.g. Pascal Bodmer, Marlen Heinz, Martin Pusch, Gabriel Singer, Katrin Premke, Carbon dynamics and their link to dissolved organic matter quality across contrasting stream ecosystems, Science of The Total Environment, Volume 553, 2016, Pages 574-586, ISSN 0048-9697, https://doi.org/10.1016/j.scitotenv.2016.02.095.), and values directly compared. This issue should at least be discussed.

Response: A new section of "4.4. Uncertainty assessment of pCO2 and flux-derived k values" was added. We assessed systematic errors and random errors of measurements, drifting chamber and anchored chambers, sampling time. We carefully read the two article and the two citations were included. In addition, the following text was added in the Method section.

"Similar to other studies (Alin et al., 2011), sampling and flux measurements in the day would tend to underestimate CO2 evasion rate (Bodmer et al., 2016)."

(5) Developing models to estimate transport coefficients is meaningful, but the process of the model development is poorly described. Additionally, which data were used for the models, and which not is confusing to me (goes along with my comment (2) above).

Response: The issue was addressed as follows.

Water samples from a total of 115 sites were collected. Floating chambers with replicates were deployed in 101 sites (32 sampling sites in Daning, 37 sites in TGR river networks and 32 sites in Qijiang). The sampling period covered spring and summer season, our sampling points are reasonable considering a water area of 433 km2. For example, 16 sites were collected for Yangtze system to examine hydrological and geomorphological controls on pCO2 (Liu et al., 2017), and 17 sites for dynamic biogeochemical controls on riverine pCO2 in the Yangtze basin (Liu et al., 2016). Similar to other studies, sampling and flux measurements in the day would tend to underestimate CO2 evasion rate (Bodmer et al., 2016).

Prior to statistical analysis, we excluded k600 data for samples with the air-water pCO2 gradient <110 $\mu$atm, since the error in the k600 calculations drastically enhances when âŰşpCO2 approaches zero (Borges et al., 2004; Alin et al., 2011), and datasets with âŰşpCO2 >110 $\mu$atm provide an error of <10% on k600 computation (please refer to the Fig. 1 in the bottom). Thus, we discarded the samples (36.7% of sampling points with flux measurements) with âŰşpCO2 <110 $\mu$atm for k600 model development, while for the flux estimations from diffusive model and floating chambers, all samples were included.

Multivariate statistics, such as correlation and stepwise multiple linear regression, were performed for the models of k600 using potential physical parameters of wind speed, water depth, and current velocity as the independent variables from both separated data and combined data (Alin et al., 2011). k models were obtained by water depth using data from the TGR rivers, while by flow velocity in the Qijiang.

We also highlighted separated or combined data used in the Tables and Figures legends.

(6) From what I see in these data, there are several running waters undersaturated with respect to CO2 (Fig. S2 and Fig. 1), and hence a sink of CO2. This aspect is totally neglected and the investigated running waters are generalized as CO2 sources to the atmosphere. In my opinion, this aspect of influxes of CO2 is very valuable and should be properly discussed.

Response: We addressed this issue by revision in the parts of "Result" and "Discussion". Firstly, we revised in the "Result" section as follows. "pCO2 varied between 50 and 4830 $\mu$atm with mean of 846 $\pm$ 819 $\mu$atm (Table 1). There were 28.7% of samples that had pCO2 levels lower than 410 $\mu$atm, while the studied rivers were overall supersaturated with reference to atmospheric CO2 and act as a source for the atmospheric CO2."

Secondly, the under-saturated pCO2 levels were further examined in the "Discussion" part.

"The calculated pCO2 levels were within the published range, but towards the lower-end of published concentrations compiled elsewhere (Cole and Caraco, 2001; Li et al., 2013). The total mean pCO2 (846 ± 819 $\mu$atm) in the TGR, Danning and Qijiang sampled was lower than one third of global river's average (3220 $\mu$atm) (Cole and Caraco, 2001). The lower pCO2 than most of the world's river systems, particularly the under-saturated values, demonstrated that heterotrophic respiration of terrestrially derived DOC was not significant. Compared with high alkalinity, the limited delivery DOC particularly in the Daning and Qijiang river systems (Figs. S2 and S3) also indicated that in-stream respiration was limited. These two river systems are characterized by karst terrain and underlain by carbonate rock, where photosynthetic uptake of dissolved CO2 and carbonate minerals dissolution considerably regulated aquatic pCO2 (Zhang et al., 2017)."

(7) As far as I see, there is some arbitrariness regarding data handling/processing. The cut-off at 110 $\mu$atm (line 198) for the air-water CO2 gradient for k600 calculations, as well as "When several extremely values are removed: : :" (line 303), needs to be described/demonstrated/justified much more clear.

Response: We provided more details on this concern. Prior to statistical analysis, we excluded k600 data for samples with the air-water pCO2 gradient <110 $\mu$atm, since the error in the k600 calculations drastically enhances when âŰşpCO2 approaches zero (Borges et al., 2004; Alin et al., 2011), and datasets with âŰşpCO2 <110 $\mu$atm provide an error of <10% on k600 computation" (see the Fig. 1 in the bottom)

Regarding ""When several extremely values are removed: : :" (L 303) for revised model, we supplied data including extremely data and excluding extremely data in Fig.

(the original Fig. 3).

(8) $CO_2$ fluxes were measured, while $pCO_2$ and transport coefficients were calculated. This should be clearly stated throughout the manuscript to be transparent.

Response: We highlighted this in the section of "Method" (see section 2.4), and changed "measured k or observed k to flux-derived k or derived k or calculated k".

(9) I am not a native English speaker, but I think that the manuscript should be revised for the English language. (see exemplarily in the specific comments and technical corrections below) I think this is a valuable study, but the combination of the points mentioned above make the manuscript hard to follow and the conclusions and main messages drawn in the current state of the manuscript too general. In a revised version, the study would get more shaped, more detailed and informative, and the conclusions and main messages can be specified and more related to the investigated region.

Response: We edited the English and revised the Ms based on comments and suggestions.

Specific comments Abstract: Line 20: Indicate how many river networks (see general comment (2))

Response: 60 rivers

Line 24: As far as I understood not when all data were included. Please be more specific here.

Response: corrected and provided details in "Method" section.

Lines 30 – 33: This is not really new. Maybe you can specify this statement for the investigated region?

Response: Addressed. The sentence was revised as follows.

"We concluded that simple parameterization of k as a function of morphological characteristics was site specific and hence highly variable in river systems of the upper Yangtze River. k models should be developed for stream studies to evaluate the contribution of these regions to atmospheric CO2."

Introduction: Line 41: Bastviken et al., 2011 totally focuses on CH4. I suggest replacing this reference with a more suitable one.

Response: Bastviken et al., 2011 was replaced by "Raymond et al., 2013; Butman and Raymond, 2011"

Line 42: But you did not present "new accurate measurement techniques" in your study, what are your reasons to mention this in the introduction?

Response: "new" as removed.

Lines 50 – 58: This equation is pretty standard knowledge and can just be described in words here. The equation can be moved to the methods.

Response: Yes, it is very standard knowledge, while the text would be more shape, and easily to follow with equations here.

Line 63: The standardized transport coefficient (k600) should be explained here.

Response: k600 (the standardized transfer coefficient at a temperature of 20 0C)

Line 80: You set the scene of seasonal precipitation, but in the study, you only measure in the wet season. This is contradictory. This issue should be discussed.

Response: We changed "concentrated seasonal precipitation" to "hydrological seasonality"

Lines 84-89: Kind of repetition and partially contradictory to the text in lines 43-49.

Response: The topic is different. The former focuses on flux determination, while the latter talked about k measurement method. We clarified the main text.

Lines 92-99: The relevance of the study (first time in this region, etc.), the objectives and how these objectives were approached should be written more clearly. At this point, the input parameters for the model development is totally unclear.

Response: We re-wrote this part as follows, we also provide details for k models.

To contribute to this debate, extensive investigation was firstly accomplished for determination of k in rivers and streams of the upper Yangtze using FC method. Models of k were further developed using hydraulic properties from flux measurements and TBL model.

Our recent study preliminarily investigated pCO2 and air – water CO2 areal flux as well as their controls from fluvial networks in the Three Gorges Reservoir (TGR) area (Li et al., 2018). The past study was based on two field works, and the diffusive models from other continents were used. In this study, we attempted to derive k levels and develop the gas transfer model in this area (mountainous streams and small rivers) for more accurate quantification of CO2 areal flux, and also to serve for the fluvial networks in the Yangtze River or others with similar hydrology and geomorphology. Moreover, we did detailed field campaigns in the two contrasting rivers Daning and Qijiang for models (Fig. 1). The study thus clearly stated distinct differences than the previous study (Li et al., 2018) by the new contributions of specific objectives and data supplements, as well as wider significance.

Our new contributions to the literature include (1) determination and controls of k levels for small rivers and streams in subtropical areas of China, and (2) new k models developed using hydraulic parameters in the subtropical mountainous river networks.

Materials and methods: Line 105: In my opinion, Figure S1 should go to the main text. There are no sampling points for Daning and Qijiang, which is confusing to me.

Response: We supplied the map in the main text as Fig. 1 (see Fig. 2 in the bottom).

Lines 105-109: Please add a reference for this statement.

Response: "Li et al., 2018" was cited here.

Lines 110-118: Please see my general comment (2): Please restructure this, make clear how many running waters were sampled where, the size of the sampled running waters (Strahler stream order is fine), and why in these three regions. Otherwise, it is hard to follow your storyline.

Response: We provided details in the section 2.2.

Water samples from a total of 115 sites were collected. Floating chambers with replicates were deployed in 101 sites (32 sampling sites in Daning, 37 sites in TGR river networks and 32 sites in Qijiang). The sampling period covered spring and summer season, our sampling points are reasonable considering a water area of 433 km2. For example, 16 sites were collected for Yangtze system to examine hydrological and geomorphological controls on pCO2 (Liu et al., 2017), and 17 sites for dynamic biogeochemical controls on riverine pCO2 in the Yangtze basin (Liu et al., 2016). Similar to other studies, sampling and flux measurements in the day would tend to underestimate CO2 evasion rate (Bodmer et al., 2016). In our sampling points, all measured fluxes were retained since the floating chambers yielded linearly increasing CO2 against time following manufacturer' specification.

Prior to statistical analysis, we excluded k600 data for samples with the air-water pCO2 gradient <110 $\mu$atm, since the error in the k600 calculations drastically enhances when âŰşpCO2 approaches zero (Borges et al., 2004; Alin et al., 2011), and datasets with âŰşpCO2 >110 $\mu$atm provide an error of <10% on k600 computation. Thus, we discarded the samples (36.7% of sampling points with flux measurements) with âŰşpCO2 <110 $\mu$atm for k600 model development, while for the flux estimations from diffusive model and floating chambers, all samples were included.

Line 141-142: What is "PP"?

Response: EGM-4 (Environmental Gas Monitor; PP SYSTEMS Corporation, USA)

**BGD**
[Figure]

Line 148: I don't really understand what you mean by this sentence, please revise.

Response: Changed to "All the solvents and reagents used in experiments were of analytical - reagent grade"

Lines 155-156: This sentence is confusing to me, please revise.

Response: Changed "The relationship was yielded when z=1 (U10=1.208×U1). " to "U10=1.208×U1 as we measured the wind speed at a height of 1 m (U1). "

Line 158: Do you mean CO2SYS? If yes, please add the corresponding reference.

Response: "Lewis et al., 1998" was cited.

Lewis, E., Wallace, D., Allison, L.J., 1998. Program developed for CO2 system calculations.; Brookhaven National Lab., Dept. of Applied Science, Upton, NY (United States); Oak Ridge National Lab., Carbon Dioxide Information Analysis Center, TN (United States), p. Medium: ED; Size: 40 p.

Line 167: What was the brand of the tubing?

Response: Changed to "CO2 impermeable rubber-polymer tubing"

Line 170: What is DC?

Response: "DC" was removed.

Line 173: Please see my general comment (4) (i)

Response: Uncertainty of chambers was discussed in the additional section 4.4. In our study, both drifting chamber and anchored chambers were used, which is dependent on in situ current velocity. The following text was added.

"In sampling sites with low and favorable flow conditions (Fig. S1), freely drifting chambers (DC) were executed, while sampling sites in rivers and streams with higher flow velocity were conducted with anchored chambers (AC) (Ran et al., 2017). AC would create overestimation of CO2 emissions in our studied region (Lorke et al., 2015)."

Line 177: The units are confusing to me. Why is there two times pressure and temperature? Please double check if the units match up in the end, to me they do not.

Response: We have carefully checked and revised.

Line 187: Please be more specific: k was calculated by reorganizing Eq (1)

Response: Revised.

Line 192: Sc to the power of 0.5? This seems weird. What do you mean here?

Response: Corrected.

Line 198: Please justify the cut-off at 110 _atm. Maybe add a figure to the supplementary material.

Response: Revised. Please refer to general comment (7).

Line 203: I read about water depth and current velocity the first time here. These measurements need to be described before.

Response: Measurements of water depth and current velocity were added in the part of "Methods".

Line 213: The pH is quite high. This in combination with influxes of $CO_2$ requires at least a short discussion about chemical enhancement.

Response: We revised the text in the section of "Result" (see the first paragraph below), and added text (see the second paragraph below) in the Discussion section (4.1).

"pH varied from 7.47 to 8.76 with exceptions of two quite high values of 9.38 and 8.87 (mean: $8.39 \pm 0.29$ from total dataset). Much lower pH was observed in TGR rivers ($8.21 \pm 0.33$) (Table 1; $p < 0.05$; Fig. S2)."

"Higher pH levels were observed in Daning and Qijiang ($p < 0.05$ by ANOVA), where more carbonate rock exists that are characterized by karst terrain. Our pH range was comparable to the recent study on the karst river in China (Zhang et al., 2017). Quite
high values (i.e., 9.38 and 8.87) were recorded in the investigated sites, where chemical enhancement would increase the influx of atmospheric CO2 to alkaline waters (Wanninkhof and Knox, 1996), while 1.7% of sampling sites that were strongly affected by chemical enhancement were not significant on a regional scale. This chemical enhancement of CO2 influx was also reported to be limited in high-pH rivers (Zhang et al., 2017)."

Wanninkhof, R., Knox, M., 1996. Chemical enhancement of CO2 exchange in natural waters. Limnology and Oceanography 41, 689-697.

Line 214: Please see my general comment (6)

Response: Addressed. Please refer to general comment (6)

Lines 218-222: This paragraph belongs to the discussion section.

Response: We moved this to Discussion (see 4.1)

Lines 223-227: This paragraph should be revised because it is not very clearly written. Please add the p-values to the text in case of significances.

Response: We re-wrote this part.

"The much higher concentrations of dissolved organic carbon (DOC) and dissolved nutrients (DTN and DTP) were observed in the TGR rivers ($p<0.01$ by ANOVA; Fig. S3). In comparison to Daning, Qijiang showed significantly higher concentrations of DOC and DTN ($p<0.05$), and much lower TDP concentration ($p<0.05$; Fig. S3)."

Lines 235-237: What is the meaning of this ratio? Please add a few words what the reader can get from this information.

Response: Removed.

Line 242: These models and how you developed them should be explained better (in the method section).

Response: We provided details in the section of "Method".

Lines 246-248: I do not understand this sentence. What do you mean by "binned"?

Response: Changed to "combined"

Discussion: Lines 270-274: How does this paragraph support the discussion of your study?

Response: These texts could support the discussion on relationship between spatial change in k values and physical characteristics (i.e., current velocity, slope and the water depth) of three river systems.

Spatial differences of k values and their controls of physical characteristics of current velocity, slope and the water depth were discussed for the three rivers systems (please refer to section 4.2).

"This could substantiate the higher k600 levels and spatial changes in k600 values of our three river systems. For instance, similar to other turbulent rivers in China (Ran et al., 2015; Ran et al., 2017), high k600 values in the TGR, Daning and Qjiang rivers were due to mountainous terrain catchment, high current velocity (10 – 150 cm/s) (Fig. 4b), bottom roughness, and shallow water depth (10 - 150 cm) (Fig. 4a). It has been suggested that shallow water enhances bottom shear, and the resultant turbulence increases k values (Alin et al., 2011; Raymond et al., 2012). These physical controls are highly variable across environmental types (Figs. 4a and 4b), hence, k values are expected to vary widely (Fig. 3). The k600 values in the TGR rivers showed wider range (1-177 cm/h; Fig. 3; Table S1), spanning more than 2 orders of magnitude across the region, and it is consistent with the considerable variability in the physical processes on water turbulence across environmental settings. Similar broad range of k600 levels was also observed in the China's Yellow basin (ca. 0-123 cm/h) (Ran et al., 2015; Ran et al., 2017).

Contrary to our expectations, no significant relationship was observed between k600

and water depth, and current velocity using the entire data in the three (TGR, Danning and Qjiang) river systems (Fig. S4). There were not statistically significant relationships between k600 and wind speed using separated data or combined data, and it is consistent with earlier studies (Alin et al., 2011; Raymond et al., 2012). Flow velocity showed linear relation with k600, and the extremely high value of k600 was observed during the periods of higher flow velocity (Fig. S4a) using combined data. Similar trend was also observed between water depth and k600 values (Fig. S4b). The lack of strong correlation between k600 and physical factors are probably due to combined effect of both flow velocity and water depth, as well as large diversity of channel morphology, both across and within river networks in the entire catchment (60, 000 km2). This is further collaborated by weak correlations between k600 and flow velocity in the TGR rivers (Fig. 4), where one or two samples were taken for a large scale examination. k600 as a function of water depth was obtained in the TGR rivers, but it explained only 30% of the variance in k600. However, model using data from Qijiang could explain 68% of the variance in k600 (Fig. 4b), and it was in line with general theory. Nonetheless, k600 from our flow velocity based model (Fig. 4b) was potentially largely overestimated with consideration of other measurements (Alin et al., 2015; Ran et al., 2015; Ran et al., 2017). When several extremely values were removed, k600 (cm/h) was parameterized as follows (k600 = 62.879FV + 6.8357, $R^2$ = 0.52, p=0.019, FV-flow velocity with a unit of m/s), and this revised model was in good agreement with the model in the river networks of the Yellow River (Ran et al., 2017), but much lower than the model developed in the Yangtze system (Liu et al., 2017) (Fig. 4c). This was reasonable because of k600 values in the Yangtze system were from large rivers with higher turbulence than Yellow and our studied rivers. Furthermore, the determined k600 using FCs was, on average, consistent with the revised model (Table 2). These differences in relationship between spatial changes in k600 values and physical characteristics further corroborated heterogeneity of channel geomorphology and hydraulic conditions across the investigated rivers."

Lines 286-288: No significances: : : But still, you developed the models considering all data? This is not at all clear to me. Did you split/separate the data set for the models? This is not clear in Table 2. Please see my general comment (2). I think these data/models are valuable, but at the moment they seem arbitrary and should be better explained. This would help to give them more weight.

Response: We now provided details of model developing in the "Method" (see the response to general comment (2), (5) and (7). We also clearly stated the separated or combined data for models in the captions of Tables and Figs.

Lines 286-309: I see a lot of results here, which are presented in the discussion for the first time. The part presenting pure results should be moved to the results section.

Response: Revised

Line 303: Please justify the removal of "extremely values". Maybe add a figure to the supplementary material. If there is no objective criteria and justification, I do not see why data should be removed.

Response: We provided details in the Method and also supplied figure with or without extremely values in the main text (Fig. 4).

Lines 327-333: Why discussing k values and not k600 values? I think this needs to be unified/consistent throughout the manuscript.

Response: We unified to k600 though k600 and k were widely discussed in previous studies.

Conclusion: Lines 358-360: Very general, but actually, the regions had to be separated/ split, no?

Response: The words were removed.

Lines 368-369: I think you should focus the conclusion on the investigated region.

Response: "in the river systems of the upper Yangtze River" was added.

Tables: Table 3: b) Why not presenting k600 values here which can be directly compared with other studies?

Response: We unified to "k600"

Figures: Fig. S1: Was always everything sampled at each point? "Samples" should be replaced by "sampling point"

Response: We moved it to the main text as Fig. 1, and "Samples" was changed to "Sampling point".

Fig. S4: There is no reference to this figure in the main text.

Response: Fig. S4 was cited in the main text.

Technical corrections Line 22: Delete "were" Line 44: add "by" or "via" before "floating chambers" Line 59: Replace "precisely" with "well" Line 127: "consisted" instead of "consists" Line 139: "pH sonde" "was" instead of "is" Line 225: "Daning" instead of "Danning"

Response: All the typos were corrected.

Alin, S.R., Rasera, M., Salimon, C.I., Richey, J.E., Holtgrieve, G.W., Krusche, A.V., Snidvongs, A., 2011. Physical controls on carbon dioxide transfer velocity and flux in low-gradient river systems and implications for regional carbon budgets. Journal of Geophysical Research-Biogeosciences 116. Bodmer, P., Heinz, M., Pusch, M., Singer, G., Premke, K., 2016. Carbon dynamics and their link to dissolved organic matter quality across contrasting stream ecosystems. Science of the Total Environment 553, 574-586. Borges, A.V., Delille, B., Schiettecatte, L.S., Gazeau, F., Abril, G., Frankignoulle, M., 2004. Gas transfer velocities of CO2 in three European estuaries (Randers Fjord, Scheldt, and Thames). Limnology and Oceanography 49, 1630-1641. Cole, J.J., Caraco, N.F., 2001. Carbon in catchments: connecting terrestrial carbon losses with aquatic metabolism. Marine and Freshwater Research 52, 101-110. Li, S., Ni, M., Mao, R., Bush, R.T., 2018. Riverine CO2 supersaturation and outgassing in a subtropical monsoonal mountainous area (Three Gorges Reservoir Region) of China. Journal of Hydrology 558, 460-469. Li, S.Y., Lu, X.X., Bush, R.T., 2013. CO2 partial pressure and CO2 emission in the Lower Mekong River. Journal of Hydrology 504, 40-56. Liu, S., Lu, X.X., Xia, X., Yang, X., Ran, L., 2017. Hydrological and geomorphological control on CO2 outgassing from low-gradient large rivers: An example of the Yangtze River system. Journal of Hydrology 550, 26-41. Liu, S., Lu, X.X., Xia, X., Zhang, S., Ran, L., Yang, X., Liu, T., 2016. Dynamic biogeochemical controls on river pCO(2) and recent changes under aggravating river impoundment: An example of the subtropical Yangtze River. Global Biogeochemical Cycles 30, 880-897. Ran, L., Li, L., Tian, M., Yang, X., Yu, R., Zhao, J., Wang, L., Lu, X.X., 2017. Riverine CO2 emissions in the Wuding River catchment on the Loess Plateau: Environmental controls and dam impoundment impact. Journal of Geophysical Research-Biogeosciences 122, 1439-1455. Wanninkhof, R., Knox, M., 1996. Chemical enhancement of CO2 exchange in natural waters. Limnology and Oceanography 41, 689-697. Zhang, T., Li, J., Pu, J., Martin, J.B., Khadka, M.B., Wu, F., Li, L., Jiang, F., Huang, S., Yuan, D., 2017. River sequesters atmospheric carbon and limits the CO2 degassing in karst area, southwest China. Science of The Total Environment 609, 92-101.

We also provided tracked main text for you reviewing

Please also note the supplement to this comment:
https://www.biogeosciences-discuss.net/bg-2018-227/bg-2018-227-AC3-supplement.pdf

———————————————

[Figure]

[Figure]

Fig. 1. Theoretical error (±%) on the computation of the gas transfer velocity of $CO_2$ ($k600$) as a function of the air–water gradient of $CO_2$ ($\Delta pCO_2$ in ppm), assuming a constant uncertainty on $\Delta pCO_2$ of ±3% (Borges et al., 2004).

**Fig. 1.**

[revised manuscript text omitted]

TDP concentration (p<0.05; Fig. S3).

**3.2. CO$_2$ flux using floating chambers**

The calculated CO$_2$ areal fluxes were higher in TGR rivers (217.7 $\pm$334.7

mmol/m$^2$/d, n = 35), followed by Daning (122.0 $\pm$239.4 mmol/m$^2$/d, n = 28) and

Qijiang rivers (50.3 $\pm$177.2 mmol/m$^2$/d, n = 32) (Fig. 2). The higher CO$_2$ evasion from the TGR rivers is consistent with high riverine $p$CO$_2$ levels. The mean CO$_2$

emission rate was 133.1 $\pm$269.1 mmol/m$^2$/d (n = 95) in all three rivers sampled. The mean CO$_2$ flux differed significantly between TGR rivers and Qijiang (Fig. 2).

**3.3. k levels**

  A total of 64 data were used (10 for Daning River, 33 for TGR rivers and 21 for Qijiang River) to develop k model after removal of samples with $\Delta p$CO$_2$ less than 110 µatm (Table 2). No significant variability in k$_{600}$ values were observed among the three rivers sampled (Fig. 3). The mean k$_{600}$ (unit in cm/h) was relatively higher in Qijiang (60.2 ±78.9), followed by Daning (50.2 ±20.1) and TGR rivers (40.4 ±37.6), while the median k$_{600}$ (unit in cm/h) was higher in Daning (50.5), followed by TGR rivers (30.0) and Qijiang (25.8) (Fig. 3; Table S1). Combined  k$_{600}$ data were averaged to 48.4 ±53.2 cm/h (95% CI: 35.1-61.7), and it is 1.5-fold higher than the median value (32.2 cm/h) (Fig. 3).

**4. Discussion**

**4.1. Determined k values relative to world rivers**

We derived first-time the k values in the subtropical streams and small rivers. Our determined k$_{600}$ levels with a 95% CI of 35.1 to 61.7 (mean: 48.4) cm/h is compared well with a compilation of data for streams and small rivers (e.g., 3-70 cm/h) (Raymond *et al.*, 2012). Our determined k$_{600}$ values are greater than the global rivers' average (8 - 33 cm/h) (Butman and Raymond, 2011; Raymond *et al.*, 2013), and much higher than mean for tropical and temperate large rivers (5-31 cm/h) (Alin *et al.*, 2011). These studies evidences that k$_{600}$ values are highly variable in streams and small rivers (Alin *et al.*, 2011; Ran *et al.*, 2015). Though the mean $k_{600}$ in the TGR, Daning and Qijiang is higher than global mean, however, it is consistent with $k_{600}$ values in the main stream and river networks of the turbulent Yellow River (42 ± 17 cm/h) (Ran *et al.*, 2015), and Yangtze (38 ±40 cm/h) (Liu *et al.*, 2017) (Table S2).

The calculated $pCO_2$ levels were within the published range, but towards the lower-end of published concentrations compiled elsewhere (Cole and Caraco, 2001; Li *et al.*, 2013). The total mean $pCO_2$ (846 ±819 µatm) in the TGR, Danning and Qijiang sampled was lower than one third of global river's average (3220 µatm) (Cole and Caraco, 2001). The lower $pCO_2$ than most of the world's river systems, particularly the under-saturated values, demonstrated that heterotrophic respiration of terrestrially derived DOC was not significant. Compared with high alkalinity, the limited delivery DOC particularly in the Daning and Qijiang river systems (Figs. S2 and S3) also indicated that in-stream respiration was limited. These two river systems are characterized by karst terrain and underlain by carbonate rock, where photosynthetic uptake of dissolved $CO_2$ and carbonate minerals dissolution considerably regulated aquatic $pCO_2$ (Zhang *et al.*, 2017).

Higher pH levels were observed in Daning and Qijiang river systems (p<0.05 by Mann-Whitney Rank Sum Test), where more carbonate rock exists that are characterized by karst terrain. Our pH range was comparable to the recent study on the karst river in China (Zhang *et al.*, 2017). Quite high values (i.e., 9.38 and 8.87) were recorded in some investigated sites, where chemical enhancement would increase the influx of atmospheric $CO_2$ to alkaline waters (Wanninkhof and Knox,

1996), while 1.7% of sampling sites that were strongly affected by chemical enhancement were not significant on a regional scale. This chemical enhancement of $CO_2$ influx 
[revised manuscript text omitted]

---

## Author Response (AR1)

**Response to Associate Editor Decision: Reconsider after major revisions** (03 Oct 2018) by David Butman

Comments to the Author:

*1. After reviewing all of the interactive comments on bg-2018-227, it is recommended that the authors return to the calculated annual emissions and possibly remove this from the discussion. As outlined from all comments, the appropriate data sets may not exists to provide a meaningful annual emission estimate given the restricted sampling period. The discussion could be limited to monsoonal periods only. If the authors would like to keep this in, it is suggested that they develop a stronger presentation on what changes across seasons related to gas transfer and concentrations.*

Response: Thank you for your helpful comment. According to your suggestion, annual emission estimate was deleted and $CO_2$ evasion upscaling was conducted during the monsoonal period in main text as follows.

"We used our measured $CO_2$ emission rates by FCs for upscaling flux estimates during monsoonal period given the sampling in this period and it was found to be 0.70 $TgCO_2$ for all rivers sampled in our study (Table 3a). The estimated emission in the monsoonal period was close to that of the revised model (0.71 ± 0.66 (95% confidence interval: 0.46-0.94) Tg $CO_2$), and using the determined k average, i.e., 0.69 ± 0.65 (95% confidence interval: 0.45-0.93) Tg $CO_2$, but slightly higher than the estimation using water-depth based model (0.54 ± 0.51 Tg $CO_2$) and Alin's model (0.53 ± 0.50 Tg $CO_2$) (Table 3b). The higher emission, i.e., 1.66 ± 1.55 (1.08-2.23) Tg $CO_2$,in the monsoonal period only using flow velocity based model may be over-estimated when compared to other models, flux from determined k (Table 3b) and previous annual estimates, i.e., our earlier annual evasion of 0.64-2.33 Tg $CO_2$/y using TBL on the TGR river networks (Li *et al.*, 2018). Moreover, our estimated $CO_2$ emission in the monsoonal period also suggests that $CO_2$ annual emissions from rivers and streams in this area were previously underestimated, i.e., 0.03 Tg $CO_2$/y (Li *et al.*, 2017) and 0.37-0.44 Tg $CO_2$/y (Yang *et al.*, 2013) as the former used TBL model with a lower k level, and the latter employed floating chambers, but they both sampled very limited tributaries (i.e., 2-3 rivers). Therefore, measurements of k must be made mandatory along with $pCO_2$ measurement in the river and stream studies." (Last paragraph in section 4.3)

Further, we strongly discussed the monsoonal sampling effects on gas transfer and $pCO_2$ concentrations, as well as annual evasion in the SOM. This could help do the comparison with other studies.

**Seasonal changes related to gas transfer and concentrations**

We used our measured $CO_2$ emission rates by FCs for upscaling flux estimates and it was found to be 1.39 $TgCO_2$/y for all rivers sampled (Table S3a). The estimated emission was close to that of the revised model (1.40 ± 1.31 (95% confidence interval: 0.91-1.87) Tg $CO_2$/y), and using the determined k average, i.e., 1.37 ± 1.28 (95% confidence interval: 0.89-1.84) Tg $CO_2$/y, but slightly higher than the estimation using water-depth based model (1.08 ± 1.01 Tg $CO_2$/y) and Alin's model (1.06 ± 1.00 Tg $CO_2$/y) (Table S3b). The estimate was within the range of our earlier work using TBL on the TGR river networks (0.64-2.33 Tg $CO_2$/y) (Li *et al.*, 2018). The higher emission, i.e., 3.29 ± 3.08 (2.15-4.43) Tg $CO_2$/y, using flow velocity based model may be over-estimated (Table 3b). Therefore, this study suggests that $CO_2$ emissions from rivers and streams in this area may be underestimated, i.e., 0.03 Tg $CO_2$/y (Li *et al.*, 2017) and 0.37-0.44 Tg $CO_2$/y (Yang *et al.*, 2013) as the former used TBL model with a lower k level, and the latter employed floating chambers, but they both sampled very limited tributaries (i.e., 2-3 rivers). Therefore, measurements of k must be made mandatory along with $pCO_2$ measurement in the river and stream studies.

As our sampling was limited to monsoonal periods only, which could not provide a meaningful annual emission estimate given the restricted sampling period. Thus, we developed a stronger discussion on what changes across seasons related to gas transfer and $pCO_2$ concentrations. As outlined in the main text, riverine $pCO_2$ in the monsoonal season in this region was 81% the level in the dry season, and current velocity was 1.7-fold higher in monsoonal season (Li *et al.*, 2018), thus $k_{600}$ was 1.6-fold higher in the monsoonal period based our model. This could be defensible due to that prior study on the Yellow River reported that k600 level in the wet season was 1.8-fold higher than in the dry season (Ran *et al.*, 2015), another study on the Wuding River demonstrated that k level in the wet season was 83%-130% of that in the dry season (Ran *et al.*, 2017). Moreover, a factor of 1.4 for water area was designated based on other monsoonal rivers. Then annual emission could be estimated at 1.59 Tg $CO_2$/y, slightly higher than the estimation using the data in the monsoonal period only.

**Table S3.** $CO_2$ emission from total rivers sampled in the study.
(a) Upscaling using $CO_2$ areal flux by FC.

| | Catchment Area $km^2$ | Water surface $km^2$ | $CO_2$ areal flux $mmol/m^2/d$ | $CO_2$ emission Tg $CO_2$/y |
|---|---|---|---|---|
| Daning | 4200 | 21.42 | 122.0 ±239.4 | 0.042 |
| Qijiang | 4400 | 30.8 | 50.3 ±177.2 | 0.025 |
| TGR river | 50000 | 377.78 | 217.7 ±334.7 | 1.321 |
| Total | | | | 1.39 |

(b) Upscaling using determined $k_{600}$ average and models (whole dataset are used here).

| | From determined $k_{600}$ mean | Flow velocity-based model (Fig. 4b) (numbers in bracket is from the revised model; Fig. 4c) | Water depth-based model (Fig. 4a) | Alin's model |
|---|---|---|---|---|

| | | | | |
|---|---|---|---|---|
| Mean | 1.37 | 3.29 (1.40) | 1.08 | 1.06 |
| S.D. | 1.28 | 3.08 (1.31) | 1.01 | 1.00 |
| 95% CI for Mean | Lower Bound | 0.89 | 2.15 (0.91) | 0.71 | 0.69 |
| | Upper Bound | 1.84 | 4.43 (1.87) | 1.46 | 1.43 |

A total water area of approx. 430 $km^2$ for all tributaries (water area is from Landsat ETM+ in 2015).

**Table S4.** Monsoonal sampling effects on annual emission

| | Monsoonal season | Dry season | Monsoonal/Dry |
|---|---|---|---|
| $pCO_2$ ($\mu atm$) | 846 | 1043 | 0.81 |
| $\Delta pCO_2$ ($\mu atm$) | 446 | 643 | 0.69 |
| $k_{600}$ (cm/h) | 48.4 | 31.3 | 1.55 |
| $CO_2$ areal flux (mmol/$m^2$/d) | 196.9 | 183.4 | |
| Water area ($km^2$) | 602 | 430 | |
| Emission (Tg $CO_2$) | 0.96 | 0.63 | |

*2. Furthermore, it is recommended that the authors provide a stronger discussion of the bias that may occur using static chambers in a river environment. There are estimates of this bias in the literature and those values should be referenced. This method is concerning, unless in very small streams with significant turbulence induced from both surface and subsurface features.*

Response: We presented strong discussion (please refer to section 4.4), and the following text was added.

Efforts have been devoted to measurement techniques (comparison of FC, eddy covariance-EC and boundary layer model-BLM) for improving $CO_2$ quantification from rivers because of a notable contribution of inland waters to the global C budget (), which could have a large effect on the magnitude of the terrestrial C sink. Prior studies reported inconsistent trends of $CO_2$ area flux by these methods. For instance, CO2 areal flux from FC was much lower than EC (Podgrajsek et al., 2014), while areal flux from FC was higher than both EC and BLM elsewhere (Erkkilä et al., 2018), however, Schilder et al (2013) demonstrated that areal flux from BLM was 33-320% of in-situ FC measurements. Albeit unsatisfied errors of varied techniques and additional perturbations from FC exist, FC method is currently a simple and preferred measurement for $CO_2$ flux because that choosing a right k value remained a major challenge and others require high workloads (Martinsen et al., 2018).

Recent study further reported fundamental differences in $CO_2$ emission rates between ACs and freely DFs (Lorke *et al.*, 2015), i.e., ACs biased the gas areal flux higher by a factor of 2.0-5.5. However, some studies observed that ACs showed reasonable agreement with other flux measurement techniques (Galfalk *et al.*, 2013), and this straightforward, inexpensive and relatively simple method AC was widely used (Ran *et al.*, 2017). Water-air interface $CO_2$ flux measurements were primarily made using ACs in our studied streams and small rivers because of relatively high current velocity; otherwise, floating chambers will travel far during the measurement period. In addition, inflatable rings were used for sealing the chamber headspace and submergence of ACs was minimal, therefore, our measurements were potentially overestimated but reasonable. We could not test the overestimation of ACs in this study, the modified FCs, i.e., DCs and integration of ACs and DCs, and multi-method comparison study including FCs, ECs and BLM should be conducted for a reliable chamber method.

*3. Furthermore, the inclusion of data that is not relevent to the calculation and interpretation of k and CO2 emissions was identified as distracting. This is in reference to the DOC/TN/TP. As outlined in two of the reviews, either identify why these are included (DOC is useful to know that you are not overestimating Alkalinity from organic acids) or remove from the manuscript.*

Response: The parts of TN and TP were removed.

*4. This effort will provide new knowledge and data from understudied rivers in SE Asia. That alone is a strong contribution. After addressing in detail these and the reviewer comments, this would be suitable for publication.*

Response: Thanks for your very positive comment.

END of Review

**Response to Anonymous Referee #1**

*1. General comments: The paper "Gas transfer velocities of CO2 in subtropical monsoonal*
*climate streams and small rivers" appears to be something of a companion piece to*
*"Riverine CO2 supersaturation and outgassing in a subtropical monsoonal mountainous*
*area (Three Gorges Reservoir Region) of China"*
*https://doi.org/10.1016/j.jhydrol.2018.01.057 published in the Journal of Hydrology. In*
*this current submission, the authors present k calculated from floating chamber flux*
*measurements and using models, and discuss the implications of the differing approaches*
*to k for making regional scale flux estimates. Using chambers to determine*
*CO2 fluxes, the authors then use pCO2 to derive the gas transfer velocity. These*
*flux-derived k values are compared to modeled k values. It is good to see the spatial*
*aspects of the gas transfer velocity addressed. However, I do not feel that there is an*
*adequate consideration of the uncertainty in the estimates/calculations provided.*
*For the flux-derived k values, there is little provided in terms of uncertainty assessments.*

**Response: We thanked the referee for the comment. In our previous article, we studied the $p$CO$_2$ and emission rate as well as their controls from fluvial networks in the TGR area, which is based on two field works in the TGR region, and the diffusive models from other studies were used. In this study, we attempted to derive k levels and develop the gas transfer model in this area (mountainous streams and small rivers) for more accurate quantification of CO$_2$ areal flux, and also to serve for the fluvial networks in the Yangtze River or others with similar hydrology and geomorphology. In addition, we did more detailed field study in the two contrasting rivers Daning and Qijiang for developing models (see the sampling locations map). This study clearly showed original contribution to the current literature and this study is different than the article published in the Journal of Hydrology. We clearly state the new contributions and significances in the last paragraph of the "Introduction" as follows.**

"Our recent study preliminarily investigated $p$CO$_2$ and air – water CO$_2$ areal flux as well as their controls from fluvial networks in the Three Gorges Reservoir (TGR) area (Li *et al.*, 2018). The past study was based on two field works, and the diffusive models from other continents were used. In this study, we attempted to derive k levels and develop the gas transfer model in this area (mountainous streams and small rivers) for more accurate quantification of CO$_2$ areal flux, and also to serve for the fluvial networks in the Yangtze River or others with similar hydrology and geomorphology. Moreover, we did detailed field campaigns in the two contrasting rivers Daning and Qijiang for models (Fig. 1). The study thus clearly stated distinct differences than the previous study (Li *et al.*, 2018) by the new contributions of specific objectives and data supplements, as well as wider significance."

We added a section (4.4.) for "**Uncertainty assessment of $p$CO$_2$ and flux-derived k values**" in the part of "Discussion".

The uncertainty of flux-derived k values mainly stem from air–water gradient of $CO_2$ ($\Delta pCO_2$ in ppm) and flux measurements (Golub et al., 2017; Lorke et al., 2015; Bodmer et al., 2016). Thus we provided uncertainty assessments caused by dominant sources of uncertainty from measurements of aquatic $pCO_2$ and $CO_2$ areal flux since uncertainty of atmospheric $CO_2$ measurement could be neglected.

[revised manuscript text omitted]

*2. pCO2 was not measured, but rather was computed based on pH, alkalinity and temperature. This would have large uncertainties that then propagate into k estimates. Golub et al. (2017, doi: 10.1002/2017JG003794) note that "freshwater researchers must make significant efforts to standardize and reduce errors in pCO2 predictions". I encourage the authors to undertake a more systematic uncertainty analysis for their pCO2 values and propagate this error into uncertainty estimates for k.*

**Response: Thanks for the comment. We reported the quality control such as systematic errors and random errors of pH and alkalinity, water temperature, as well as non-carbonate alkalinity effects. Please refer to the response above.**

*3. Further, the authors here excluded deriving k values for samples that did not have a very large gradient in CO2 across the air-water interface. The authors chose 110 uatm as the threshold for excluding data, but this was presented without any indication of choice of threshold, making it appear rather arbitrary. Given the pH of the rivers sampled and the pCO2 that was at times undersaturated, this appears rather problematic*

*in that it introduces bias that carries through to the regional estimates provided.*

**Response: We addressed this issue as follows.**
  "Prior to statistical analysis, we excluded $k_{600}$ data for samples with the air-water $pCO_2$ gradient <110 μatm, since the error in the $k_{600}$ calculations drastically enhances when $\triangle pCO_2$ approaches zero (Borges et al., 2004), and datasets with $\triangle pCO_2$ >110 μatm provide an error of <10% on $k_{600}$ computation (see the Fig. as follows)"

      The additional section 4.4 **Uncertainty assessment of $pCO_2$ and flux-derived k values** included uncertainty of pH and scaling-up estimation, for example, effects of chemical enhancement for quite high pH values.

[Figure]

Figure Theoretical error (±%) on the computation of the gas transfer velocity of $CO_2$ ($k_{600}$) as a function of the air–water gradient of $CO_2$ ($\triangle pCO_2$ in ppm), assuming a constant uncertainty on $\triangle pCO_2$ of ±3% (Borges et al., 2004).

Borges, A.V., Delille, B., Schiettecatte, L.S., Gazeau, F., Abril, G., Frankignoulle, M., 2004a. Gas transfer velocities of CO2 in three European estuaries (Randers Fjord, Scheldt, and Thames). Limnology & Oceanography 49, 1630-1641.

*4. The authors in this paper refer to their k values as "observed", but these are in fact derived, and so need to have uncertainty better characterized. Upscaling from X floating chamber measurements to a river network draining 58000 km2. How many flux measurements were made with floating chambers is not clearly stated, but it appears to be about 100 all made during summer 2016. Going from summer measurements for 100 points to annual estimates for 58000 km2 also requires some consideration of error propagation and bias. Fluxes were only retained when the floating chambers yielded linearly increasing CO2 against time, which again biases against low flux locations.*

**Response: We have changed "observed, or measured" to "flux-derived or derived", and discussed the uncertainty of k values as mentioned above. We agreed that more samples could improve the $CO_2$ estimates, while our sampling locations were much more or at least comparable to the other publications.**

A total of 115 discrete grab samples were collected (each sample consisted of three replicates). Floating chambers with replicates were deployed in 101 sites (32 sampling sites in Daning, 37 sites in TGR river networks and 32 sites in Qijiang). The sampling period covers spring and summer season, our sampling points are reasonable considering a water area of 433 $km^2$. For example, 16 sites were collected for Yangtze system to examine hydrological and geomorphological controls on $pCO_2$ (i.e., Liu et al., 2017), and 17 sites for dynamic biogeochemical controls on riverine $pCO_2$ in the Yangtze basin (Liu et al., 2016, Global Biogeochemical Cycles).

In our sampling points, all measured fluxes were retained since the floating chambers yielded linearly increasing $CO_2$ against time following manufacturer' specification.

Liu, S., Lu, X.X., Xia, X., Yang, X., Ran, L., 2017. Hydrological and geomorphological control on CO2 outgassing from low-gradient large rivers: An example of the Yangtze River system. Journal of Hydrology 550, 26-41.

*5. Of the attempted flux measurements, what fraction was discarded?*

**Response: We revised as follows.**
"Prior to statistical analysis, we excluded $k_{600}$ data for samples with the air-water $pCO_2$ gradient <110 μatm, since the error in the $k_{600}$ calculations drastically enhances when $\triangle pCO_2$ approaches zero (Borges *et al.*, 2004; Alin *et al.*, 2011), and datasets with $\triangle pCO_2$ >110 μatm provide an error of <10% on $k_{600}$ computation. Thus, we discarded the samples (36.7% of sampling points with flux measurements) with $\triangle pCO_2$ <110 μatm for $k_{600}$ model development, while for the flux estimations from diffusive model and floating chambers, all samples were included."

*6. Finally, a minor point is that the authors state several times that theirs is the first determination of k for subtropical streams and small rivers. I would point the authors towards global syntheses on CO2 evasion as well as individual studies that include k estimates.*

**Response: We agree the comment, and addressed this issue. Several sentences for implication of k determination and comparison with other k studies were added in the part of Discussion. We also re-organised and added sub-headings of "Discussion".**

"4.1. Determined k values relative to world rivers; 4.2. Hydraulic controls of k600; 4.3. Implications for large scale estimation; 4.4. Uncertainty assessment of $pCO_2$ and flux-derived k values"

*Minor comments*

*7. Finally, a minor point is that the authors state several times that theirs is the first determination of k for subtropical streams and small rivers. I would point the authors towards global syntheses on CO2 evasion as well as individual studies that include k*

*estimates.*

**Response: We have presented k levels related to global rivers (please refer to 4.1) and implications of k for large scale estimation (see section 4.3).**

*8. The figure S1 does not show the sample locations within the Daning or Qiijiang basins. These may be the same locations as Figure 1 in Li et al. (2018) Journal of Hydrology doi: 10.1016/j.jhydrol.2018.01.057?*

**Response: The sampling sites and study aims are different than previous study (Please refer to the section of 2.1 and the last paragraph in the "Introduction"). In the revised Ms, we supplied the map of sampling locations in the main text as Fig. 1.**

We added several sentences in the section of "INTRODUCTION" to highlight the differences between our study and previous study, as well as what is advanced by this study (please refer to the first Comment).

*9. There are a number of grammatical issues throughout the paper that the authors should address.*

**Response: We carefully edited English, and also get helps from a native English scientist.**

The additional Table was added to the Table S2 in SOM.

|  | Current velocity m/s | Water depth m/s | Wind speed m/s | $k_{600}$ cm/h | Reference |
|---|---|---|---|---|---|
| Mekong tributary | 0.39±0.28 | 0.9±0.6 | 0.7±0.6 | 23.3±17.3 | Alin et al., 2011 |
| Yellow | 1.8 |  | 1.8 (1.2-2.3) | 42±17 | Ran et al., 2015 |
| Yangtze | 1.2±1.5 |  | 1.2±1.1 | 38±40 | Liu et al., 2017 |
| Mekong stem | 0.92±0.42 |  | 1.8±1.2 | 15±9 | Alin et al., 2011 |

[Figure]

**Fig. 1.** Map of sampling locations of major rivers and streams in the Three Gorges Reservoir region, China (main text).

**Response to Anonymous Referee #2**

*The manuscript reports on transfer velocities of CO2 (K) in streams and small rivers for assessing the gas fluxes. CO2 released from lakes and rivers has been recently recognized as an important component in the global carbon cycle. The accurate estimation of CO2 flux is still challenging primarily due to the difficulty in obtaining an appropriate K value. The topic would be of great interest for the community of scientists working on carbon cycles and can be considered for publication. However, the current version need to revised (see below).*

**Response**: We thanked the referee for the positive comment. We revised the Ms based on the comments and suggestions as follows.

General comments
*1. As emphasized by the authors, the study focuses on the subtropical monsoonal streams and small rivers which are characterized by large seasonal variations in climate and discharge. Hence, the K value in these rivers should also have obvious seasonality. Unfortunately, the samples presented in this study were collected in the rainy season. The K value were calculated based on the one-time sampling campaign, which might result in a certain amount of errors on the annual flux estimation. Regarding this, the uncertainty of the sampling data and the calculations, as well as the reliability of the argument should be sufficiently discussed.*

**Response**: We agreed the comment. Sampling seasonality largely impacted riverine $pCO_2$ and gas transfer velocity and thus water-air interface $CO_2$ evasion rate. In our Ms, we sampled waters in the rainy season due to that it showed wider range of flow velocity and thus rainy season covered the k levels in the whole hydrological season. Rainy season generally had higher current velocity and thus higher gas transfer velocity, while aquatic $pCO_2$ was variable with seasonality. Thus, we estimated CO2 emission in monsoonal period instead of annual emission (please refer to section 4.3).

We added a section of "**4.4. Uncertainty assessment of $pCO_2$ and flux-derived k values**" in the part of "Discussion". In this section, effects and uncertainty of sampling seasonality on errors of annual $CO_2$ flux estimation were included.

Sampling seasonality considerably regulated riverine $pCO_2$ and gas transfer velocity and thus water-air interface $CO_2$ evasion rate (Li *et al.*, 2012; Ran *et al.*, 2015). We sampled waters in wet season (monsoonal period) due to that it showed wider range of flow velocity and thus it covered the $k_{600}$ levels in the whole hydrological season. Wet season generally had higher current velocity and thus higher gas transfer velocity (Ran *et al.*, 2015), while aquatic $pCO_2$ was variable with seasonality. We recently reported that riverine $pCO_2$ in the wet season was 81% the level in the dry season (Li *et al.*, 2018), and prior study on the Yellow River reported that k level in the wet season was 1.8-fold higher than in the dry season (Ran *et al.*, 2015), while another study on the Wuding River demonstrated that k level in the wet season was 83%-130% of that in the dry season (Ran *et al.*, 2017). Thus, we acknowledged a certain amount of errors on the annual flux estimation from sampling campaigns during the wet season in the TGR area, while this uncertainty could not be significant because that the diluted $p\mathrm{CO_2}$ could alleviate the overestimated emission by increased k level in the wet season (stronger discussion please refer to SOM).

*2. In my point of view, the variation in K value of the rivers studied are obvious and need to be discussed. In addition, the spatial difference of K values is only sorted out for the three river systems (Daning, Qijiang and TGR). I would suggest the authors examine the variations of K following the physical characteristics of rivers (such as the current velocity, slope and the water depth) or/and the river orders.*

**Response: Spatial differences of k values were discussed for the three rivers systems (please refer to section 4.2). We discussed the controls of physical characteristics of current velocity, slope and the water depth, while river orders were not extracted.**

"This could substantiate the higher $k_{600}$ levels and spatial changes in $k_{600}$ values of our three river systems. For instance, similar to other turbulent rivers in China (Ran *et al.*, 2015; Ran *et al.*, 2017), high $k_{600}$ values in the TGR, Daning and Qjiang rivers were due to mountainous terrain catchment, high current velocity (10 – 150 cm/s) (Fig. 4b), bottom roughness, and shallow water depth (10 - 150 cm) (Fig. 4a). It has been suggested that shallow water enhances bottom shear, and the resultant turbulence increases k values (Alin *et al.*, 2011; Raymond *et al.*, 2012). These physical controls are highly variable across environmental types (Figs. 4a and 4b), hence, k values are expected to vary widely (Fig. 3). The $k_{600}$ values in the TGR rivers showed wider range (1-177 cm/h; Fig. 3; Table S1), spanning more than 2 orders of magnitude across the region, and it is consistent with the considerable variability in the physical processes on water turbulence across environmental settings. Similar broad range of $k_{600}$ levels was also observed in the China's Yellow basin (ca. 0-123 cm/h) (Ran *et al.*, 2015; Ran *et al.*, 2017).

Contrary to our expectations, no significant relationship was observed between $k_{600}$ and water depth, and current velocity using the entire data in the three (TGR, Danning and Qjiang) river systems (Fig. S4). There were not statistically significant relationships between $k_{600}$ and wind speed using separated data or combined data, and it is consistent with earlier studies (Alin *et al.*, 2011; Raymond *et al.*, 2012). Flow velocity showed linear relation with $k_{600}$, and the extremely high value of $k_{600}$ was observed during the periods of higher flow velocity (Fig. S4a) using combined data. Similar trend was also observed between water depth and $k_{600}$ values (Fig. S4b). The lack of strong correlation between $k_{600}$ and physical factors are probably due to combined effect of both flow velocity and water depth, as well as large diversity of channel morphology, both across and within river networks in the entire catchment (60, 000 $\mathrm{km}^2$). This is further collaborated by weak correlations between $k_{600}$ and flow velocity in the TGR rivers (Fig. 4), where one or two samples were taken for a large scale examination. $k_{600}$ as a function of water depth was obtained in the TGR rivers, but it explained only 30% of the variance in $k_{600}$. However, model using data from Qijiang could explain 68% of the variance in $k_{600}$ (Fig. 4b), and it was in line with general theory. Nonetheless, $k_{600}$ from our flow velocity based model (Fig. 4b) was potentially largely overestimated with consideration of other measurements (Alin *et al.*, 2015; Ran *et al.*, 2015; Ran *et al.*, 2017). When several extremely values were removed, $k_{600}$ (cm/h) was parameterized as follows ($k_{600} = 62.879FV + 6.8357$, $R^2 = 0.52$, p=0.019, FV-flow velocity with a unit of m/s), and this revised model was in good agreement with the model in the river networks of the Yellow River (Ran *et al.*, 2017), but much lower than the model developed in the Yangtze system (Liu *et al.*, 2017) (Fig. 4c). This was reasonable because of $k_{600}$ values in the Yangtze system were from large rivers with higher turbulence than Yellow and our studied rivers. Furthermore, the determined $k_{600}$ using FCs was, on average, consistent with the revised model (Table 2). These differences in relationship between spatial changes in $k_{600}$ values and physical characteristics further corroborated heterogeneity of channel geomorphology and hydraulic conditions across the investigated rivers."

*3. The pCO2 calculated in this paper is between 50-4830ppm, which indicates that the river pCO2 value is sometimes much lower than that of the atmosphere, that is, the studied rivers can sometimes absorb CO2 from the atmosphere. It seems that the annual CO2 flux for the whole basin was calculated in this paper based on the averaged K value from the observed results using floating chamber method. The question is that is it reliable to estimate both directions of the CO2 flux at the air-water interface (including river CO2 outgassed to the atmosphere and the atmospheric CO2 input to rivers) by using the same K value? Or what uncertainty will it cause?*

**Response**: We thanked for your critical comment. Worldwide studies reported the dependence of k on flow dynamics; while k average from statistical analysis with $\Delta pCO_2 > 110$ µatm was normally used in riverine $CO_2$ flux estimation in rivers where a broad of range of $pCO_2$ occurred (Alin et al., 2011). Considering that k was largely dependent on hydraulic characteristics, this uncertainty was not discussed. The method employed here for scaling up $CO_2$ estimation was widely used (e.g., Alin et al., 2011). We added one section 4.4 ( see Discussion part) on systematic errors and empiric random errors of $pCO_2$, $CO_2$ areal flux and k from $pCO_2$ determinations, sampling seasonality, flux measurements etc.

*4. This study measured DOC, DTN, and DTP, but the authors did not mention these measurements in the discussion section. What is the relationship between these variables and the K values?*

**Response**: k values are reported to be dominated by physical characteristics of river systems (Borges et al., 2004; Alin et al., 2011; Raymond et al., 2012). We re-wrote this part and pattern of nutrients in the "Result" section was removed.

Specific comments
*5. L 94-97: I would suggest rephrase these sentences, since they cannot convey clearly the real contribution or scientific merit of this study.*

**Response**: We revised as follows.
"Our new contributions to the literature include (1) determination and controls of k levels for small rivers and streams in subtropical areas of China, and (2) new k models developed in the subtropical mountainous river networks"

*6. L 111-112, 117: The classification method of the river order used here should be clarified. The number of a river order defined by different classification system may represent different size or hierarchy of a river.*

**Response**: We addressed this issue, and provided the map of sampling location (Fig. 1). We also supplied methodology of water areal extraction in section 2.5.

*7. L 214-216: This statement is problematic. Clearly, the studied rivers are not always supersaturated reference to atmospheric CO2 as the pCO2 in rivers is between 50- 4830 uatm.*

**Response**: We changed to "$pCO_2$ varied between 50 and 4830 μatm with mean of 846 ± 819 μatm (Table 1). There were 28.7% of samples that had $pCO_2$ levels lower than 410 μatm, while the studied rivers were overall supersaturated with reference to atmospheric $CO_2$ and act as a source for the atmospheric $CO_2$."

*8. L 274-285 These arguments need more solid evidence to support. As mentioned in the general comments, I would suggest that the authors focus on discussions on relationship between spatial change in K values and physical characteristics of rivers or/ and the river orders.*

**Response**: Spatial differences of k values were discussed for the three rivers systems (please refer to section 4.2). We discussed the controls of physical characteristics of current velocity, slope and the water depth.

*9. L 497-498 The water area is a very critical parameter for the calculation of CO2 flux in a basin, so the acquisition of water area is essential and should be described more in detail. For example, what is the resolution of the satellite image? In addition, the variation of surface area of water between wet and dry seasons should be considered.*

**Response**: We provided the information for the acquisition of water area and citation was included (please refer to Methodology section).

**2.5. Estimation of river water area**

Water surface is an important parameter for $CO_2$ efflux estimation, while it depends on its climate, channel geometry and topography. River water area therefore largely fluctuates with much higher areal extent of water surface particularly in monsoonal season. However, most studies do not consider this change, and a fraction of the drainage area is used in river water area calculation (Zhang *et al.*, 2017). In our study, a 90 m resolution SRTM DEM (Shuttle Radar Topography Mission digital elevation model) data and Landsat images in dry season were used to delineate river network, and thus water area (Zhang *et al.*, 2018), whilst, stream orders were not extracted. Water area of river systems is generally much higher in monsoonal season in comparison to dry season, for instance, Yellow River showed 1.4-fold higher water area in the wet season than in the dry season (Ran *et al.*, 2015). Available dry-season image was likely to underestimate $CO_2$ estimation.

Ran, L.S., Lu, X.X., Yang, H., Li, L.Y., Yu, R.H., Sun, H.G., Han, J.T., 2015. CO2 outgassing from the Yellow River network and its implications for riverine carbon cycle. Journal of Geophysical Research-Biogeosciences 120, 1334-1347.
Zhang, T., Li, J., Pu, J., Martin, J.B., Khadka, M.B., Wu, F., Li, L., Jiang, F., Huang, S., Yuan, D., 2017. River sequesters atmospheric carbon and limits the CO2 degassing in karst area, southwest China. Science of The Total Environment 609, 92-101.
Zhang, J., Li, SY., Dong, RZ., Jiang, CS., 2018. Physical evolution of the Three Gorges Reservoir in Holocene using advanced SVM on Landsat images and SRTM DEM data. Environ Sci Pollut Res 25, 14911-14918.

*10. Finally, I would suggest the authors polish the English grammar and writing, as well as the figs presenting.*

**Response**: We carefully edited the Ms and re-organized the Tables and Figures presentation.

**Response to Anonymous Referee #3**

*General comments*
*The manuscript of Li et al. presents measured CO2 fluxes, transport coefficients based on CO2, and calculated pCO2 data of running waters in a subtropical monsoonal climate zone. These data are complemented by among others water chemistry parameters such as DOC, DTN, DTP, as well as hydrogeomorphology data (e.g. water depth, flow velocity). They provide data and insights about transport coefficients for a so far understudied region and highlight the spatial variability and subsequent uncertainty for regional upscale estimates.*
*By investigating the key parameter for CO2 flux estimates - the transport coefficient - in an understudied region, Li et al. address a very relevant topic. Narrowing down the uncertainties of regional upscaling estimates of riverine CO2 fluxes is of wide interest, hence this study would make a good contribution to the literature and the subject matter is thus of interest to Biogeosciences readers.*

**Response: We thank you for your overall positive comments, and accordingly revised the Ms.**

*However, in my opinion, the manuscript has some problems:*
*(1) The terminology used in this manuscript is quite confusing to me. It seems to me that "streams",*
*"rivers", "river networks" are used interchangeably (without definition and consistency), which makes it hard to follow the red line of the story. The terminology needs to be clarified and unified.*

**Response: Based on the delineation of river systems by Alin et al., 2011 (JGR), small rivers and streams encompass rivers with channels < 100 m. We have clarified the term in the method.**

*(2) The sampling design is not very clear to me. All investigated running waters seem to be in the Three Gorges Reservoir (TGR) region, but in addition two larger streams (Daning and Qijiang) were sampled. In the results and discussion, these investigated running waters are combined, sometimes split, which makes it hard to follow (in the main text and tables). In my opinion, these three "regions" need to be presented in a unified way (always separated or combined, possibly both in each table and figure), and presented more clearly in the text.*

**Response: We provided both separated and combined data in Tables and Figures, p***lease refer to Figs 2 and 3, as well as Figs. S2 and S3. In addition, we also clearly stated this in the "Method" part.***

"Spatial differences (Daning, Qijiang and entire tributaries of TGR region) were tested using the nonparametric Mann Whitney U-test. Multivariate statistics, such as correlation and stepwise multiple linear regression, were performed for the models of $k_{600}$ using potential physical parameters of wind speed, water depth, and current velocity from separated data and combined data (Alin *et al.*, 2011)."

*(3) One of the main messages is the presentation of transport coefficients in a subtropical monsoonal climate zone, which is interesting, but I can imagine that there is a large difference in the wet and dry season. However, all the measurements were done in the wet season. I think this issue should be clearly acknowledged and discussed.*

**Response: We agreed your opinion and addressed this issue in an additional section 4.4. We**

**sampled in the monsoonal period as it covered the flow velocity in the whole hydrological year. Albeit $k_{600}$ is higher in wet season than dry season, our main objectives are to develop models of k rather than the annual evasion. In this section, effects and uncertainty of sampling seasonality on errors of annual $CO_2$ flux estimation were also included.**

"Sampling seasonality considerably regulated riverine $pCO_2$ and gas transfer velocity and thus water-air interface $CO_2$ evasion rate (Li *et al.*, 2012; Ran *et al.*, 2015). We sampled waters in wet season (monsoonal period) due to that it showed wider range of flow velocity and thus it covered the $k_{600}$ levels in the whole hydrological season. Wet season generally had higher current velocity and thus higher gas transfer velocity (Ran *et al.*, 2015), while aquatic $pCO_2$ was variable with seasonality. We recently reported that riverine $pCO_2$ in the wet season was 81% the level in the dry season (Li *et al.*, 2018), and prior study on the Yellow River reported that k level in the wet season was 1.8-fold higher than in the dry season (Ran *et al.*, 2015), while another study on the Wuding River demonstrated that k level in the wet season was 83%-130% of that in the dry season (Ran *et al.*, 2017). Thus, we acknowledged a certain amount of errors on the annual flux estimation from sampling campaigns during the wet season in the TGR area, while this uncertainty could not be significant because that the diluted $pCO_2$ could alleviate the overestimated emission by increased k level in the wet season (stronger discussion please refer to SOM)."

*(4) There are two technical issues: (i) The measurements with the floating chambers are poorly described. The only information Li et al. provide is that the floating chambers were "deployed". If the flux measurements are done in an anchored or free floating manner is critical (see e.g. Lorke, A., Bodmer, P., Noss, C., Alshboul, Z., Koschorreck, M., Somlai-Haase, C., Bastviken, D., Flury, S., McGinnis, D. F., Maeck, A., Müller, D., and Premke, K.: Technical note: drifting versus anchored flux chambers for measuring greenhouse gas emissions from running waters, Biogeosciences, 12, 7013-7024, https://doi.org/10.5194/bg-12-7013-2015, 2015.). Hence, this issue needs to be addressed clearly. (ii) It seems that all the flux and pCO2 measurements were done distributed during the day. The fact that there is a diurnal cycle of CO2 was not considered (see e.g. Pascal Bodmer, Marlen Heinz, Martin Pusch, Gabriel Singer, Katrin Premke, Carbon dynamics and their link to dissolved organic matter quality across contrasting stream ecosystems, Science of The Total Environment, Volume 553, 2016, Pages 574-586, ISSN 0048-9697, https://doi.org/10.1016/j.scitotenv.2016.02.095.), and values directly compared. This issue should at least be discussed.*

**Response: A new section of "4.4. Uncertainty assessment of $pCO_2$ and flux-derived k values" was added. We assessed systematic errors and random errors of measurements, drifting chamber and anchored chambers, sampling time. We carefully read the two article and the two citations were included. In addition, the following text was added in the Method section.**

"Similar to other studies (Alin et al., 2011), sampling and flux measurements in the day time would tend to underestimate $CO_2$ evasion rate (*Bodmer et al., 2016*)."

*(5) Developing models to estimate transport coefficients is meaningful, but the process of the model development is poorly described. Additionally, which data were used for the models, and which not is confusing to me (goes along with my comment (2) above).*

**Response: The issue was addressed as follows.**

Water samples from a total of 115 sites were collected. Floating chambers with replicates were deployed in 101 sites (32 sampling sites in Daning, 37 sites in TGR river networks and 32 sites in Qijiang). The sampling period covered spring and summer season, our sampling points are reasonable considering a water area of 433 km$^2$. For example, 16 sites were collected for Yangtze system to examine hydrological and geomorphological controls on $pCO_2$ (Liu *et al.*, 2017), and 17 sites for dynamic biogeochemical controls on riverine $pCO_2$ in the Yangtze basin (Liu *et al.*, 2016). Similar to other studies, sampling and flux measurements in the day would tend to underestimate $CO_2$ evasion rate (Bodmer *et al.*, 2016).

Prior to statistical analysis, we excluded $k_{600}$ data for samples with the air-water $pCO_2$ gradient <110 µatm, since the error in the $k_{600}$ calculations drastically enhances when $\triangle pCO_2$ approaches zero (Borges *et al.*, 2004; Alin *et al.*, 2011), and datasets with $\triangle pCO_2$ >110 µatm provide an error of <10% on $k_{600}$ computation (please refer to the Fig. 1 in the bottom). Thus, we discarded the samples (36.7% of sampling points with flux measurements) with $\triangle pCO_2$ <110 µatm for $k_{600}$ model development, while for the flux estimations from diffusive model and floating chambers, all samples were included.

Multivariate statistics, such as correlation and stepwise multiple linear regression, were performed for the models of $k_{600}$ using potential physical parameters of wind speed, water depth, and current velocity as the independent variables from both separated data and combined data (Alin *et al.*, 2011). k models were obtained by water depth using data from the TGR rivers, while by flow velocity in the Qijiang.

We also highlighted separated or combined data used in the Tables and Figures.

*(6) From what I see in these data, there are several running waters undersaturated with respect to CO2 (Fig. S2 and Fig. 1), and hence a sink of CO2. This aspect is totally neglected and the investigated running waters are generalized as CO2 sources to the atmosphere. In my opinion, this aspect of influxes of CO2 is very valuable and should be properly discussed.*

**Response: We addressed this issue by revision in the parts of "Result" and "Discussion".**
*Firstly, we revised in the "Result" section as follows.* "$pCO_2$ varied between 50 and 4830 µatm with mean of 846 ±819 µatm (Table 1). There were 28.7% of samples that had $pCO_2$ levels lower than 410 µatm, while the studied rivers were overall supersaturated with reference to atmospheric $CO_2$ and act as a source for the atmospheric $CO_2$."

*Secondly, the under-saturated $pCO_2$ levels were further examined in the "Discussion" part.*

"The calculated $pCO_2$ levels were within the published range, but towards the lower-end of published concentrations compiled elsewhere (Cole and Caraco, 2001; Li *et al.*, 2013). The total mean $pCO_2$ (846 ±819 µatm) in the TGR, Danning and Qijiang sampled was lower than one third of global river's average (3220 µatm) (Cole and Caraco, 2001). The lower $pCO_2$ than most of the world's river systems, particularly the under-saturated values, demonstrated that heterotrophic respiration of terrestrially derived DOC was not significant. Compared with high alkalinity, the limited delivery DOC particularly in the Daning and Qijiang river systems (Figs. S2 and S3) also indicated that in-stream respiration was limited. These two river systems are characterized by karst terrain and underlain by carbonate rock, where photosynthetic uptake of dissolved $CO_2$ and carbonate minerals dissolution considerably regulated aquatic $pCO_2$ (Zhang *et al.*, 2017).*"*

*(7) As far as I see, there is some arbitrariness regarding data handling/processing. The cut-off at 110 μatm (line 198) for the air-water CO2 gradient for k600 calculations, as well as "When several extremely values are removed: : :" (line 303), needs to be described/demonstrated/justified much more clear.*

**Response: We provided more details on this concern.**
    Prior to statistical analysis, we excluded $k_{600}$ data for samples with the air-water $pCO_2$ gradient <110 μatm, since the error in the $k_{600}$ calculations drastically enhances when $\triangle pCO_2$ approaches zero (Borges et al., 2004; Alin et al., 2011), and datasets with $\triangle pCO_2$ <110 μatm provide an error of <10% on $k_{600}$ computation" (see the Fig. 1 in the bottom)

***Regarding "*"When several extremely values are removed: : :" (L 303) for revised model, we supplied data including extremely data and excluding extremely data in Fig. 4 (the original Fig. 3).***

*(8) CO2 fluxes were measured, while pCO2 and transport coefficients were calculated. This should be clearly stated throughout the manuscript to be transparent.*

**Response: We highlighted this in the section of "Method" (see section 2.4), and changed "measured k or observed k to flux-derived k or derived k or calculated k".**

*(9) I am not a native English speaker, but I think that the manuscript should be revised for the English language. (see exemplarily in the specific comments and technical corrections below)*
*I think this is a valuable study, but the combination of the points mentioned above make the manuscript hard to follow and the conclusions and main messages drawn in the current state of the manuscript too general. In a revised version, the study would get more shaped, more detailed and informative, and the conclusions and main messages can be specified and more related to the investigated region.*

**Response: We edited the English and revised the Ms based on comments and suggestions.**

Specific comments
Abstract:
*Line 20: Indicate how many river networks (see general comment (2))*

**Response: 60 rivers**

*Line 24: As far as I understood not when all data were included. Please be more specific here.*

**Response: Corrected and provided details in "Method" section.**

*Lines 30 – 33: This is not really new. Maybe you can specify this statement for the investigated region?*

**Response: Addressed. The sentence was revised as follows.**

"We concluded that simple parameterization of k as a function of morphological characteristics was site specific and hence highly variable in river systems of the upper Yangtze. k models should be developed for stream studies to evaluate the contribution of these regions to atmospheric $CO_2$."

Introduction:
*Line 41: Bastviken et al., 2011 totally focuses on CH4. I suggest replacing this reference with a more suitable one.*

**Response: Bastviken et al., 2011 was replaced by "Raymond et al., 2013; Butman and Raymond, 2011"**

*Line 42: But you did not present "new accurate measurement techniques" in your study, what are your reasons to mention this in the introduction?*

**Response: "new" was removed.**

*Lines 50 – 58: This equation is pretty standard knowledge and can just be described in words here. The equation can be moved to the methods.*

**Response:** Yes, it is very standard knowledge, while the text would be more shape, and easily to follow with equations here.

*Line 63: The standardized transport coefficient (k600) should be explained here.*

**Response: $k_{600}$ (the standardized transfer coefficient at a temperature of 20 $^0$C)**

*Line 80: You set the scene of seasonal precipitation, but in the study, you only measure in the wet season. This is contradictory. This issue should be discussed.*

**Response: We changed "concentrated seasonal precipitation" to "hydrological seasonality"**

*Lines 84-89: Kind of repetition and partially contradictory to the text in lines 43-49.*

**Response: The topic is different. The former focuses on flux determination, while the latter talked about k measurement method. We clarified the main text.**

*Lines 92-99: The relevance of the study (first time in this region, etc.), the objectives and how these objectives were approached should be written more clearly. At this point, the input parameters for the model development is totally unclear.*

**Response: We re-wrote this part as follows, we also provide details for k models.**

To contribute to this debate, extensive investigation was firstly accomplished for determination of k in rivers and streams of the upper Yangtze using FC method. Models of k were further developed using hydraulic properties from flux measurements and TBL model.

Our recent study preliminarily investigated $pCO_2$ and air – water $CO_2$ areal flux as well as their controls from fluvial networks in the Three Gorges Reservoir (TGR) area (Li *et al.*, 2018). The past study was based on two field works, and the diffusive models from other continents were used. In this study, we attempted to derive k levels and develop the gas transfer model in this area (mountainous streams and small rivers) for more accurate quantification of $CO_2$ areal flux, and also to serve for the fluvial networks in the Yangtze River or others with similar hydrology and geomorphology. Moreover, we did detailed field campaigns in the two contrasting rivers Daning and Qijiang for models (Fig. 1). The study thus clearly stated distinct differences than the previous study (Li *et al.*, 2018) by the new contributions of specific objectives and data supplements, as well as wider significance.

Our new contributions to the literature include (1) determination and controls of k levels for small rivers and streams in subtropical areas of China, and (2) new k models developed using hydraulic parameters in the subtropical mountainous river networks.

*Materials and methods:*
*Line 105: In my opinion, Figure S1 should go to the main text. There are no sampling points for Daning and Qijiang, which is confusing to me.*

**Response: We supplied the map in the main text as Fig. 1 (see Fig. 2 in the bottom).**

*Lines 105-109: Please add a reference for this statement.*

**Response: "Li et al., 2018" was cited here.**

*Lines 110-118: Please see my general comment (2): Please restructure this, make clear how many running waters were sampled where, the size of the sampled running waters (Strahler stream order is fine), and why in these three regions. Otherwise, it is hard to follow your storyline.*

**Response: We provided details in the section 2.2.**

Water samples from a total of 115 sites were collected. Floating chambers with replicates were deployed in 101 sites (32 sampling sites in Daning, 37 sites in TGR river networks and 32 sites in Qijiang). The sampling period covered spring and summer season, our sampling points are reasonable considering a water area of 433 $km^2$. For example, 16 sites were collected for Yangtze system to examine hydrological and geomorphological controls on $pCO_2$ (Liu *et al.*, 2017), and 17 sites for dynamic biogeochemical controls on riverine $pCO_2$ in the Yangtze basin (Liu *et al.*, 2016). Similar to other studies, sampling and flux measurements in the day would tend to underestimate $CO_2$ evasion rate (Bodmer *et al.*, 2016). In our sampling points, all measured fluxes were retained since the floating chambers yielded linearly increasing $CO_2$ against time following manufacturer' specification.

Prior to statistical analysis, we excluded $k_{600}$ data for samples with the air-water $pCO_2$ gradient <110 μatm, since the error in the $k_{600}$ calculations drastically enhances when $\triangle pCO_2$ approaches zero (Borges *et al.*, 2004; Alin *et al.*, 2011), and datasets with $\triangle pCO_2$ >110 μatm provide an error of <10% on $k_{600}$ computation. Thus, we discarded the samples (36.7% of sampling points with flux measurements) with $\triangle pCO_2$ <110 μatm for $k_{600}$ model development, while for the flux estimations from diffusive model and floating chambers, all samples were included.

*Line 141-142: What is "PP"?*

**Response: EGM-4 (Environmental Gas Monitor; PP SYSTEMS Corporation, USA)**

*Line 148: I don't really understand what you mean by this sentence, please revise.*

**Response: Changed to "All the solvents and reagents used in experiments were of analytical - reagent grade"**

*Lines 155-156: This sentence is confusing to me, please revise.*

**Response: Changed "The relationship was yielded when z=1 ($U_{10}=1.208 \times U_1$). " to "$U_{10}=1.208 \times U_1$ as we measured the wind speed at a height of 1 m ($U_1$). "**

*Line 158: Do you mean CO2SYS? If yes, please add the corresponding reference.*

**Response: "Lewis et al., 1998" was cited.**

Lewis, E., Wallace, D., Allison, L.J., 1998. Program developed for CO2 system calculations.; Brookhaven National Lab., Dept. of Applied Science, Upton, NY (United States); Oak Ridge National Lab., Carbon Dioxide Information Analysis Center, TN (United States), p. Medium: ED; Size: 40 p.

*Line 167: What was the brand of the tubing?*

**Response: Changed to "$CO_2$ impermeable rubber-polymer tubing"**

*Line 170: What is DC?*

**Response: "DC" was removed.**

*Line 173: Please see my general comment (4) (i)*

**Response: Uncertainty of chambers was discussed in the additional section 4.4. In our study, both drifting chamber and anchored chambers were used, which is dependent on *in situ* current velocity. The following text was added.**

"In sampling sites with low and favorable flow conditions (Fig. S1), freely drifting chambers (DC) were executed, while sampling sites in rivers and streams with higher flow velocity were conducted with anchored chambers (AC) (Ran *et al.*, 2017). AC would create overestimation of $CO_2$ emissions in our studied region (Lorke et al., 2015)."

*Line 177: The units are confusing to me. Why is there two times pressure and temperature? Please double check if the units match up in the end, to me they do not.*

**Response: We have carefully checked and revised.**

*Line 187: Please be more specific: k was calculated by reorganizing Eq (1)*

**Response: Revised.**

*Line 192: Sc to the power of 0.5? This seems weird. What do you mean here?*

**Response: Corrected.**

*Line 198: Please justify the cut-off at 110 _atm. Maybe add a figure to the supplementary material.*

**Response: Revised. Please refer to general comment (7).**

*Line 203: I read about water depth and current velocity the first time here. These measurements need to be described before.*

**Response: Measurements of water depth and current velocity were added in the part of "Methods".**

*Line 213: The pH is quite high. This in combination with influxes of CO2 requires at least a short discussion about chemical enhancement.*

**Response: We revised the text in the section of "Result" (see the first paragraph below), and added text (see the second paragraph below) in the Discussion section (4.1).**

"$p$H varied from 7.47 to 8.76 with exceptions of two quite high values of 9.38 and 8.87 (mean: 8.39 $\pm$ 0.29 from total dataset). Much lower $p$H was observed in TGR rivers (8.21 $\pm$ 0.33) (Table 1; p<0.05; Fig. S2)."

"Higher pH levels were observed in Daning and Qijiang (p<0.05 by ANOVA), where more carbonate rock exists that are characterized by karst terrain. Our pH range was comparable to the recent study on the karst river in China (Zhang *et al.*, 2017). Quite high values (i.e., 9.38 and 8.87) were recorded in the investigated sites, where chemical enhancement would increase the influx of atmospheric $CO_2$ to alkaline waters (Wanninkhof and Knox, 1996), while 1.7% of sampling sites that were strongly affected by chemical enhancement were not significant on a regional scale. This chemical enhancement of $CO_2$ influx was also reported to be limited in high-pH rivers (Zhang *et al.*, 2017)."

Wanninkhof, R., Knox, M., 1996. Chemical enhancement of CO2 exchange in natural waters. Limnology and Oceanography 41, 689-697.

*Line 214: Please see my general comment (6)*

**Response: Addressed. Please refer to general comment (6)**

*Lines 218-222: This paragraph belongs to the discussion section.*

**Response: We moved this to Discussion (see 4.1)**

*Lines 223-227: This paragraph should be revised because it is not very clearly written. Please add the p-values to the text in case of significances.*

**Response: We re-wrote this part, and nutrients were removed.**

*Lines 235-237: What is the meaning of this ratio? Please add a few words what the reader can get from this information.*

**Response: Removed.**

*Line 242: These models and how you developed them should be explained better (in the method section).*

**Response: We provided details in the section of "Method".**

*Lines 246-248: I do not understand this sentence. What do you mean by "binned"?*

**Response: Changed to "combined"**

Discussion:
*Lines 270-274: How does this paragraph support the discussion of your study?*

**Response: These texts could support the discussion on relationship between spatial change in k values and physical characteristics (i.e., current velocity, slope and the water depth) of three river systems**.

Spatial differences of k values and their controls of physical characteristics of current velocity, slope and the water depth were discussed for the three rivers systems (please refer to section 4.2).

"This could substantiate the higher $k_{600}$ levels and spatial changes in $k_{600}$ values of our three river systems. For instance, similar to other turbulent rivers in China (Ran *et al.*, 2015; Ran *et al.*, 2017), high $k_{600}$ values in the TGR, Daning and Qjiang rivers were due to mountainous terrain catchment, high current velocity (10 – 150 cm/s) (Fig. 4b), bottom roughness, and shallow water depth (10 - 150 cm) (Fig. 4a). It has been suggested that shallow water enhances bottom shear, and the resultant turbulence increases k values (Alin *et al.*, 2011; Raymond *et al.*, 2012). These physical controls are highly variable across environmental types (Figs. 4a and 4b), hence, k values are expected to vary widely (Fig. 3). The $k_{600}$ values in the TGR rivers showed wider range (1-177 cm/h; Fig. 3; Table S1), spanning more than 2 orders of magnitude across the region, and it is consistent with the considerable variability in the physical processes on water turbulence across environmental settings. Similar broad range of $k_{600}$ levels was also observed in the China's Yellow basin (ca. 0-123 cm/h) (Ran *et al.*, 2015; Ran *et al.*, 2017).

Contrary to our expectations, no significant relationship was observed between $k_{600}$ and water depth, and current velocity using the entire data in the three (TGR, Danning and Qjiang) river systems (Fig. S4). There were not statistically significant relationships between $k_{600}$ and wind speed using separated data or combined data, and it is consistent with earlier studies (Alin *et al.*, 2011; Raymond *et al.*, 2012). Flow velocity showed linear relation with $k_{600}$, and the extremely high value of $k_{600}$ was observed during the periods of higher flow velocity (Fig. S4a) using combined data. Similar trend was also observed between water depth and $k_{600}$ values (Fig. S4b). The lack of strong correlation between $k_{600}$ and physical factors are probably due to combined effect of both flow velocity and water depth, as well as large diversity of channel morphology, both across and within river networks in the entire catchment (60, 000 $km^2$). This is further collaborated by weak correlations between $k_{600}$ and flow velocity in the TGR rivers (Fig. 4), where one or two samples were taken for a large scale examination. $k_{600}$ as a function of water depth was obtained in the TGR rivers, but it explained only 30% of the variance in $k_{600}$. However, model using data from Qijiang could explain 68% of the variance in $k_{600}$ (Fig. 4b), and it was in line with general theory. Nonetheless, $k_{600}$ from our flow velocity based model (Fig. 4b) was potentially largely overestimated with consideration of other measurements (Alin *et al.*, 2015; Ran *et al.*, 2015; Ran *et al.*, 2017). When several extremely values were removed, $k_{600}$ (cm/h) was parameterized as follows ($k_{600} = 62.879FV + 6.8357$, $R^2 = 0.52$, p=0.019, FV-flow velocity with a unit of m/s), and this revised model was in good agreement with the model in the river networks of the Yellow River (Ran *et al.*, 2017), but much lower than the model developed in the Yangtze system (Liu *et al.*, 2017) (Fig. 4c). This was reasonable because of $k_{600}$ values in the Yangtze system were from large rivers with higher turbulence than Yellow and our studied rivers. Furthermore, the determined $k_{600}$ using FCs was, on average, consistent with the revised model (Table 2). These differences in relationship between spatial changes in $k_{600}$ values and physical characteristics further corroborated heterogeneity of channel geomorphology and hydraulic conditions across the investigated rivers."

*Lines 286-288: No significances: : : But still, you developed the models considering all data? This is not at all clear to me. Did you split/separate the data set for the models? This is not clear in Table 2. Please see my general comment (2). I think these data/models are valuable, but at the moment they seem arbitrary and should be better explained. This would help to give them more weight.*

**Response: We now provided details of model developing in the "Method" (see the response to general comment (2), (5) and (7). We also clearly stated the separated or combined data for models in the captions of Tables and Figs.**

*Lines 286-309: I see a lot of results here, which are presented in the discussion for the first time. The part presenting pure results should be moved to the results section.*

**Response: Revised. The following text was moved to the "RESULT".**
"Contrary to our expectations, no significant relationship was observed between $k_{600}$ and water depth, and current velocity using the entire data in the three river systems (TGR, Danning and Qjiang) (Fig. S4). There were not statistically significant relationships between $k_{600}$ and wind speed using separated data or combined data. Flow velocity showed linear relation with $k_{600}$, and the extremely high value of $k_{600}$ was observed during the periods of higher flow velocity (Fig. S4a) using combined data. Similar trend was also observed between water depth and $k_{600}$ values (Fig. S4b). $k_{600}$ as a function of water depth was obtained in the TGR rivers, but it explained only 30% of the variance in $k_{600}$. However, model using data from Qijiang could explain 68% of the variance in $k_{600}$ (Fig. 4b), and it was in line with general theory. "

*Line 303: Please justify the removal of "extremely values". Maybe add a figure to the supplementary material. If there is no objective criteria and justification, I do not see why data should be removed.*

**Response: We provided details in the Method and also supplied figure with or without extremely values in the main text (Fig. 4).**

*Lines 327-333: Why discussing k values and not k600 values? I think this needs to be unified/consistent throughout the manuscript.*

**Response: We unified to $k_{600}$ though $k_{600}$ and k were widely discussed in previous studies.**

*Conclusion:*
*Lines 358-360: Very general, but actually, the regions had to be separated/ split, no?*

**Response: The words were removed.**

*Lines 368-369: I think you should focus the conclusion on the investigated region.*

**Response: "in the river systems of the upper Yangtze River" was added.**

*Tables:*
*Table 3: b) Why not presenting k600 values here which can be directly compared with other studies?*

**Response: We unified to "$k_{600}$"**

*Figures:*
*Fig. S1: Was always everything sampled at each point? "Samples" should be replaced by "sampling point"*

**Response: We moved it to the main text as Fig. 1, and "Samples" was changed to "Sampling point".**

*Fig. S4: There is no reference to this figure in the main text.*

**Response: Fig. S4 was cited in the main text.**

*Technical corrections*
*Line 22: Delete "were"*
*Line 44: add "by" or "via" before "floating chambers"*
*Line 59: Replace "precisely" with "well"*
*Line 127: "consisted" instead of "consists"*
*Line 139: "pH sonde" "was" instead of "is"*
*Line 225: "Daning" instead of "Danning"*

**Response: All the typos were corrected.**

[revised manuscript text omitted]

**(TGR rivers)**

[revised manuscript text omitted]

**(Qijiang River)**

| Date | Time | River | T(air) °C | Wind speed m/s | T(water) °C | pH | EC µs/cm | TDS mg/L | River width m | River depth m | Water velocity m/s | Alk µeq/L | DOC mg/L | TDN mg/L | TDP mg/L |
|---|---|---|---|---|---|---|---|---|---|---|---|---|---|---|---|
| 2016.8.15 | 13:30 | BZD1 | 39.6 | 0.5 | 30 | 8.41 | 351.6 | 229 | 3 | 20 | 0.18 | 2920 | 1.34 | 2.58 | 10.05 |
| | 14:30 | BZD2 | 40.5 | 1 | 25 | 8.59 | 317 | 206 | 20 | 40 | 0.5 | 2472 | 1.61 | 2.09 | 10.05 |

| Date | Time | Code | | | | | | | | | | | | | |
|---|---|---|---|---|---|---|---|---|---|---|---|---|---|---|---|
| | 15:30 | BZD3 | 37.3 | 0.7 | 31 | 8.67 | 301.8 | 196 | | | | 2656 | 2.32 | 1.99 | 5.03 |
| | 15:40 | BZD4 | 34.4 | 0.6 | 31 | 8.52 | 269.1 | 175 | | | | 1824 | 14.88 | 0.53 | 5.03 |
| | 16:10 | BZD5 | 33.7 | 1.6 | 32 | 8.52 | 282.5 | 184 | 20 | 80 | 0.6 | 2336 | 1.48 | 0.87 | 5.03 |
| | 16:45 | BZD6 | 34.4 | 2.2 | 30 | 8.3 | 448.2 | 292 | 2 | 50 | 0.05 | 2216 | 1.65 | 1.89 | 5.03 |
| 2016.8.16 | 9:10 | BSY1 | 31 | 0.4 | 22 | 8.64 | 322.3 | 210 | 3 | 60 | 0.8 | 3056 | 2.00 | 1.01 | 5.03 |
| | 9:50 | BSY2 | 33.1 | 0.5 | 23.3 | 8.49 | 345 | 224 | 6 | 50 | 0.5 | 2896 | 1.65 | 0.97 | 10.05 |
| | 10:25 | BSY3 | 31.6 | 1 | 23.5 | 7.94 | 374.7 | 244 | 20 | 45 | 0.2 | 2520 | 1.39 | 1.12 | |
| | 13:40 | BSK1 | 32.2 | 1 | 26.8 | 8.63 | 513.8 | 334 | 1 | 15 | 0.8 | 2552 | 4.46 | 1.99 | 5.03 |
| | 14:20 | BSK2 | 36.1 | 1.2 | 21 | 8.27 | 338.6 | 220 | 2 | 35 | 0.8 | 2288 | 32.01 | 2.05 | 10.05 |
| | 15:00 | BSK3 | 41.8 | 0.5 | 30.5 | 8.67 | 286.7 | 186 | 12 | 50 | 0.3 | 2600 | 3.72 | 0.97 | 10.05 |
| | 15:40 | BSK4 | 35.4 | 1.3 | 31 | 8.16 | 416.2 | 271 | 1.5 | 60 | 0.4 | 1896 | 1.88 | 0.99 | 5.03 |
| | 16:00 | BSK5 | 38.3 | 0.8 | 26.5 | 8.68 | 334.5 | 218 | 3 | 25 | 0.7 | 2048 | 2.53 | 1.57 | 10.05 |
| | 16:20 | BSK6 | 40 | 0.8 | 29 | 8.56 | 381.2 | 248 | 2 | 50 | 1 | 1728 | 1.50 | 1.29 | 20.10 |
| | 17:00 | BSK7 | 38.1 | 0.6 | 32.3 | 8.76 | 335.2 | 218 | 2 | 10 | 0.5 | 1984 | 1.65 | 0.86 | 5.03 |
| 2016.8.17 | 9:30 | BSK8 | 30.2 | 0.8 | 23.3 | 8.48 | 368.9 | 240 | 11 | 15 | 0.3 | 3112 | 1.64 | 1.55 | 10.05 |
| | 10:00 | BSK9 | 32.3 | 0.4 | 26 | 8.4 | 362.9 | 236 | 11 | 15 | 0.3 | 2632 | 1.72 | 1.45 | 45.23 |
| | 11:00 | BSK10 | 34.4 | 0.2 | 29 | 8.56 | 367.4 | 239 | 20 | 35 | 0.2 | 2368 | 1.86 | 1.29 | |
| | 11:24 | BSK11 | 35.5 | 2.8 | 29.1 | 8.52 | 410.8 | 207 | 5 | 50 | 1.2 | 2544 | 1.40 | 0.71 | 5.03 |
| | 12:45 | BSK12 | 38.5 | 1.6 | 30 | 8.46 | 474.8 | 309 | 12 | 50 | 1 | 2904 | 1.44 | 1.33 | 40.20 |
| | 15:00 | BYD1 | 35 | 0.3 | 29.5 | 8.45 | 663.6 | 431 | 1 | 10 | 0.1 | 2240 | 2.25 | 2.38 | 10.05 |
| | 16:30 | BYD2 | 32.7 | 0.8 | 30 | 8.58 | 217.6 | 142 | 4 | 10 | 0.1 | 1808 | 4.74 | 1.03 | 5.03 |
| | 17:45 | BYD3 | 33 | 0.5 | 29 | 8.45 | 1087.9 | 707 | | | | 4488 | 2.00 | 1.62 | 15.08 |
| | 18:14 | BYD4 | 32.6 | 0.2 | 32 | 8.55 | 1099.5 | 716 | 5 | 30 | 1.2 | 3592 | 2.49 | 1.58 | 15.08 |
| | 17:00 | BYD5 | 32 | 0.2 | 30.5 | 8.53 | 1051.9 | 684 | 4 | 30 | 1.2 | 4080 | 1.73 | 1.26 | 5.03 |
| | 19:30 | BQJ1 | 32.3 | 1.5 | 30 | 8.56 | 623 | 405 | 40 | 45 | 0.6 | 3784 | 1.83 | 1.43 | 15.08 |
| 2016.8.18 | 10:00 | BQJ2 | 37 | 1.2 | 29.5 | 8.35 | 396.4 | 258 | | | | 2416 | 2.99 | 1.26 | 95.48 |

| | | | | | | | | | | | | | |
|---|---|---|---|---|---|---|---|---|---|---|---|---|---|
| 11:20 | BPH1 | 33.2 | 0.7 | | 31.9 | 8.49 | 1350.3 | 878 | | | | 3856 | 8.98 | 1.47 | 25.13 |
| 15:00 | BSX1 | 41.4 | 0.1 | | 32 | 8.42 | 150.9 | 98 | | | | 600 | 3.46 | 0.64 | 5.03 |
| 16:00 | BSX2 | 41.2 | 1 | | 34 | 8.23 | 179.9 | 117 | 8 | 10 | 0.1 | 1424 | 3.31 | 1.33 | 30.15 |
| 16:30 | BQG3 | 37.1 | 0.1 | | 32 | 8.43 | 422.2 | 274 | 15 | 30 | 0.5 | 2376 | 2.46 | 1.71 | 85.43 |

---

## Referee Report (RR1)

The paper has been greatly improved, and the reviewers' comments have been carefully replied. In my opinion, the research quality is now acceptable and can be published.

---

## Author Response (AR2)

**Response letter to associate editor**

**Associate Editor Decision: Publish subject to minor revisions (review by editor)** (17 Dec 2018) by David Butman
Comments to the Author:
Dear Authors,

After two complete reviews it is of our opinions that the manuscript is nearing publication quality. However, we agree with the points raised by Anonymous Referee #3 and would like these to be addressed prior to publication. We feel that this reviewer in particular has provided a comprehensive evaluation of the work, and we do not see these additional suggestions to be a significant burden. We look forward to a revised version of the manuscript soon.

Sincerely,
David Butman

Response: Thanks for your positive comments; we can accommodate all comments and suggestions from referees. Please see them in the response letter to Referees.

**Response letter to Referee #2**

**Comment on the revised manuscript by Siyue Li et al.**
The paper has been greatly improved, and the reviewers' comments have been carefully replied. In my opinion, the research quality is now acceptable and can be published.

Response: Thanks for your very positive comments and your hard work on our Ms.

**Response letter to Referee # 3**

Review bg-2018-227_R1

General comments
I can see that a lot of effort has been put in the revisions and I feel that the manuscript has definitely improved, it is clearer and much easier to follow. Changes such as restricting the upscaling to the monsoonal period is much more appropriate in my point of view, the cutoff of at delta pCO2 110 µatm for k600 calculations is well clarified, the overview map (Fig.1) gives a good impression over the sampling effort/sampling area, and the separation of the datasets for different purposes is more clear.

However, in my point of view, there are still some critical points which need to be addressed:
• Anchored vs. drifting chambers: I appreciate that you addressed this issue in the manuscript. Well, your k600 values are close to the average of Ran et al. (2015) (measured with drifting chambers) and Liu et al. (2017) (measured with static chambers in canoe shape), this indicates that your potential overestimation is limited. However, since you have a mix of anchored and drifting chamber measurements, you added a considerable amount of variability related to k600 values to your dataset. Potentially, this is part of the reason why you did not find significant correlations using the entire/complete data set? If possible, I suggest testing the relationships with chamber derived k600 values and flow velocity/depth only with the drifting chamber data. Alternatively, address this issue in the chapter where you discuss the uncertainty of the data (4.4).

Response: Because of our rivers are locating in the mountainous area, anchored chambers are mostly used. Furthermore, the cutoff of at delta pCO2 110 µatm for k600 calculations largely reduced the number of data for $k_{600}$ model. Thus, we can not separately use anchored and drifting chamber measurements for $k_{600}$ models. Based on the comment, we discussed the uncertainty in the section of 4.4.
  "Our $k_{600}$ values were close to the average of Ran et al. (2015) (measured with drifting chambers) and Liu et al. (2017) (measured with static chambers in canoe shape), this indicated that our potential overestimation was limited. However, since we had very limited drifting chamber measurements because of high current velocity, the relationships with chamber derived $k_{600}$ values and flow velocity/depth only with the drifting chamber data could not be tested. Whereas, we acknowledged that $k_{600}$ could be over-estimated using AFs."

• The k600 models are actually only valid for a subset of the data. Nevertheless, they are applied for the whole dataset. I would appreciate some thoughts why you think this is still meaningful. What does it mean in terms of generalization or if readers would like to apply the developed models in other regions?

Response: Thanks for your comment. Our model was from a subset of the data, while CO2 flux from our model was in good agreement with the fluxes from FC, determined k and other models when the developed model was applied for the whole dataset (please refer to Tables 2 and 3). We concluded that the model can be used for riverine CO2 flux at catchment scale via the comparison of the fluxes from variable methods though it can not be used at individual site scale. Thus, the model here can be used at catchment scale or regional scale with similar hydrology and topography. In fact, it is hard to test the applicability of models while most studies even used models from other regions. We addressed this issue by adding the following text.

"Our model was from a subset of the data (i.e., Qijiang), while $CO_2$ flux from our model was in good agreement with the fluxes from FC, determined k and other models when the developed model was applied for the whole dataset (please refer to Tables 2 and 3). The comparison of the fluxes from variable methods suggested that the model can be used for riverine $CO_2$ flux at catchment scale though it can not be used at individual site scale. Thus, the model here can be used at catchment scale or regional scale with similar hydrology and topography. Recent studies did not test the applicability of models when $k_{600}$ models from other regions were employed".

• The k600 vs. flow velocity model (Fig. 4b): Sorry, but I cannot follow you there. If "extremely" values are removed (which in my opinion still needs to be justified and clearly described which ones and why), the R2 gets reduced and the p-value gets worse. Please justify and describe the strategy of removing data points. If this cannot be done in an appropriate manner, I don't see any reason why data points should be removed.

Response: We have discussed this issue (see the second paragraph in section 4.2).
 "The extremely high values (two values of 260 and 274 cm/h) are outside of the global ranges and also considerably higher than $k_{600}$ values in Asian rivers. Furthermore, the revised model (two extremely values 260 and 274 cm/h were excluded) was comparable to the published models (Fig. 4), i.e., models of Ran et al. (2015) (measured with drifting chambers) and Liu et al. (2017) (measured with static chambers in canoe shape), which suggested that exclusion of the two extremely values were reasonable and urgent, this was further supported by the $CO_2$ flux using different approaches (Tables 2 and 3)."

I still think that this is a valuable study which would make a good contribution to the literature, but the above-mentioned points need to be addressed before considering publication in Biogeosciences.

Response: I thank you for your positive comment.

Specific comments

Abstract:
Line 22: Explain the meaning of k600 already here, where you mention it for the first time. In general, I suggest not jumping between k600 and k in the abstract (i.e. stick to k600 after mentioning it for the first time).

Response: We corrected this issue. "gas transfer velocity normalized to a Schmidt number of 600 ($k_{600}$) at a temperature of 20 ℃" was added

Lines 24-26: Please make clear that the derived model for k600 is only based on a subset of the data.

Response: "based on a subset of the data" was added.

Line 26: Add "e.g. lakes" after open waters.

Response: "e.g. lakes" was added.

Line 34: There are k600 models for streams (see e.g. Raymond et al., 2012). Do you mean for the specific regions/watersheds? Please be more specific here.

Response: Corrected.

Introduction:
Line 59: Add "in situ" before "temperature".

Response: Addressed.

Lines 65-66: Please add the information that the standardized k600 is valid for freshwaters.

Response: Addressed.

Lines 97-98: This sentence is not clear to me, please rephrase.

Response: Changed to "Models of k were further developed using hydraulic properties (i.e., flow velocity, water depth) by flux measurements with chambers and TBL model."

Line 108: Please rephrase "diffusive models from other continents", it sounds very vague.

Response: Changed to "diffusive models from other rivers/regions"

Lines 105-115: This is a good and important paragraph, but in my opinion it breaks the flow of the introduction. I suggest implementing/moving it before line 98 (i.e. the new contributions to the literature).

Response: Done.

Materials and methods
Lines 134-135: So if I understood correctly, according to the definition you use, the Daning and Qijiang are rivers and the rest are TGR streams and small rivers? If that is the case, unify the terminology in the complete MS (text, figures, tables), ev. define it already in the introduction, and stick to it. Otherwise, this is confusing.

Response: We defined this in the "Introduction". We classified river systems as follows, "Daning, Qijiang, and the rest are TGR streams and small rivers (abbreviation in TGR rivers)" in the Introduction section.

Line 151: As far as I understood, the nutrients were excluded. Please clarify.

Response: Corrected.

Lines 158-159: Please rephrase/revise the last part of the sentence (i.e. "with an accuracy is better than 0.2%").

Response: Corrected.

Line 167: Sorry, but I still don't really understand what you mean with this sentence. Do you intend to describe the quality of the used solvents and reagents?

Response: We rephrased the sentence. "All the used solvents and reagents in experiments were of analytical-reagent grade."

Line 170: How was flow velocity measured? It plays a major role later on, and I think it is important to know how it was measured.

Response: The following text was added.
"and flow velocity was determined using a portable flow meter LS300-A (China), the meter shows an error of <1.5%."

Line 182: Not sure what you mean/refer to with this sentence, please clarify.

Response: We changed to the text as follows.

   Aqueous $p\mathrm{CO_2}$ was computed from the measurements of pH, total alkalinity, and water temperature using $\mathrm{CO_2}$ System ($k_1$ and $k_2$ are from Millero, 1979) (Lewis et al., 1998). This program can yield high quality data (Li et al., 2013;Li et al., 2012;Borges et al., 2004).

Lines 197-201: How many AC and how many DC measurements were done?

Response: DC measurements are used in sampling sites with low flow conditions, i.e., current velocity of < 0.1 m/s, a total of 6 sites were measured by DC. We provided additional information in the main text.

Lines 200-201: Please add the range of overestimation here.
Response: Addressed.

Lines 212-213: The manufacturer of the EGM specified the 0.95 R2 threshold?

Response: Yes. In fact, our observations always show the liner regressions with $R^2 >$ 0.95.

Lines 246-247: I think this aspect should be addressed in the discussion section in which you discuss the uncertainty of the data (4.4).

Response: Section 4.4 focuses on $pCO_2$ and $k_{600}$ values. I prefer to leave this part here, while I would leave it up to the editor.

Line 256: Do you mean the TBL model? Please be consistent with the terminology.

Response: "TBL" was added.

Lines 262-264: So the data of the other large river (Daning) could not be used at all? Please make this clear.

Response: We clearly stated as follows.
"k models were obtained by water depth using data from the TGR rivers, while by flow velocity in the Qijiang, whilst, models were not developed for Daning and combined data."

Results:
For results in general: Please indicate the absolute value of p, i.e. not only < 0.05.

Response: Corrected.

Line 274: Significantly lower? Please be precise.

Response: We have changed "Much lower" to "Significantly lower"

Lines 286-290: To me is not clear, where DOC is significantly higher. Please rephrase this sentence.

Response: we rephrased the text as follows.

"There was significantly higher concentration of dissolved organic carbon (DOC) in the TGR rivers (12.83 ±7.16 mg/l) (p<0.001; Fig. S3) than Daning and Qijiang Rivers. Moreover, Qijiang showed significantly higher concentration of DOC than Daning (3.76 ±5.79 vs 1.07 ±0.33 mg/l in Qijiang and Daning) (p<0.001 by Mann-Whitney Rank Sum Test; Fig. S3)."

Lines 314-320: This seems contradictory to me: No significant relationship with current velocity using the "entire" data set, but significant relationship of flow velocity using "combined" data. What is the difference between current velocity and flow velocity, and between "entire" and "combined" data? Please make this clear.

Response: We are sorry for this mistake. We rephrased the text as follows because that no significant relations between $k_{600}$ and flow velocity while they have slightly linear correlations. It means that flow velocity more or less contributes to $k_{600}$ using the combined data.
"Contrary to our expectations, no significant relationship was observed between $k_{600}$ and water depth, and current velocity using the entire data in the three river systems (TGR streams and small rivers, Danning and Qjiang) (Fig. S4). There were not statistically significant relationships between $k_{600}$ and wind speed using separated data or combined data. Flow velocity showed slightly linear relation with $k_{600}$, and the extremely high value of $k_{600}$ was observed during the periods of higher flow velocity (Fig. S4a) using combined data."

Discussion:
For the discussion: In terms of the desired funnel shape (from detailed to broad), I suggest starting the discussion with 4.4 (Uncertainty assessment of pCO2 and flux-derived k600 values).

Response: The section 4.4 is now moved to be Section 4.1.

Lines 351-359: Thanks for adding a paragraph discussing the chemical enhancement. Nevertheless, I got a bit confused: did you actually calculate the chemical enhancement? ("…of sampling sites that were strongly affected by chemical enhancement…") And how did you come up with 1.7%? Where did u make the cut-off in terms of pH? Please make this clearer by e.g. having a look at Alshboul Z, Lorke A (2015) Carbon Dioxide Emissions from Reservoirs in the Lower Jordan Watershed. PLoS ONE 10(11): e0143381. doi:10.1371/journal.pone.0143381.

Response: We have looked at the article and this citation is included. Correspondingly, we revised this part as follows.

"Higher pH levels were observed in Daning and Qijiang river systems (p<0.05 by Mann-Whitney Rank Sum Test), where more carbonate rock exists that are characterized by karst terrain. Our pH range was comparable to the recent study on the karst river in China (Zhang et al., 2017). Quite high values (8.39 ±0.29, ranging between 7.47 and 9.38; 95% confidence interval: 8.33-8.44) could increase the importance of the chemical enhancement, nonetheless, few studies did take chemical enhancement into account (Wanninkhof and Knox, 1996; Alshboul and Lorke, 2015). The chemical enhancement can increase the $CO_2$ areal flux by a factor of several folds in lentic systems with low gas transfer velocity, whist enhancement factor decreased quickly as $k_{600}$ increased (Alshboul and Lorke, 2015). Our studied rivers are located in mountainous area with high $k_{600}$, which could cause minor chemical enhancement factor. This chemical enhancement of $CO_2$ flux was also reported to be limited in high-pH and also turbulent rivers (Zhang et al., 2017)."

Lines 457-461: Please revise this sentence, to me it is quite hard to understand.

Response: We rewrote this part as follows.
The $CO_2$ evasion comparison by variable approaches also implied that the original flow velocity based model (two extremely $k_{600}$ values were included; Fig. 4b) largely over-estimated the $CO_2$ fluxes, i.e., 1.66 ±1.55 (1.08-2.23) Tg $CO_2$, was 2.3-3 fold higher than other estimations (Table 3b), and our earlier evasion using TBL on the TGR river networks (Li et al., 2018).

Tables
Table 2: I thought if combined data are used then the models do not work? This seems contradictory with the table caption. Furthermore, I do not understand the meaning of the first header in relation to the categories below. Table footnote c: Revised how? By taking out extreme values? Please add some additional information here.

Response: We ADDRESSED the issue in the main text and also in the caption of Fig. 4.
Response: We have discussed applicability and extension of the model in the main text and in the caption of Fig.4 (also see the second paragraph in section 4.2).

We revised Table 2 and the footnotes were revised as follows.

CI-Confidence Interval

[a]Flow velocity – based model is from a subset of the data (please refer to Fig. 4)

[b]Mean value determined using floating chambers (FC).

c-This figure is revised to be 49.6 cm/h if the model ($k_{600} = 62.879FV + 6.8357$, $R^2 = 0.52$, p=0.019) is used (the model is obtained by taking out two extremely values; please refer to Fig. 4c), and the corresponding $CO_2$ areal flux is $203 \pm 190$ mmol/m$^2$/d.

Table 3a: Please add also the standard deviation/CI for the annual emission to be transparent in terms of uncertainty.

Response: "Standard deviation of areal flux" can reflect the uncertainty of CO2 emission.

Table 3b: Please add the information that you also present emission data here, I guess in Tg CO2/y.

Response: The estimated CO2 emission indicated the evasion during the monsoonal period of May through Oct based on the suggestion from referees and editor
"(Tg $CO_2$ during May through October)" was added.

Technical corrections:

Lines 124-125: Replace "concentrated in April through September" with "concentrated between April and September".

Response: Revised.

Line 128: Add "data of" before 48.

Response: Revised.

Line 208: Delete P0 and T0.

Response: Revised.

Line 342: Change "lower than one third" to "one third lower".

Response: Revised.

[revised manuscript text omitted]

The  higher concentration of in the TGR rivers (12.83 ±7.16 mg/l) than Daning and Qijiang

Rivers (3.76 ±5.79 vs 1.07 ±0.33 mg/l in Qijiang and Daning) (p<0.001

; Fig. S3). Moreover, Qijiang showed significantly higher concentration of DOC than Daning (3.76 ±5.79 vs 1.07 ±

0.33 mg/l in Qijiang and Daning) (p<0.001 by Mann-Whitney Rank Sum Test; Fig.

S3).

**3.2. CO$_2$ flux using floating chambers**

The calculated CO$_2$ areal fluxes were higher in TGR rivers (217.7 ±334.7

mmol/m$^2$/d, n = 35), followed by Daning (122.0 ±239.4 mmol/m$^2$/d, n = 28) and

Qijiang rivers (50.3 ±177.2 mmol/m$^2$/d, n = 32) (Fig. 2). The higher CO$_2$ evasion from the TGR rivers is consistent with high riverine $p$CO$_2$ levels. The mean CO$_2$

emission rate was 133.1 $\pm$269.1 mmol/m$^2$/d (n = 95) in all three rivers sampled. The mean $CO_2$ flux differed significantly between TGR rivers and Qijiang (Fig. 2).

**3.3. k levels**

A total of 64 data were used (10 for Daning River, 33 for TGR rivers and 21 for

Qijiang River) to develop k model after removal of samples with $\triangle pCO_2$ less than 110

μatm (Table 2). No significant variability in $k_{600}$ values were observed among the three rivers sampled (Fig. 3). The mean $k_{600}$ (unit in cm/h) was relatively higher in

Qijiang (60.2 $\pm$78.9), followed by Daning (50.2 $\pm$20.1) and TGR rivers (40.4 $\pm$37.6), while the median $k_{600}$ (unit in cm/h) was higher in Daning (50.5), followed by TGR

rivers (30.0) and Qijiang (25.8) (Fig. 3; Table S1). Combined $k_{600}$ data were averaged to 48.4 $\pm$53.2 cm/h (95% CI: 35.1-61.7), and it is 1.5-fold higher than the median value (32.2 cm/h) (Fig. 3).

Contrary to our expectations, no significant relationship was observed between

$k_{600}$ and water depth, and current velocity using the entire data in the three river systems (TGR streams and small rivers, Danning and Qjiang) (Fig. S4). There were not statistically significant relationships between $k_{600}$ and wind speed using separated data or combined data. Flow velocity showed slightly linear relation with $k_{600}$, and the extremely high value of $k_{600}$ was observed during the periods of higher flow velocity (Fig. S4a) using combined data. Similar trend was also observed between water depth and $k_{600}$ values (Fig. S4b). $k_{600}$ as a function of water depth was obtained in the TGR

rivers, but it explained only 30% of the variance in $k_{600}$. However, model using data from Qijiang could explain 68% of the variance in $k_{600}$ (Fig. 4b), and it was in line with general theory.

**4. Discussion**

**4.1. Uncertainty assessment of $p$CO$_2$ and flux-derived k$_{600}$ values**

The uncertainty of flux-derived k values mainly stem from $\Delta p$CO$_2$ (unit in ppm)

and flux measurements (Bodmer et al., 2016;Golub et al., 2017;Lorke et al., 2015).

Thus we provided uncertainty assessments caused by dominant sources of uncertainty from measurements of aquatic $p$CO$_2$ and CO$_2$ areal flux since uncertainty of atmospheric CO$_2$ measurement could be neglected.

In our study, aquatic $p$CO$_2$ was computed based on pH, alkalinity and water temperature rather than directly measured. Recent studies highlighted $p$CO$_2$

uncertainty caused by systematic errors over empiric random errors (Golub et al.,

2017). Systematic errors are mainly attributed to instrument limitations, i.e., sondes of pH and water temperature. The relative accuracy of temperature meters was $\pm0.1\ ^0$C

according to manufacturers' specifications, thus the uncertainty of water T propagated on uncertainty in $p$CO$_2$ was minor (Golub et al., 2017). Systematic errors therefore stem from pH, which has been proved to be a key parameter for biased $p$CO$_2$

estimation calculated from aquatic carbon system (Li et al., 2013;Abril et al., 2015).

We used a high accuracy of pH electrode and the pH meters were carefully calibrated using CRMs, and *in situ* measurements showed an uncertainty of $\pm0.01$. We then run an uncertainty of $\pm0.01$ pH to quantify the $p$CO$_2$ uncertainty, and an uncertainty of $\pm3\%$

[revised manuscript text omitted]

---

## Author Response (AR3)

**Associate Editor Decision: Publish subject to technical corrections** (16 Jan 2019)
by David Butman
Comments to the Author:
Dear Authors, please see the attached document. I am pleased to see this nearly ready for publication. You have appeared to address all comments well.

Response: We have carefully checked the Ms.

A couple points.

1. Please re-read for grammatical errors - I found some but there are more as I was unable to go through the entire document closely.
2. Please be sure that your expression of unit - Tg $CO_2$ etc. reflects the actual unit you are measuring. Are you saying mass of $CO_2$ or mass of C... this is important and can be confusing.
3. Please try and be consistent with font sizes in tables that will be included in the main text.
Other technical comments are provided in the version I have provided back.

Response: Thanks for your suggestions, David Butman. We have carefully checked the Ms and addressed all concerns.